# Sample-Efficient Distributionally Robust Multi-Agent Reinforcement Learning via Online Interaction

**Zain Ulabedeen Farhat**[1,*] **, Debamita Ghosh**[1,*]**, George K. Atia**[1,2]**, Yue Wang**[1,2]
[1] Department of Electrical & Computer Engineering    [2] Department of Computer Science
University of Central Florida, Orlando, FL 32816, USA
{za464241, de881780, george.atia, yue.wang}@ucf.edu

## Abstract

Well-trained multi-agent systems can fail when deployed in real-world environments due to model mismatches between the training and deployment environments, caused by environment uncertainties including noise or adversarial attacks. Distributionally Robust Markov Games (DRMGs) enhance system resilience by optimizing for worst-case performance over a defined set of environmental uncertainties. However, current methods are limited by their dependence on simulators or large offline datasets, which are often unavailable. This paper pioneers the study of online learning in DRMGs, where agents learn directly from environmental interactions without prior data. We introduce the *Multiplayer Optimistic Robust Nash Value Iteration (MORNAVI)* algorithm and provide the first provable guarantees for this setting. Our theoretical analysis demonstrates that the algorithm achieves low regret and efficiently finds the optimal robust policy for uncertainty sets measured by Total Variation divergence and Kullback-Leibler divergence. These results establish a new, practical path toward developing truly robust multi-agent systems.

## 1 Introduction

Multi-agent reinforcement learning (MARL), along with its stochastic game-based mathematical formulation (Shapley, 1953; Littman, 1994), has emerged as a cornerstone paradigm for intelligent multi-agent systems capable of complex, coordinated behavior. It provides the theoretical and algorithmic foundation for enabling multiple agents to learn, adapt, and make sequential decisions in shared, dynamic environments. Its practical impacts span strategic gaming, where MARL agents have achieved superhuman mastery (Silver et al., 2016; Vinyals et al., 2019); autonomous transportation, where it is used to coordinate fleets of vehicles to navigate complex traffic scenarios (Shalev-Shwartz et al., 2016; Hua et al., 2025); and distributed robotics, where teams of robots learn to execute tasks (Lowe et al., 2017; Matignon et al., 2012).

Despite the remarkable progress in MARL, a fundamental and pervasive challenge severely restricts its reliable deployment in practice: the *Sim-to-Real* gap (Zhao et al., 2020; Peng et al., 2018). A standard pipeline of RL involves training extensively within a high-fidelity simulator and then deploying in practice. However, any simulator inevitably fails to capture the full richness and complexity of the real world, omitting subtle physical effects, unpredictable sensor noise, unmodeled system dynamics, or latent environmental factors (Padakandla et al., 2020; Rajeswaran et al., 2017). Consequently, a policy that appears optimal within the simulation can be brittle and perform poorly—or even fail catastrophically—when deployed into the noisy, unpredictable environment.

This vulnerability to model mismatch is magnified exponentially in the multi-agent context: this uncertainty is amplified through a cascading feedback loop of agent interactions. A minor, unmodeled perturbation that affects one agent can cause it to deviate from its expected behavior. This deviation alters the environment for its peers, who in turn must adapt their policies. Their adaptations further change the dynamics for all other agents, including the one first affected. This can trigger a chain of

---
*The first two authors contributed equally.

unpredictable responses, destabilizing the collective strategy and leading to a highly non-stationary learning environment far more volatile than that caused by strategic adaptation alone (Papoudakis et al., 2019; Canese et al., 2021; Wong et al., 2023). The entire multi-agent system becomes fragile, as the intricate inter-agent dependencies act as amplifiers for even the smallest model inaccuracies.

To enable MARL against such uncertainty, the framework of Distributionally Robust Markov Games (DRMGs) offers a principled and powerful solution (Zhang et al., 2020; Kardeş et al., 2011). DRMG approach embraces a principle of pessimism. It defines an uncertainty set of plausible environment models centered around the nominal one, and the goal is to maximize the worst-case expected returns across the entire uncertainty set. This robust optimization strategy yields two profound benefits. First, it provides a formal performance guarantee: if the true environment lies within the uncertainty set, the policy's performance is guaranteed to be no worse than the optimized worst-case value. Second, it acts as a powerful regularizer, forcing agents to discover more generalizable policies that are inherently less sensitive to perturbations, thereby enhancing generalization even to environments outside the set (Vinitsky et al., 2020; Abdullah et al., 2019; Liu et al., 2025).

However, despite its theoretical appeal, the current body of research on DRMGs is built upon assumptions that create a critical disconnect from the realities of many high-stakes applications. The prevailing algorithmic frameworks fall into two main categories: those that assume access to a generative model (Shi et al., 2024; Jiao & Li, 2024), which is tantamount to having a perfect, queryable oracle or simulator, and those designed for the offline setting (Li et al., 2025; Blanchet et al., 2023), which presuppose the existence of a large, static, and sufficiently comprehensive dataset collected beforehand. These assumptions are untenable in precisely the domains where robustness is most crucial. Consider applications in autonomous systems (Demontis et al., 2022) or personalized healthcare (Wang et al., 2018; Lu et al., 2021). In these settings, creating a high-fidelity simulator is often impossible, and pre-collecting a dataset that covers all critical scenarios is infeasible. Agents have no choice but to learn online, through direct, sequential interaction with the complex and unknown real world. In this online paradigm, data is not a free commodity to be sampled at will; it is earned through experience, where every action has a real cost and naive exploration can lead to severe or irreversible outcomes. This necessitates a new class of algorithms that can navigate the exploration-exploitation tradeoff under the additional burden of worst-case environmental uncertainty.

We aim for robustness that survives contact with reality: agents must cope with misspecification while learning purely from experience. Without simulators or sizable offline datasets, existing approaches struggle to bridge theory and practice. This shortfall clarifies the gap we address and motivates the central question of our work: ***How to design provably effective online algorithms for distributionally robust Markov games?***

In this paper, we answer the above question by designing a model-based online algorithm for DRMGs and providing corresponding theoretical guarantees. Our contributions are summarized as follows.

**Hardness in Online DRMGs:** We first reveal the inherent hardness of online learning in DRMGs. Specifically, we first show that the online learning can suffer from the support shifting issue, where the support of the worst-case kernel is not fully covered by the support of the nominal environment, by constructing a hard instance that incurs an $\Omega\big(K \min\{H, \prod_i A_i\}\big)$-regret for any algorithm. Moreover, we use another example to show that even without the support shifting issue, the regret can still have a minimax lower bound of $\Omega(\sqrt{K \prod_i A_i})$. Here, $K$ is the number of iteration episodes, $H$ is the DRMG horizon, and $\prod_i A_i$ is the size of the joint action space. These results directly imply the hardness of online learning, compared to other well-posed learning schemes, including generative model (Shi et al., 2025; Jiao & Li, 2024) or offline learning (Li et al., 2025).

**A Framework for Online Robust MARL:** We introduce $f$-MORNAVI, a novel model-based meta-algorithm designed specifically for online learning in DRMGs. Our framework pioneers a dual approach that synergizes the *pessimism* required for robust optimization with the *optimism* essential for provably efficient online exploration. At its core, $f$-MORNAVI learns the nominal environment model from online interactions and then incorporates a carefully constructed, data-driven bonus term, $\beta$. This bonus term is uniquely tailored to the geometry of the chosen uncertainty set, guiding exploration while guaranteeing that the learned policy is robust to worst-case model perturbations. We further present two concrete instantiations of our framework for uncertainty sets defined by Total Variation (TV) distance and Kullback-Leibler (KL) divergence.

**Near-Optimal Regret Bounds for Online DRMGs:** We establish the first known theoretical guarantees for online learning in general-sum DRMGs by providing rigorous, high-probability regret bounds for our algorithms. The regret measures the performance gap between our algorithm and an optimal robust policy, thus formally characterizing the sample complexity needed to solve the DRMG. We futher prove that our algorithms converge to an $\epsilon$-optimal robust policy with high sample efficiency (see Corollary 6). Our results are significant as they are the first to demonstrate that finding a robust equilibrium in a general-sum DRMG is achievable in a sample-efficient manner through online interaction, without requiring a simulator or a pre-collected dataset.

## 2 PROBLEM FORMULATION

### 2.1 DISTRIBUTIONALLY ROBUST MARKOV GAMES

A *Distributionally Robust Markov Game* (DRMG) can be specified as $\mathcal{MG}_{\text{rob}} = \{\mathcal{M}, \mathcal{S}, \mathcal{A}, H, \{\mathcal{P}_i\}_{i \in \mathcal{M}}, r\}$, where $\mathcal{M} = \{1, ..., m\}$ is the set of $m$ agents, $\mathcal{S} = \{1, 2, \ldots, S\}$ denotes the finite state space, $\mathcal{A}$ denotes the joint action space for all agents as $\mathcal{A} = \mathcal{A}_1 \times \cdots \times \mathcal{A}_m$, where $\mathcal{A}_i = \{1, 2, \ldots, A_i\}$ is the action space of agent $i$, and $H$ denotes the horizon length. We consider non-stationary DRMGs, i.e., $r$ is the reward function: $r = \{r_{i,h}\}_{1 \le i \le m, 1 \le h \le H}$ with $r_{i,h} : \mathcal{S} \times \mathcal{A} \mapsto [0, 1]$. Specifically, for any $(i, h, s, \mathbf{a}) \in \mathcal{M} \times [H] \times \mathcal{S} \times \mathcal{A}$, $r_{i,h}(s, \mathbf{a})$ is the immediate (deterministic) reward received by the $i$-th agent in state $s$ when the joint action profile is $\mathbf{a}$. Agents in a DRMG maintain their own uncertainty sets of transition kernels $\mathcal{P}_i$, to capture the potential environment uncertainties in their perspective. At each step, the environment transits following an arbitrary kernel from the uncertainty set.

Drawing inspiration from the rectangularity condition in robust single-agent RL (Iyengar, 2005; Wiesemann et al., 2013; Zhou et al., 2021; Shi et al., 2023), and following standard DRMG studies (Shi et al., 2024; 2025; Zhang et al., 2020), we consider the *agent-wise $(s, \boldsymbol{a})$-rectangular uncertainty set*, due to its computational tractability[1]. Specifically, for each agent $i$, the DRMG specify an uncertainty set $\mathcal{P}_i$, which is independently defined over all horizons, states, and joint actions:

$$\mathcal{P}_i = \bigotimes_{(h,s,\mathbf{a}) \in [H] \times \mathcal{S} \times \mathcal{A}} \mathcal{P}_{i,h,f}^{\rho_i}(s, \mathbf{a}), \tag{1}$$

where $\otimes$ denotes the Cartesian product. At step $h$, if all agents take a joint action $\mathbf{a_h}$ at the state $s_h$, the transition kernel can be chosen arbitrarily from the prescribed uncertainty set $\mathcal{P}_{i,h,f}^{\rho_i}(s_h, \mathbf{a_h})$. We consider the uncertainty set $\mathcal{P}_{i,h,f}^{\rho_i}(s, \mathbf{a})$ centered on a *nominal kernel* $P^\star$:

**Definition 1** ($f$-Divergence Uncertainty Set). The $f$-divergence uncertainty set is defined as:

$$\mathcal{P}_{i,h,f}^{\rho_i}(s, \mathbf{a}) = \left\{ P_h \in \Delta(\mathcal{S}) : f\left(P_h, P_h^\star(\cdot|s, \mathbf{a})\right) \le \rho_i \right\}, \tag{2}$$

where the $f$-divergence is $f\left(P_h, P_h^\star(\cdot|s, \boldsymbol{a})\right) = \sum_{s' \in \mathcal{S}} f\left(\frac{P_h(s')}{P_h^\star(s'|s,\boldsymbol{a})}\right) P_h^\star(s'|s, \boldsymbol{a})$.

The $f$-divergence uncertainty sets with different $f$ have been extensively studied in distributionally robust RL (Clavier et al., 2023; Shi et al., 2023; Panaganti et al., 2022; Yang et al., 2022; Wang et al., 2024d; Zhang et al., 2025). In this work, we focus on TV and KL-divergence.

**Robust Value Functions.** For a DRMG, each agent aims to maximize its own worst-case performance over all possible transition kernels in its own (possibly different) prescribed uncertainty set. The strategy of agent $i$ taking actions is captured by a policy $\pi_i = \{\pi_{i,h} : \mathcal{S} \to \Delta(\mathcal{A}_i)\}_{h=1}^H$. Since the immediate rewards and transition kernels are determined by the joint actions, the worst-case performance of the $i$-th agent over its own uncertainty set $\mathcal{P}_i$ is determined by a joint policy $\pi = \{\pi_h : \mathcal{S} \to \Delta(\mathcal{A})\}_{h=1}^H$, which we refer to as the robust value function $V_{i,h}^{\pi,\rho_i}$ and the robust $Q$-function $Q_{i,h}^{\pi,\rho_i}$, for an initial state $s$ and initial action $\mathbf{a}$: $Q_{i,h}^{\pi,\rho_i}(s, \mathbf{a}) \triangleq$

$\inf_{\tilde{P} \in \mathcal{P}_i} \mathbb{E}_{\pi, \tilde{P}} \left[ \sum_{t=h}^H r_{i,t}(s_t, \mathbf{a}_t) \ \middle| \ s_h = s, \mathbf{a}_h = \mathbf{a} \right]$, and $V_{i,h}^{\pi,\rho_i}(s) \triangleq \sum_{\mathbf{a}} \pi(\mathbf{a}|s) Q_{i,h}^{\pi,\rho_i}(s, \mathbf{a})$,

where the expectation is taken over the randomness of the joint policy $\pi$ and the kernel $\tilde{P}$.

---

[1] Robust MDPs without rectangular assumption can be NP-hard to solve (Wiesemann et al., 2013).

**Solutions to DRMGs.** Due to different objectives among players, the goal of a DRMG is to achieve some notions of equilibrium (Fudenberg & Tirole, 1991). For any given joint policy $\pi$, $\pi_{-i}$ is the marginal policies of all agents excluding the $i$-th agent. The agent $i$'s best response policy to $\pi_{-i}$, $\pi_i^{\dagger,\rho_i}(\pi_{-i})$, is the policy that maximizes its own robust value function, at the given step $h$ and state $s$: $\pi_i^{\dagger,\rho_i}(\pi_{-i}) \triangleq \arg\max_{\pi_i' \in \Delta(\mathcal{A}_i)} V_{i,h}^{(\pi_{-i} \times \pi_i'),\rho_i}(s)$. The corresponding robust value function is

$$V_{i,h}^{\dagger,\pi_{-i},\rho_i}(s) \triangleq \max_{\pi_i' \in \Delta(\mathcal{A}_i)} V_{i,h}^{\pi_i' \times \pi_{-i},\rho_i}(s). \tag{3}$$

The goal of a DRMG is to compute an equilibrium policy (Fudenberg & Tirole, 1991), such that each agent's policy is the best response to the others, so that no single agent can improve its robust value by deviating while the rest remain fixed. Standard notions of equilibria include *robust Nash Equilibrium (NE)*, *robust Coarse Correlated Equilibrium (CCE)*, and *robust Correlated Equilibrium (CE)* (their existence is shown (Blanchet et al., 2023)), defined as follows:

**Robust $\varepsilon$-NE.** A product policy $\pi \in \Delta(\mathcal{A}_1) \times \cdots \times \Delta(\mathcal{A}_m)$ is a *robust-$\varepsilon$ NE* if for any $s \in \mathcal{S}$: $\text{gap}_{\text{NE}}(\pi, s) \triangleq \max_{i \in \mathcal{M}} \left\{ V_{i,1}^{\dagger,\pi_{-i},\rho_i}(s) - V_{i,1}^{\pi,\rho_i}(s) \right\} \le \varepsilon$.

Robust NE ensures that, the agent $i$'s policy induced by the NE is a best response policy to the remaining agents' joint policy (up to $\epsilon$), thus no agent can improve its worst-case performance—evaluated over its own uncertainty set $\mathcal{P}_i$—by unilaterally deviating from the NE.

**Robust $\varepsilon$-CCE.** A (possibly correlated) joint policy $\pi \in \Delta(\mathcal{A})$ is a *robust-$\varepsilon$ CCE* if for any $s \in \mathcal{S}$: $\text{gap}_{\text{CCE}}(\pi, s) \triangleq \max_{i \in \mathcal{M}} \left\{ V_{i,1}^{\dagger,\pi_{-i},\rho_i}(s) - V_{i,1}^{\pi,\rho_i}(s) \right\} \le \varepsilon$. Robust CCE relaxes the notion of NE by allowing for potentially correlated policies, while still ensuring that no agent has an incentive to unilaterally deviate from it.

**Robust $\varepsilon$-CE.** A joint policy $\pi \in \Delta(\mathcal{A})$ is a *robust-$\varepsilon$ CE* if for any $s \in \mathcal{S}$: $\text{gap}_{\text{CE}}(\pi, s) \triangleq \max_{i \in \mathcal{M}} \left\{ \max_{\phi \in \Phi_i} V_{i,1}^{\phi \diamond \pi,\rho_i}(s) - V_{i,1}^{\pi,\rho_i}(s) \right\} \le \varepsilon$. Here, a strategy modification $\phi \triangleq \{\phi_{h,s}\}_{(h,s) \in [H] \times \mathcal{S}}$ for player $i$ is a set of $[H] \times \mathcal{S}$ functions from $\mathcal{A}_i$ to itself. Let $\Phi_i$ denote the set of all possible strategy modifications for player $i$. Given a joint policy $\pi$, applying a modification $\phi$ yields a new joint policy $\phi \diamond \pi$, which matches $\pi$ everywhere except that at each state $s$ and timestep $h$, player $i$'s action $a_i$ is replaced by $\phi_{h,s}(a_i)$.

**Online Learning in DRMGs.** We consider online learning in DRMGs, aiming to compute equilibria $\{\text{NASH}, \text{CCE}, \text{CE}\}$ via interaction with the nominal environment $P^\star$ over $K \in \mathbb{N}$ episodes. Each episode starts from $s_1^k$, proceeds with a policy $\pi^k$ chosen from experience, and ends with an update for the next round. We use *robust regret* as our performance metric, which compares the learned outcome to the target equilibrium in the presence of model error.

**Definition 2** (Robust Regret). Let $\pi^k$ be the execution policy in the $k^{\text{th}}$ episode. After a total of $K$ episodes, the corresponding robust regret is defined as $\text{Regret}_{\{\text{NASH},\text{CCE},\text{CE}\}}(K) = \sum_{k=1}^K \text{gap}_{\{\text{NASH},\text{CCE},\text{CE}\}}(\pi^k, s_1^k)$.

Notably, if an algorithm has a sub-linear regret, it achieves a robust equilibrium as $K \to \infty$.

## 3 OPTIMISTIC ROBUST NASH VALUE ITERATION

We then present Multiplayer Optimistic Robust Nash Value Iteration for $f$-Divergence Uncertainty Set ($f$-MORNAVI), a meta-algorithm for episodic, finite-horizon DRMGs with interactive data collection. $f$-MORNAVI handles general $f$-divergences, with emphasis on KL and TV.

### 3.1 ALGORITHM DESIGN

Our algorithm has the following three stages.

**Stage 1: Nominal Transition Estimation (Line 4).** At the start of each episode $k \in [K]$, we maintain an estimate of the nominal kernel $P^\star$ using the historical data $\mathbb{D} = \{(s_h^\tau, \mathbf{a}_h^\tau, s_{h+1}^\tau)\}_{\tau=1,h=1}^{k-1,H}$

---

**Algorithm 1:** $f$-MORNAVI

---

1: **Input:** Uncertainty level $\rho_i > 0$ for all $i \in \mathcal{M}$.
2: **Initialize:** Dataset $\mathbb{D} = \emptyset$
3: **for** episode $k = 1, \ldots, K$ **do**
4:     Compute the transition kernel estimator $\widehat{P}_h^k(s, \mathbf{a}, s')$
5:     Set $\overline{V}_{H+1}^{k,\rho_i}(\cdot) = \underline{V}_{H+1}^{k,\rho_i}(\cdot) = 0$ for all $i \in \mathcal{M}$.
6:     **for** step $h = H, \ldots, 1$ **do**
7:         **for** $\forall (s, \mathbf{a}) \in \mathcal{S} \times \mathcal{A}$ **do**
8:             $\overline{Q}_{i,h}^{k,\rho_i}(s, \boldsymbol{a}) = \min\left\{r_{i,h}(s, \boldsymbol{a}) + \sigma_{\widehat{\mathcal{P}}_{i,h,f}^{\rho_i}(s,\boldsymbol{a})}[\overline{V}_{i,h+1}^{k,\rho_i}] + \beta_{i,h,f}^k(s, \boldsymbol{a})\right\}$
9:             $\underline{Q}_{i,h}^{k,\rho_i}(s, \boldsymbol{a}) = \max\left\{r_{i,h}(s, \boldsymbol{a}) + \sigma_{\widehat{\mathcal{P}}_{i,h,f}^{\rho_i}(s,\boldsymbol{a})}[\underline{V}_{i,h+1}^{k,\rho_i}] - \beta_{i,h,f}^k(s, \boldsymbol{a}),\ 0\right\}$
10:         **end for**
11:         **for** $\forall s \in \mathcal{S}$ **do**
12:             $\pi_h^k(\cdot|s) = \text{EQUILIBRIUM}\left(\left\{\overline{Q}_{i,h}^{k,\rho_i}(s, \cdot)\right\}_{i \in \mathcal{M}}\right)$
13:             $\overline{V}_{i,h}^{k,\rho_i}(s) = \mathbb{E}_{\boldsymbol{a} \sim \pi^k(\cdot|s)}\left[\overline{Q}_{i,h}^{k,\rho_i}(s, \mathbf{a})\right], \underline{V}_{i,h}^{k,\rho_i}(s) = \mathbb{E}_{\boldsymbol{a} \sim \pi^k(\cdot|s)}\left[\underline{Q}_{i,h}^{k,\rho_i}(s, \mathbf{a})\right]$
14:         **end for**
15:     **end for**
16:     Receive initial State $s_1^k \in \mathcal{S}$
17:     **for** step $h = 1, \ldots, H$ **do**
18:         Take action $\mathbf{a}_h^k \sim \pi_h^k(\cdot \mid s_h^k)$, observe reward $r_h(s_h^k, \mathbf{a}_h^k)$ and next state $s_{h+1}^k$.
19:     **end for**
20:     Set $\mathbb{D} = \mathbb{D} \cup \{(s_h^k, \mathbf{a}_h^k, s_{h+1}^k)\}_{h=1}^H$.
21: **end for**
22: **Output:** Return policy $\pi^{\text{out}} = \{\pi^k\}_{k=1}^K$.

---

collected from past interactions with the training environment. Specifically, $f$-MORNAVI updates the empirical transition kernel for each tuple $(h, s, \mathbf{a}, s') \in [H] \times \mathcal{S} \times \mathcal{A} \times \mathcal{S}$ as follows:

$$\widehat{P}_h^k(s'|s, \mathbf{a}) = \frac{N_h^k(s, \mathbf{a}, s')}{N_h^k(s, \mathbf{a})}(\text{if } N_h^k(s, \mathbf{a}) > 0), \text{ and } \widehat{P}_h^k(s'|s, \mathbf{a}) = \frac{1}{|\mathcal{S}|}(\text{if } N_h^k(s, \mathbf{a}) = 0), \quad (4)$$

where $N_h^k(s, \mathbf{a}, s')$ and $N_h^k(s, \mathbf{a})$, are calculated on the current dataset $\mathbb{D}$ by $N_h^k(s, \mathbf{a}, s') = \sum_{\tau=1}^{k-1} \mathbf{1}\{(s_h^\tau, \mathbf{a}_h^\tau, s_{h+1}^\tau) = (s, \mathbf{a}, s')\}$, and $N_h^k(s, \mathbf{a}) = \sum_{s' \in \mathcal{S}} N_h^k(s, \mathbf{a}, s')$. Note that we adopt a model-based approach that estimates transition kernels. Although this leads to higher memory consumption, model-free DRMGs are inherently challenging due to the non-linearity of worst-case expectation w.r.t. nominal kernels, which makes model-free estimators biased or sample-inefficient (Liu et al., 2022; Wang et al., 2023d; 2024c; Zhang et al., 2025).

**Stage 2: Optimistic Robust Planning (Lines 5–9).** The $f$-MORNAVI constructs the episode policy $\pi^k$ via optimistic robust planning based on the empirical model $\widehat{P}^k$. This involves estimating an upper bound on the robust value function, following the principle of Upper-Confidence-Bound (UCB) methods, which are well-established in online vanilla RL (Auer & Ortner, 2010; Azar et al., 2017; Zanette & Brunskill, 2019; Zhang et al., 2021b; Ménard et al., 2021; Zhang et al., 2024b), and this optimism encourages exploration of less-visited state–action pairs.

To this end, $f$-MORNAVI maintains a bonus term at each episode $k$, capturing the gap between the robust value function under $\widehat{P}^k$ and that under the true model. This bonus is added to the robust Bellman estimate to ensure its optimism. Specifically, for each $(h, s, \mathbf{a}) \in [H] \times \mathcal{S} \times \mathcal{A}$, we set

$$\overline{Q}_{i,h}^{k,\rho_i}(s, \boldsymbol{a}) = \min\left\{r_{i,h}(s, \boldsymbol{a}) + \sigma_{\widehat{\mathcal{P}}_{i,h,f}^{\rho_i}(s,\boldsymbol{a})}[\overline{V}_{i,h+1}^{k,\rho_i}] + \beta_{i,h,f}^k(s, \boldsymbol{a}),\ H\right\}. \quad (5)$$

$$\underline{Q}_{i,h}^{k,\rho_i}(s, \boldsymbol{a}) = \max\left\{r_{i,h}(s, \boldsymbol{a}) + \sigma_{\widehat{\mathcal{P}}_{i,h,f}^{\rho_i}(s,\boldsymbol{a})}[\underline{V}_{i,h+1}^{k,\rho_i}] - \beta_{i,h,f}^k(s, \boldsymbol{a}),\ 0\right\}. \quad (6)$$

Here, $\sigma_{\mathcal{P}}[V] = \inf_{P \in \mathcal{P}} \mathbb{E}_P[V]$ is the support function of $V$ over the uncertainty set $\mathcal{P}$, and can be calculated through its dual representation (see Lemma 7); $\widehat{\mathcal{P}}_{i,h,f}^{\rho_i}$ is the uncertainty set centered at $\widehat{P}^k$ from eq. 4: $\widehat{\mathcal{P}}_{i,h,f}^{\rho_i}(s, \mathbf{a}) = \left\{ P_h \in \Delta(\mathcal{S}) : f\left(P_h, \widehat{P}_h^k(\cdot|s, \mathbf{a})\right) \leq \rho_i \right\}$.

Each of these estimates in eq. 5 and eq. 6 are based on estimated robust Bellman operators (see Appendix C for details) and a bonus term $\beta_{i,h,f}^k(s, \boldsymbol{a}) \geq 0$. The bonus term is constructed (we will discuss the construction later) to ensure the estimation becomes a confidence interval of the true robust value function, i.e., $Q_{i,h}^{\dagger, \pi_{-i}, \rho_i}(s, \mathbf{a}) \in [\underline{Q}_{i,h}^{k,\rho_i}(s, \mathbf{a}), \overline{Q}_{i,h}^{k,\rho_i}(s, \mathbf{a})]$, with high probability.

**EQUILIBRIUM subroutine (Line 8).** Given robust $Q$-function estimates $\underline{Q}_{i,h}^{k,\rho_i}(s, \mathbf{a})$ and $\overline{Q}_{i,h}^{k,\rho_i}(s, \mathbf{a})$ for $i \in \mathcal{M}$ at step $h$, the sub-routine EQUILIBRIUM $\in \{\mathsf{NASH}, \mathsf{CCE}, \mathsf{CE}\}$ finds a corresponding equilibrium $\pi_h^k(\cdot|s)$ for the matrix-form game with pay-off matrices $\{\overline{Q}_{i,h}^{k,\rho_i}(s, \cdot)\}_{i \in \mathcal{M}}$:

$$\pi_h^k(\cdot|s) \leftarrow \text{EQUILIBRIUM}\left(\left\{\overline{Q}_{i,h}^{k,\rho_i}(s, \cdot)\right\}_{i \in \mathcal{M}}\right). \tag{7}$$

Note that finding a NE can be PPAD-hard (Daskalakis et al., 2009), but computing CE or CCE remains tractable in polynomial time (Liu et al., 2021). We follow standard MG studies, assuming EQUILIBRIUM can be executed, and mainly focus on sample complexity and statistic efficiency.

We then update the estimation of $V_h^{\dagger, \pi_{-i}, \rho}$ as

$$\overline{V}_{i,h}^{k,\rho_i}(s) = \mathbb{E}_{\boldsymbol{a} \sim \pi^k(\cdot|s)}\left[\overline{Q}_{i,h}^{k,\rho_i}(s, \mathbf{a})\right] \quad \text{and} \quad \underline{V}_{i,h}^{k,\rho_i}(s) = \mathbb{E}_{\boldsymbol{a} \sim \pi^k(\cdot|s)}\left[\underline{Q}_{i,h}^{k,\rho_i}(s, \mathbf{a})\right]. \tag{8}$$

Note that while the lower estimate in eq. 6 does not influence policy execution directly, it plays a crucial role in constructing valid exploration bonuses and ensuring strong theoretical guarantees. By leveraging both upper and lower bounds, the algorithm performs optimistic robust planning, enabling structured, uncertainty-aware exploration that balances exploration, exploitation, and robustness.

**Stage 3: Execution of Policy and Data Collection (Lines 10–16).** After evaluating the policy $\{\pi_h^k\}_{h=1}^H$ for episode $k$, the learner takes action based on $\pi_h^k$ and observes the reward $r_h(s_h^k, \mathbf{a}_h^k)$ and next state $s_{h+1}^k$, which get appended to the historical dataset collected till episode $k - 1$.

## 4 HARDNESS OF ONLINE LEARNING

In this section, we aim to discuss the inherent hardness of online learning in DRMGs from two aspects: (1) When there is a support shift issue, no MARL algorithm can obtain a sub-linear regret on a certain DRMG; (2) Even if there is no support shift issue, there exists a DRMG such that any online algorithm suffers from the curse of multi-agency. This is a separation between DRMGs with interactive data collection and generative model/offline data, and also between DRMGs with non-robust MGs, showing the inherent challenges of online DRMGs.

### 4.1 HARDNESS WITH SUPPORT SHIFT

Support shift (Lu et al., 2024) refers to the case that the support of the worst-case transition kernel is not covered by the support of the nominal kernel. It can happen when, for instance, the uncertainty set is defined through TV. It will result in a challenge that, for those states that are not covered by the nominal kernel, there is no data available, so that the agent can never learn the optimal robust policy efficiently. Specifically, we derive the following result to illustrate the hardness.

**Theorem 1.** *There exists a TV-DRMG, such that any online learning algorithm satisfies that:*

$$\inf_{\mathcal{ALG}} \mathbb{E}[Regret_{NASH}(K)] \geq \Omega\left(\rho K \cdot \min\{H, \prod_{i \in \mathcal{M}} A_i\}\right).$$

Our construction is deferred to Example 10 in Appendix. This regret bound is linear in the number of episodes $K$, creating a combinatorial explosion that makes the problem information-theoretically intractable. Moreover, our result shows that when the game horizon $H$ is large enough, the minimax lower bound depends on the joint action space, showing the hardness of online learning compared to generative models and offline settings.

## 4.2 Hardness without support shift

We then illustrate the hardness of online DRMGs when there is no support shift. Note that when the uncertainty set is defined through, e.g., KL divergence, the worst-case support will be covered by the nominal one, so there will not be any support shift. However, we construct another example to show that, even without the support shift, online learning can still be challenging and inefficient.

**Theorem 2** (Lower Bound for Robust Learning without Support Shift). *There exists a DRMG, such that any learning algorithm suffers the following cumulative regret lower bound over $K$ episodes:*

$$\inf_{\mathcal{ALG}} \mathbb{E}[\text{Regret}_{\text{NASH}}(K)] \geq \Omega\left(\sqrt{K \prod_{i \in \mathcal{M}} A_i}\right).$$

This result illustrates that, even without any support shift, some hard instance can require at least $\Omega\left(\sqrt{K \prod_i A_i}\right)$ regret. Our result thus suggests that the dependence on the joint action space may be inevitable in online DRMGs, which suffer from the curse of multi-agency. Specifically, in DRMGs, agents need to solve the robust optimization (i.e., estimate the support function $\sigma_{\mathcal{P}}(\cdot)$), which requires knowledge of the whole transition kernels to find the worst-case from the uncertainty set. Thus, the agents have to explore the whole model, introducing an inevitable dependence on $\prod_i A_i$. In non-robust MGs, however, agents can estimate the single nominal performance merely from samples instead of model estimations, thus the multi-agency curse can be broken.

## 5 Theoretical Guarantees

### 5.1 Regret Bound for Total Variation

As discussed in Section 4, no efficient algorithm can be expected due to the support shifting issue. We hence adopt a standard fail-state assumption (Lu et al., 2024; Liu et al., 2024) to ensure the worst-case kernel support will be covered by the nominal one, bypassing the issue.

**Assumption 3** (Failure States). *For any agent $i$, there exists an (agent-specified) set of failure states $\mathcal{S}_f^i \subseteq \mathcal{S}$, such that $r_i(s, \boldsymbol{a}) = 0$, and $P_h^\star(s'|s, \boldsymbol{a}) = 0$, $\forall \boldsymbol{a} \in \mathcal{A}, \forall s \in \mathcal{S}_f^i, \forall s' \notin \mathcal{S}_f^i$.*

This assumption is only needed for the TV case. Assumption 3 is a standard assumption in single-agent robust RL studies (Panaganti et al., 2022; Lu et al., 2024), and we adapt it to multi-agent cases.

We then present our threotical guarantees.

**Theorem 4** (Upper bound of TV-MORNAVI). *Denote $\rho_{\min} := \min_{i \in \mathcal{M}} \rho_i$. For any $\delta \in (0, 1)$,*

*we set $\beta_{i,h,f}^k(s, \boldsymbol{a})$ as* $\sqrt{\frac{c_1 \iota \operatorname{Var}_{\widehat{P}_h^k(\cdot|s,\boldsymbol{a})}\left[\frac{\overline{V}_{i,h+1}^{k,\rho_i} + \underline{V}_{i,h+1}^{k,\rho_i}}{2}\right]}{N_h^k(s,\boldsymbol{a}) \vee 1}} + \frac{c_2 H^2 S \iota}{\sqrt{N_h^k(s,\boldsymbol{a}) \vee 1}} + \frac{2\mathbb{E}_{\widehat{P}_h^k(\cdot|s,\boldsymbol{a})}\left[\overline{V}_{i,h+1}^{k,\rho_i} - \underline{V}_{i,h+1}^{k,\rho_i}\right]}{H} +$

$\frac{1}{\sqrt{K}}$, *where* $\iota = \log\left(S^2(\prod_{i=1}^m A_i)H^2 K^{3/2}/\delta\right)$ *and $c_1, c_2$ are absolute constants. Then, under Assumption 3, for* EQUILIBRIUM *being one of* $\{$NASH, CE, CCE$\}$, *with probability at least $1 - \delta$, the regret of our TV-MORNAVI algorithm can be bounded as:* $\text{Regret}_{\{\text{NASH,CCE,CE}\}}(K) =$

$\tilde{\mathcal{O}}\left(\sqrt{\min\{\rho_{\min}^{-1}, H\} H^2 S K \left(\prod_{i \in \mathcal{M}} A_i\right)}\right).$

### 5.2 Regret Bound for KL-Divergence

We then study the regret bound of the KL-divergence set. As discussed, the KL set is free from the support issue hence no additional assumption is required. Our regret bound result is as follows.

**Theorem 5.** *For any $\delta$, set $\beta_{i,h,f}^k(s, \boldsymbol{a})$ in KL-DRMG as* $\frac{2c_f H}{\rho_i} \sqrt{\frac{\iota}{\left(N_h^k(s,\boldsymbol{a}) \vee 1\right) \widehat{P}_{\min,h}^k(s,\boldsymbol{a})}} +$

$\sqrt{\frac{1}{K}}$, *where* $\widehat{P}_{\min,h}^k(s,\boldsymbol{a}) = \min_{s' \in \mathcal{S}}\{\widehat{P}_h^k(s'|s,\boldsymbol{a}) : \widehat{P}_h^k(s'|s,\boldsymbol{a}) > 0\}$, $\iota = \log\left(S^2(\prod_{i=1}^m A_i)H^2 K^{3/2}/\delta\right)$, *and $c_f$ is an absolute constant. Then for* EQUILIBRIUM *being one of* $\{$NASH, CE, CCE$\}$, *with probability at least $1 - \delta$, it holds that*

$$\text{Regret}_{\{\text{NASH,CCE,CE}\}}(K) \quad = \quad \tilde{\mathcal{O}}\left(\sqrt{H^4 \exp(2H^2) KS\left(\prod_{i \in \mathcal{M}} A_i\right)\left(\rho_{\min}^2 P_{\min}^\star\right)^{-1}}\right), \quad where$$

$P_{\min}^\star \triangleq \min_{(s,\boldsymbol{a},s',h):P_h(s'|s,\boldsymbol{a})>0} P(s'|s,\boldsymbol{a})$ *is the smallest positive entry of the nominal kernel.*

We note that the $\exp(H)$ term is inherently from the duality form of the distributionally robust optimization with KL-ball (see equation 12). It is standard in existing robust RL studies under KL settings, and can be directly replaced by $(P_{\min}^\star)^{-1}$ (see, e.g., (Panaganti & Kalathil, 2022; Blanchet et al., 2023; Ghosh et al., 2025; Si et al., 2020; Xu et al., 2023; Zhou et al., 2021)). It reflects the inherent hardness of the KL-based robust RL, and is inevitable in sample complexity. In practice, for moderate horizons, $P_{\min}^\star > 0$, and non-vanishing $\sigma$, these worst-case factors remain controlled and do not pose serious issues.

We then briefly discuss the construction of $\beta$ under the two cases. Recall that in our meta–algorithm $f$-MORNAVI, for each agent $i$, episode $k$ and step $h$, we maintain an optimistic and a pessimistic robust $Q$–estimate $\overline{Q}_{k,i,h}^{\rho_i}(s,a), \underline{Q}_{k,i,h}^{\rho_i}(s,a)$, defined via the empirical robust Bellman operators as in eqs 5-6, and shifted by an exploration bonus $\beta_{i,h,f}^k(s,a) \geq 0$. We use $\sigma_{\mathcal{P}}[V] := \inf_{P \in \mathcal{P}} \mathbb{E}_P[V]$ for the support function over the uncertainty set. The purpose of the bonus is to make these estimates form a tight, uniform high–probability confidence interval around the true robust $Q$–values, i.e.

$$Q_{i,h}^{\dagger,\pi_{-i},\rho_i}(s,a) \in \left[\underline{Q}_{k,i,h}^{\rho_i}(s,a),\ \overline{Q}_{k,i,h}^{\rho_i}(s,a)\right] \quad \text{for all } (i,h,k,s,a). \tag{9}$$

**TV–uncertainty.** For TV–balls, we use the dual representation of the robust Bellman operator in equation 11. Under Assumption 3 (failure states), it holds that $\min_s V(s) = 0$, and the deviation between the true and empirical robust operators at $(h,s,a)$ then decomposes as

$$\left|\sigma_{\mathcal{P}_{\text{TV}}^{\rho_i}(P_h^\star(\cdot|s,a))}[V] - \sigma_{\mathcal{P}_{\text{TV}}^{\rho_i}(\widehat{P}_h^k(\cdot|s,a))}[V]\right| \leq \max_{\eta \in [0,H/\rho_{\min}]} \left|\mathbb{E}_{P_h^\star(\cdot|s,a)}[V_\eta] - \mathbb{E}_{\widehat{P}_h^k(\cdot|s,a)}[V_\eta]\right|.$$

To simultaneously control the estimation error for all $(i,h,k,s,a)$ and all value functions of the form $V = V_{k,i,h+1}^{\rho_i}$ and $\underline{V}_{k,i,h+1}^{\rho_i}$, we utilize the standard $\epsilon$-net (Shi & Chi, 2024; Li et al., 2024a) of the interval $[0, H/\rho_{\min}]$, and construct a Bernstein–type concentration inequality for empirical expectations of the random functions $V_\eta$ as

$$\left|\mathbb{E}_{P_h^\star(\cdot|s,a)}[U] - \mathbb{E}_{\widehat{P}_h^k(\cdot|s,a)}[U]\right| \lesssim \sqrt{\frac{\text{Var}_{\widehat{P}_h^k(\cdot|s,a)}(U)\,\iota}{N_h^k(s,a) \vee 1}} + \frac{H^2\sqrt{S\iota}}{\sqrt{N_h^k(s,a) \vee 1}}, \tag{10}$$

for all $U$ with $\|U\|_\infty \leq H$. In our algorithm we set $U = \frac{\overline{V}_{k,i,h+1}^{\rho_i} + \underline{V}_{k,i,h+1}^{\rho_i}}{2}$, and $\Delta V := \overline{V}_{k,i,h+1}^{\rho_i} - \underline{V}_{k,i,h+1}^{\rho_i}$, which allows us to relate the variance under $P^\star$ and $\widehat{P}^k$ and to control the gap $\mathbb{E}[\Delta V]$ that appears in the robustness amplification term. Combining equation 10 with these comparisons yields

$$\left|\sigma_{\mathcal{P}_{\text{TV}}^{\rho_i}(P_h^\star(\cdot|s,a))}[V_{k,i,h+1}^{\rho_i}] - \sigma_{\mathcal{P}_{\text{TV}}^{\rho_i}(\widehat{P}_h^k(\cdot|s,a))}[V_{k,i,h+1}^{\rho_i}]\right|$$

$$\lesssim \sqrt{\frac{\text{Var}_{\widehat{P}_h^k(\cdot|s,a)}\left[\frac{1}{2}(\overline{V}_{k,i,h+1}^{\rho_i} + \underline{V}_{k,i,h+1}^{\rho_i})\right]\iota}{N_h^k(s,a) \vee 1}} + \frac{H^2\sqrt{S\iota}}{\sqrt{N_h^k(s,a) \vee 1}} + \frac{1}{H}\,\mathbb{E}_{\widehat{P}_h^k(\cdot|s,a)}[\Delta V].$$

This motivates choosing the TV–bonus as $\beta_{i,h,f}^k(s,a) = \sqrt{\frac{c_1\iota\,\text{Var}_{\widehat{P}_h^k(\cdot|s,a)}\left[\frac{1}{2}(\overline{V}_{k,i,h+1}^{\rho_i} + \underline{V}_{k,i,h+1}^{\rho_i})\right]}{N_h^k(s,a) \vee 1}} +$ $\frac{2}{H}\,\mathbb{E}_{\widehat{P}_h^k(\cdot|s,a)}[\Delta V] + \frac{c_2 H^2\sqrt{S\iota}}{\sqrt{N_h^k(s,a) \vee 1}} + \frac{1}{\sqrt{K}}$. With this choice, Lemma 20 shows that equation 9 holds under TV–uncertainty.

**KL–uncertainty.** For KL–balls we again appeal to the dual formulation equation 12. Thus, the robust Bellman operator becomes a *log–moment generating function* of $V$. The key difficulty is that we now need to control the deviation between the true and empirical log–MGFs,

$$\left|-\frac{1}{\lambda}\log\mathbb{E}_{P_h^\star(\cdot|s,a)}\left[\exp(-\lambda V)\right] + \frac{1}{\lambda}\log\mathbb{E}_{\widehat{P}_h^k(\cdot|s,a)}\left[\exp(-\lambda V)\right]\right|,$$

uniformly over all $(i, h, k, s, a)$ and the random value functions $V = V_{k,i,h+1}^{\rho_i}$ generated by the algorithm. We utilize Hoeffding's inequality to derive a self–normalized concentration inequality for empirical MGFs: $\left|\log \mathbb{E}_{P^\star}[e^{-\lambda V}] - \log \mathbb{E}_{\widehat{P}^k}[e^{-\lambda V}]\right| \lesssim \sqrt{\frac{\iota}{(N_h^k(s,a)\vee 1) P_{\min,h}^\star(s,a)}}$.

Multiplying both sides by $H/\rho_i$ (since $\lambda \asymp \rho_i/H$) and using the boundedness $\|V\|_\infty \leq H$ to control higher–order terms in the MGF expansion, we obtain the local deviation

$$\left|\sigma_{\mathcal{P}_{\mathrm{KL}}^{\rho_i}(P_h^\star(\cdot|s,a))}[V] - \sigma_{\mathcal{P}_{\mathrm{KL}}^{\rho_i}(\widehat{P}_h^k(\cdot|s,a))}[V]\right| \lesssim \frac{H}{\rho_i}\sqrt{\frac{\iota}{(N_h^k(s,a) \vee 1) P_{\min,h}^\star(s,a)}}.$$

Since only the support of $P^\star$ matters, and we only observe empirical transitions, we replace $P_{\min,h}^\star(s,a)$ by its empirical counterpart $\widehat{P}_{\min,h}^k(s,a)$, at the cost of an extra factor that is absorbed into the constants (cf. Lemma 31). This leads to the KL–bonus as $\beta_{i,h,f}^k(s,a) = 2c_f \frac{H}{\rho_i}\sqrt{\frac{\iota}{(N_h^k(s,a)\vee 1) \widehat{P}_{\min,h}^k(s,a)}} + \sqrt{\frac{1}{K}}$.

### 5.3 SAMPLE COMPLEXITY

As a direct corollary, we derive the sample complexity to learn an $\varepsilon$-equilibrium. Using a standard online-to-batch conversion (Cesa-Bianchi et al., 2001), we have the following results.

**Corollary 6** (Sample Complexity). *With probability at least $1-\delta$, and under the settings of Theorem 4 and Theorem 5, the number of samples required to find an $\epsilon$-approximate equilibrium is bounded as:*

$$KH = \begin{cases} \tilde{\mathcal{O}}\left(\epsilon^{-2}\min\left\{\rho_{\min}^{-1}, H\right\} H^3 S\left(\prod_{i\in\mathcal{M}} A_i\right)\right), & \text{for TV-DRMG} \\ \tilde{\mathcal{O}}\left(\epsilon^{-2} H^5 \exp(2H^2) S\left(\prod_{i\in\mathcal{M}} A_i\right)\left(\rho_{\min}^2 P_{\min}^\star\right)^{-1}\right), & \text{for KL-DRMG} \end{cases}.$$

Our results hence imply that, despite the inherent hardness of online learning in DRMGs, our algorithm is able to learn an equilibrium with efficient sample complexity. As we shall discuss in the next section, our complexity bounds are near-optimal (except the term $\prod_{i\in\mathcal{M}} A_i$).

## 6 COMPARISON WITH PRIOR WORKS AND DISCUSSION

We then compare our results with prior works (the detailed Comparisons are shown in Table 1).

In (Li et al., 2025), $f(H, \rho) = (H\rho - 1 + (1 - \rho)^H)/\rho^2$.

A substantial body of research on DRMGs has focused on two primary settings: (i) generative model setting, where the agents can freely sample from all state-action pairs (Shi et al., 2025; 2024; Jiao & Li, 2024); (ii) offline setting, which relies on a comprehensive, pre-collected dataset (Blanchet et al., 2023; Li et al., 2025). As we discuss in Section 4, both of these avoid exploration and are therefore easier than the online regime we consider. Despite this added difficulty, our algorithm attains complexities comparable to those reported for the generative and offline settings.

For both uncertainty sets, our results match or improve upon previous results and the minimax lower bound in all parameters except for the action-product term, $\prod_i A_i$, under the generative model setting. In the offline setting, if the dataset is generated uniformly, the convergence coefficients $C_{u/p}^\star$ from (Li et al., 2025; Blanchet et al., 2023) introduce an additional $\prod_i A_i$ term into the sample complexity. Consequently, our results also match or surpass the offline complexity in all parameter dependence. This raises an important open question: **Can any DRMG learning algorithm overcome the curse of multi-agency and eliminate the dependence on $\prod_i A_i$ under general settings?**

While some works (Shi et al., 2025; Jiao & Li, 2024; Li et al., 2025; Ma et al., 2023) have achieved independence from $\prod_i A_i$, it remains unclear whether these improvements are applicable to general DRMGs. Specifically, the results in (Shi et al., 2025) and (Jiao & Li, 2024) are developed for special uncertainty sets with desirable properties. For instance, the fictitious TV uncertainty set in (Shi et al., 2025) allows the global transition kernel to be estimated from a single agent's local information; and robust RL under contamination models is known to be equivalent to a non-robust problem with a specific discount factor (Wang et al., 2023a). And the improvement in the offline setting is attributed to the benefits of the coverage coefficient.

Table 1: Comparison with prior results. $C^\star_{u/p}$ are coverage coefficients for offline learning.

| Setting & Algorithm | Uncertainty Set | Sample Complexity |
|---|---|---|
| **Generative** (Shi et al., 2024) | TV | $\tilde{\mathcal{O}}\left(\epsilon^{-2}H^3 S(\prod_{i\in\mathcal{M}} A_i)\min\left\{\rho_{\min}^{-1}, H\right\}\right)$ |
| **Generative** (Jiao & Li, 2024) | Contamination | $\tilde{\mathcal{O}}(\epsilon^{-2}H^3 S(\sum_{i\in\mathcal{M}} A_i)\min\left\{\rho_{\min}^{-1}, H\right\})$ |
| **Generative** (Shi et al., 2025) | TV (fictitious) | $\tilde{\mathcal{O}}\left(\epsilon^{-4}H^6 S(\sum_{i\in\mathcal{M}} A_i)\min\left\{\rho_{\min}^{-1}, H\right\}\right)$ |
| **Offline** (Blanchet et al., 2023) | KL | $\tilde{\mathcal{O}}\left(\epsilon^{-2}\rho_{\min}^{-2}C^\star_u H^4 \exp(H)S^2(\prod_{i\in\mathcal{M}} A_i)\right)$ |
| | TV | $\tilde{\mathcal{O}}\left(\epsilon^{-2}C^\star_u H^4 S^2(\prod_{i\in\mathcal{M}} A_i)\right)$ |
| **Offline** (Li et al., 2025) | TV | $\tilde{\mathcal{O}}\left(\epsilon^{-2}C^\star_p H^4 S(\sum_{i=1}^{m} A_i)\min\left\{f(H,\rho), H\right\}\right)$ |
| **Online** (Ma et al., 2023) | KL | $\tilde{\mathcal{O}}(\epsilon^{-2}H^5 S(\max_i\{A_i\})^2)$ (with an oracle) |
| **Online** (**Our work**) | TV | $\tilde{\mathcal{O}}\left(\epsilon^{-2}H^3 S(\prod_{i\in\mathcal{M}} A_i)\min\left\{\rho_{\min}^{-1}, H\right\}\right)$ |
| | KL | $\tilde{\mathcal{O}}\left(\epsilon^{-2}\rho_{\min}^{-2}(P^\star_{\min})^{-1}H^5\exp(2H^2)S\left(\prod_{i\in\mathcal{M}} A_i\right)\right)$ |
| **Generative** *Lower bound* (Shi et al., 2024) | TV | $\tilde{\Omega}\left(\epsilon^{-2}H^3 S(\max_{i\in\mathcal{M}} A_i)\min\left\{\rho_{\min}^{-1}, H\right\}\right)$ |

The only online method (which also breaks the curse of multi-agency) is presented in (Ma et al., 2023). However, their algorithm relies on additional assumptions about uncertainty sets and a powerful oracle. This oracle is required to provide an $\epsilon$-accurate estimation of the worst-case performance, $\sigma_{\mathcal{P}_i}[V]$ (see Theorem 12 of (Ma et al., 2023)), without any need for exploration. A central challenge in the analysis of robust learning algorithms is precisely quantifying this estimation error, as demonstrated in works like (Shi et al., 2023; Xu et al., 2023; Panaganti & Kalathil, 2022; Liu & Xu, 2024). By assuming the existence of such an oracle, they bypass this core challenge, which significantly reduces their sample complexity. Moreover, their results need additional assumptions on the radius $\rho$. For instance, it is assumed that $\rho \leq \frac{P^\star_{\min}}{H}$, whereas ours do not require any of them.

Therefore, the complexity reduction in these works is in fact a blessing of their specific uncertainty set structures, the properties of offline coverage coefficients, or the use of an impractical oracle. Given our lower bound derived in Section 4, we argue that the dependence on the joint action space may be inevitable in DRMGs. In the robust settings, agents need to estimate the entire nominal kernel so that they can learn the worst-case from the uncertainty set through distributionally robust optimization, which requires samples from all joint actions to estimate the whole transition kernel; whereas in the non-robust case, there is only one transition kernel and agents can use samples to directly estimate the performance under it, instead of estimating the whole transition model. We leave the exploration of this direction, including whether practical relaxations and techniques can avoid it, for future work.

## 7 CONCLUSION

In this paper, we introduced the Multiplayer Optimistic Robust Nash Value Iteration (MORNAVI) algorithm, pioneering the study of online learning in DRMGs. Our work provides the first provable guarantees for this challenging setting, demonstrating that MORNAVI achieves low regret and efficiently identifies optimal robust policies for TV-divergence and KL-divergence uncertainty sets. This research establishes a practical path toward developing truly robust multi-agent systems that learn directly from environmental interactions. Despite the inherent hardness of online DRMGs, our algorithm achieves complexity results comparable to the generative model and offline settings. This work also highlights a critical open question: whether online DRMG learning algorithms can overcome the curse of multi-agency and eliminate the dependence on the joint action space size. Future work will explore this fundamental challenge to advance the scalability of robust MARL.

ACKNOWLEDGMENT

This work was supported by DARPA under Agreement No. HR0011-24-9-0427 and NSF under Award CCF-2106339.

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

## A  USE OF LARGE LANGUAGE MODELS

We used ChatGPT only as a general-purpose assistant for language editing and typesetting. Its role was limited to (i) improving grammar, style, and readability, and (ii) LaTeX support—adjusting algorithm placement, tidying BibTeX entries and citation styles, and resolving compile issues (e.g., Type-3 font warnings and package conflicts). All ideas, derivations, and final claims were conceived, checked, and validated by the authors, who bear full responsibility for the paper's content.

## B  RELATED WORKS

In this section we discuss other related works.

**Single-Agent Robust RL.** Robust RL for single-agent settings has been extensively studied across a wide range of formulations. In particular, a substantial body of work has examined the generative-model setting (Clavier et al., 2023; Liu et al., 2022; Panaganti & Kalathil, 2022; Ramesh et al., 2024; Shi et al., 2023; Wang et al., 2023b;c;d; 2024b;e; 2023a; Wang & Zou, 2022; Xu et al., 2023; Yang et al., 2022; 2023; Roch et al., 2025b;a; Xu et al., 2025), where the agent is assumed to have access to a simulator. These studies develop distributionally robust RL algorithms under various uncertainty sets, including TV, KL, $\chi^2$, and Wasserstein divergences. Another, and arguably more challenging, line of research focuses on the offline setting (Blanchet et al., 2023; Ma et al., 2022; Panaganti et al., 2022; Shi & Chi, 2024; Zhang et al., 2024a; Liu & Xu, 2024; Wang et al., 2024d; Blanchet et al., 2023; Wang et al., 2024a). In this setting, the agent must learn exclusively from a fixed offline dataset, without the ability to collect additional online samples. Finally, we consider the online setting (Badrinath & Kalathil, 2021; Dong et al., 2024; Li et al., 2022; Liang et al., 2024; Wang & Zou, 2021), where the agent learns exclusively through direct interaction with the environment. Prior work spans model-based, model-free, and policy-gradient approaches, with some methods, such as the policy optimization algorithm of (Dong et al., 2024), achieving sublinear regret guarantees.

**Robust MARL.** Besides the distributionally robust Markov games we considered in our paper, there are also other works that investigate robustness in MARL for cooperative tasks, where all agents share a unified objective. (Bukharin et al., 2023) enhance robustness through adversarial regularization, perturbing the environment to encourage Lipschitz-continuous policies. (Lin et al., 2020) explore adversarial attacks on MARL agents as a means of improving resilience, while (Li et al., 2019) extend this approach to continuous action spaces by modifying the MADDPG algorithm (Lowe et al., 2017) to focus on worst-case actions—a narrower interpretation of worst-case optimization in robust RL. (Wang et al., 2022) studied robust MARL with network agents.

Another line of research focuses on the robustness in MARL under observation uncertainty, under the formulation of partially observable MDPs. The framework of observation-robust games is proposed in (He et al., 2023; Han et al., 2024). Observation-robust cooperative MARL is studied in (Zhou et al., 2024).

**Non-Robust Markov Games.** Markov games (MGs), or stochastic games, introduced by (Shapley, 1953), form the standard foundation for MARL, particularly in equilibrium learning. Comprehensive surveys such as (Busoniu et al., 2008; Oroojlooy & Hajinezhad, 2023; Zhang et al., 2021a) offer thorough coverage of the field's evolution. Early work in MARL focused on asymptotic convergence guarantees (Littman et al., 2001; Littman & Szepesvári, 1996), whereas recent research emphasizes finite-sample analyses to establish non-asymptotic guarantees, especially for learning Nash equilibria (NE)—a central solution concept. The existence of NE in general-sum MGs was shown by (Fink, 1964), and the algorithmic foundation was laid by the seminal work of (Littman, 1994). Classical algorithms such as Nash-Q (Hu & Wellman, 2003), FF-Q (Littman et al., 2001), and correlated-Q learning (Greenwald & Hall, 2003) were proposed to compute NE and its variants. However, computing NE in general-sum multi-player settings remains PPAD-complete (Daskalakis, 2013), and no polynomial-time algorithms exist for this case (Jin et al., 2023; Deng et al., 2023). In contrast, the two-player zero-sum setting admits tractable solutions, with the first polynomial-time algorithm developed by (Hansen et al., 2013). To address the computational intractability in general-sum MGs, attention has shifted to weaker notions like CE and CCE, with polynomial-time algorithms such as V-learning (Jin et al., 2022; Mao & Başar, 2023; Song et al., 2022) and Nash value iteration (Liu et al., 2021) enabling efficient computation. Furthermore, significant progress in finite-sample analysis—spanning both model-based and model-free algorithms—has been achieved

in the two-player zero-sum setting, as evidenced by (Bai & Jin, 2020; Xie et al., 2020; Cui et al., 2023; Chen et al., 2022; Liu et al., 2021; Feng et al., 2024; Li et al., 2024b), advancing the theoretical understanding of equilibrium learning in standard MARL without robustness considerations.

## C  DRMG WITH $f$-DIVERGENCE UNCERTAINTY SET

We review the formulation of DRMG with $f$-divergence uncertainty sets. This framework operates under the $\mathcal{S} \times \mathcal{A}$-rectangularity assumption, where the nominal transition probability $P^\star$ and the agent-specific radius $\rho_i$ for $i \in \mathcal{M}$ define the robust problem as per Definition 1.

**Lemma 7** (Strong duality for $f$-divergence). *Let $\mathcal{P}_f^{\rho_i}(s, \boldsymbol{a})$ be an $f$-divergence uncertainty set as defined in Definition 1. For any value function $V_i : \mathcal{S} \to \mathbb{R}_+$ and a nominal transition kernel $P^\star : \mathcal{S} \times \mathcal{A} \to \Delta(\mathcal{S})$, the worst-case expected value, $\sigma_{\mathcal{P}_f^{\rho_i}(s,\boldsymbol{a})}[V_i] := \inf_{P \in \mathcal{P}_f^{\rho_i}(s,\boldsymbol{a})} [\mathbb{P}V_i](s, \boldsymbol{a})$, admits a dual representation given by:*

$$\sigma_{\mathcal{P}_{i,h,f}^{\rho_i}(s,\mathbf{a})}[V] = \sup_{\lambda \geq 0, \, \eta \in \mathbb{R}} \left\{ -\lambda \sum_{s \in \mathcal{S}} P^\star(s) f^\star \left( \frac{\eta - V(s)}{\lambda} \right) - \lambda \rho_i + \eta \right\},$$

*where $f^\star$ is the convex conjugate of $f$.*

The detailed proof is given in Lemma B.1 of (Yang et al., 2022).

**Corollary 8** (Dual representation for TV and KL-divergence). *Under the assumption of $\mathcal{S} \times \mathcal{A}$-rectangularity, the dual representation from Lemma 7 simplifies to the following for two specific cases of $f$-divergence. For any value function $V : \mathcal{S} \to [0, H]$ and a nominal distribution $P_h^\star$ over the next states:*

***TV-Divergence.*** *For an uncertainty set defined by TV-divergence, where $f(t) = \frac{1}{2}\left|t - 1\right|$, the robust expectation $\sigma_{\mathcal{P}_{i,h,TV}^{\rho_i}(s,\mathbf{a})}[V_i]$ is expressed as:*

$$\sigma_{\mathcal{P}_{i,h,TV}^{\rho_i}(s,\mathbf{a})}[V_i] = \sup_{\eta \in [0,H]} \left\{ -\mathbb{E}_{P_h^\star(\cdot|s,\mathbf{a})}\left[ \max(0, \eta - V_i) \right] - \frac{\rho}{2} \max(0, \eta - \min_{s' \in \mathcal{S}} V_i(s')) + \eta \right\}. \tag{11}$$

***KL-Divergence.*** *For an uncertainty set defined by KL-divergence, with $f(t) = t \log(t)$, the robust expectation $\sigma_{\mathcal{P}_{i,h,KL}^{\rho_i}(s,\mathbf{a})}[V_i]$ is expressed as:*

$$\sigma_{\mathcal{P}_{i,h,KL}^{\rho_i}(s,\mathbf{a})}[V_i] = \sup_{\eta \in [\underline{\eta}, H/\rho_i]} \left\{ -\eta \log \left( \mathbb{E}_{P_h^\star(\cdot|s,\mathbf{a})} \left[ \exp \left\{ -\frac{V_i}{\eta} \right\} \right] \right) - \eta \rho_i \right\}. \tag{12}$$

ROBUST BELLMAN EQUATIONS.

Analogous to standard MGs, the following proposition provides the robust Bellman equation for DRMGs. In particular, the robust value functions $V_{i,h}^{\pi,\rho_i}(s)$ associated with any joint policy $\pi$ for all $(i, h, s) \in \mathcal{M} \times [H] \times \mathcal{S}$ obeys the following proposition given below:

**Proposition 9** (Robust Bellman Equation). *Under the $\mathcal{S} \times \mathcal{A}$-rectangularity assumption, for any nominal transition kernel $P^\star$ and joint policy $\pi$, the robust Bellman equation holds for any $(i, h, s, \boldsymbol{a})$:*

$$Q_{i,h}^{\pi,\rho_i}(s, \boldsymbol{a}) = r_{i,h}(s, \boldsymbol{a}) + \sigma_{\mathcal{P}_{i,h}^{\rho_i}(s,\boldsymbol{a})} \left[ V_{i,h+1}^{\pi,\rho_i} \right] \tag{13}$$

$$V_{i,h}^{\pi,\rho_i}(s) = \mathbb{E}_{\boldsymbol{a} \sim \pi_h(\cdot|s)} \left[ Q_{i,h}^{\pi,\rho_i}(s, \boldsymbol{a}) \right] \tag{14}$$

The detailed proof of Proposition 9 for finite-horizon RMDP is given in (Blanchet et al., 2023, Proposition 2.3). We emphasize that the robust Bellman equation in 14 is fundamentally grounded in the agent-wise $(s, \mathbf{a})$-rectangularity condition imposed on the uncertainty set. This condition decouples the dependencies of uncertainty across agents, state-action pairs, and time steps, thereby enabling the recursive structure of the Bellman equation.

# D  NUMERICAL EXPERIMENTS

In this section, we develop numerical experiments to validate our theoretical results. We highlight that numerical experiment for Markov games can be significantly challenging due to, e.g., the equilibrium identification challenge and computational barrier (Shoham & Leyton-Brown, 2008), hence we use some small-scale experiments to validate our results.

## D.1  FULLY COOPERATIVE DRMG

As the first step in numerical experiment, we design a 2-agent, 2-step fully cooperative DRMG (with identical rewards for both players), to illustrate the separation between our robust learning algorithm and the non-robust ones in standard Markov games.

The game is formally defined by the following components:

- **Agents** ($\mathcal{M}$)**:** The set of agents is $\mathcal{M} = \{1, 2\}$.
- **Horizon** ($H$)**:** The game has a finite horizon of $H = 2$.
- **State Space** ($\mathcal{S}$)**:** The state space is $\mathcal{S} = \{s_0, s_H, s_M, s_T\}$. The game always starts in state $s_0$ at $h = 1$. The states $s_H$ (High), $s_M$ (Medium), and $s_T$ (Trap) are the potential states for $h = 2$, and the episode terminates after this step.
- **Action Space** ($\mathcal{A}$)**:** Each agent has two actions, $\mathcal{A}_i = \{0, 1\}$ for $i \in \mathcal{M}$. The joint action space is $\mathcal{A} = \mathcal{A}_1 \times \mathcal{A}_2$, with joint actions $a = (a_1, a_2) \in \{(0, 0), (0, 1), (1, 0), (1, 1)\}$.

In our game, agents receive no reward at the first step: $r_{i,1}(s_0, a) = 0$ for all $i, a$. At step $h = 2$, the reward $r_{i,2}(s, a)$ for both agents is determined by the current state $s \in \{s_H, s_M, s_T\}$ and the joint action $a$. The rewards are defined as:

- **At $s_H$ (High):** This is the high-reward state, where $r_{i,2}(s_H, a) = 1$ for all $i, a$.
- **At $s_M$ (Medium):** This is a medium-reward state, where $r_{i,2}(s_M, a) = 0.6$ for all $i, a$.
- **At $s_T$ (Trap):** This is the low-reward, trap state, where $r_{i,2}(s_T, a) = 0$ for all $i, a$.

We then set the nominal transition kernel from $s_0$ at $h = 1$, $P_1^\star(\cdot|s_0, a)$. The probabilities are detailed as follows:

Table 2: Nominal transition probabilities $P_1^\star(\cdot|s_0, a)$ from the start state.

| Joint Action $a$ | $P_1^\star(s_H|s_0, a)$ | $P_1^\star(s_M|s_0, a)$ | $P_1^\star(s_T|s_0, a)$ | Description |
|---|---|---|---|---|
| $a = (1, 1)$ | 0.90 | 0.00 | 0.10 | Risky (high reward, trap support) |
| $a = (0, 0)$ | 0.60 | 0.40 | 0.00 | Safe (no trap support) |
| $a = (1, 0)$ | 0.50 | 0.25 | 0.25 | Mediocre |
| $a = (0, 1)$ | 0.50 | 0.25 | 0.25 | Mediocre |

It can be seen that, under the nominal kernel, the risky action is preferred as it has higher probability to transit to $s_H$. However, when there are model mismatch between the training and deploying environment, and under the risky action, the probability of transiting to the Trap state $s_T$ becomes higher, then the non-robust equilibrium becomes sub-optimal. On the other hand, our robust learning considers the worst-case, so it prefers to take the safe action. We will numerically show that our robust learning algorithm will learn a more robust policy that performs better under model uncertainties or the sim-to-real gap.

We aim to numerically verify two of our claims: (1). Our MORNAVI algorithm converges to the robust equilibria; And (2). The robust equilibria learned are more robust against model uncertainty compared to non-robust ones.

Specifically, we construct the uncertainty set as a KL-divergence ball centered at $P_h^\star$ as in Equation (2), which $\rho_i = \rho$. We then implement our algorithm (Algorithm 1) together with the non-robust Nash value iteration (Liu et al., 2021) as the baseline. Due to the hardness of computing Nash equilibria (which is PPAD-hard in the worst-case (Deng et al., 2023)), we compute the CCE for the games.

We develop two experiments as follows. Firstly, we run both algorithms (we set $\rho = 0.25$ in our algorithm) for 10 times, and plot the averaged robust value function of Player 1 against the total number of samples. We also plot the standard deviation to show statistical errors. Secondly, we test the learned equilibria from both algorithm under different uncertainty radii $\rho$. For different $\rho$, we compute the robust value function of Player 1 (since both players have identical performance) under the KL-ball, to showcase the robustness of our algorithm. The experiment results are shown in Figure 1.

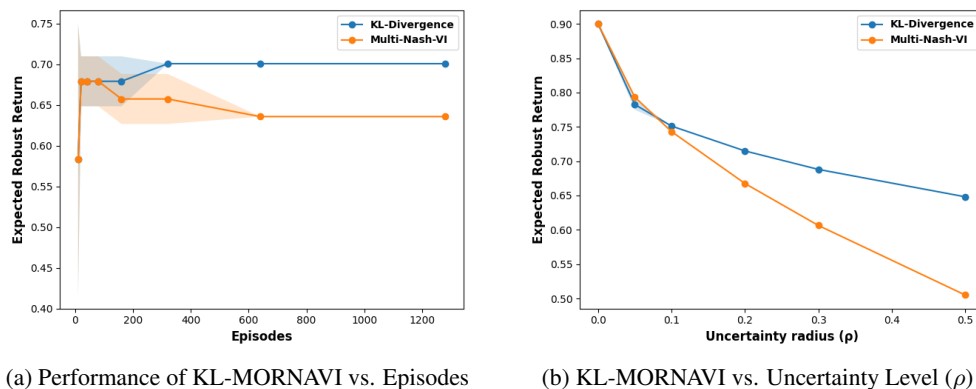

(a) Performance of KL-MORNAVI vs. Episodes        (b) KL-MORNAVI vs. Uncertainty Level ($\rho$)

Figure 1: $f$-MORNAVI v.s. Multi-Nash-VI under KL-Divergence

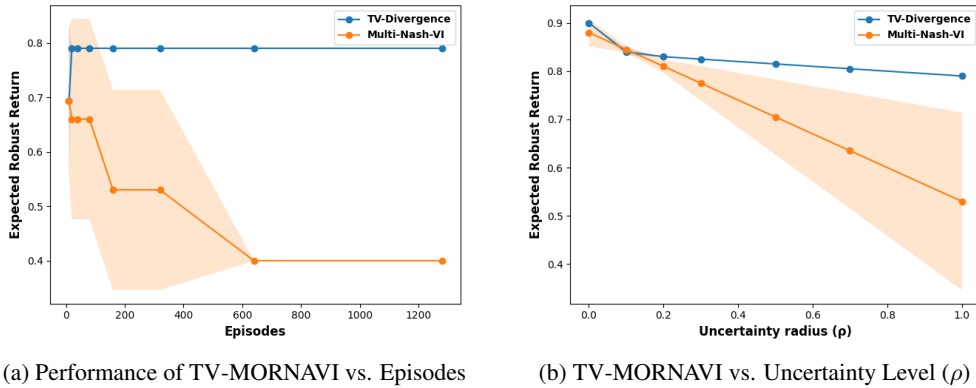

(a) Performance of TV-MORNAVI vs. Episodes        (b) TV-MORNAVI vs. Uncertainty Level ($\rho$)

Figure 2: $f$-MORNAVI v.s. Multi-Nash-VI under TV-Divergence

As the results shown, our algorithm converges to the robust equilibrium, validating the convergence of our theoretical results and convergence guarantees. Moreover, our robust equilibrium shows an enhanced robustness when model mismatch exists. Specifically, when $\rho \approx 0$ and there is no model mismatch, then the non-robust algorithm outperforms ours (as we are conservative and robust while non-robust is optimization for the nominal kernel); However, when the uncertainty radius increasing and model mismatch is introduced, performance of the non-robust equilibrium decreases significantly, whereas ours shows a more stable and robust performance. We trained the plots of uncertainty level radius for 400 episodes. Our results hence validate our theoretical results and claims.

Similarly, we develop experiments with TV-based uncertainty set, and plot results in Figure 2. As results shown, our algorithm converges to a robust equilibrium, which is more stable and robust against model uncertainties. Our results hence align with and validate our theoretical findings.

### D.2 GENERAL-SUM DRMG

We then slightly modify the fully cooperative DRMG considered, transferring it to a general-sum DRMG, to further validate our theoretical results.

We set the nominal kernel as follows. At step 1, the nominal transition $P_1^\star(\cdot \mid s_0, a)$ is

$$P_1^\star(\cdot \mid s_0, a) = \begin{cases} 0.82\,\delta_{s_H} + 0.18\,\delta_{s_T}, & a = (1,1) \text{ (risky)}, \\ 0.60\,\delta_{s_H} + 0.40\,\delta_{s_M}, & a = (0,0) \text{ (safe)}, \\ 0.48\,\delta_{s_H} + 0.22\,\delta_{s_M} + 0.30\,\delta_{s_T}, & a \in \{(1,0),(0,1)\} \text{ (off-diag)}. \end{cases}$$

At step 2 the kernel is absorbing: $P_2^\star(s' \mid s, a) = \mathbf{1}\{s' = s\}$ for $s \in \{s_H, s_M, s_T\}$.

The rewards are settled as follows. At the terminal step (step 2), each terminal state induces a $2\times2$ matrix game; let $R^{(1)}(s), R^{(2)}(s) \in \mathbb{R}^{2\times2}$ denote the row/column players' payoffs. We set

$$\textbf{High:} \quad R^{(1)}(s_H) = \begin{bmatrix} 0.55 & 0.90 \\ 1.00 & 1.20 \end{bmatrix}, \qquad R^{(2)}(s_H) = \begin{bmatrix} 0.70 & 0.85 \\ 0.90 & 1.00 \end{bmatrix},$$

$$\textbf{Medium:} \quad R^{(1)}(s_M) = \begin{bmatrix} 0.45 & 0.35 \\ 0.35 & 0.30 \end{bmatrix}, \qquad R^{(2)}(s_M) = \begin{bmatrix} 0.65 & 0.55 \\ 0.50 & 0.45 \end{bmatrix},$$

$$\textbf{Trap:} \quad R^{(1)}(s_T) = \mathbf{0}, \qquad R^{(2)}(s_T) = \mathbf{0}.$$

Both players then have different rewards and the game becomes a general-sum DRMG.

Similarly, we implement our algorithms with non-robust baseline under both KL and TV uncertainty sets. We plot the performance of both players (as they are different). Our observations from the experiment results remain the same. In Figure 3a and Figure 4a, our robust algorithm converges to a robust equilibrium (sample) efficiently. And in Figure 3b and Figure 4b, the robust equilibria learned by our algorithms maintain a more robust and stable performance under model mismatches, showcasing the enhanced robustness of our methods in MARL settings.

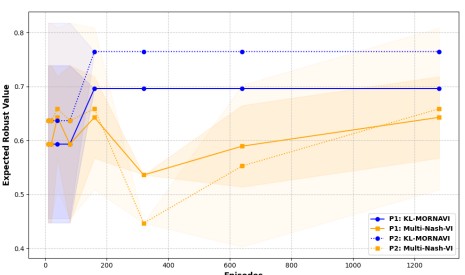

(a) Performance of KL-MORNAVI vs. Episodes

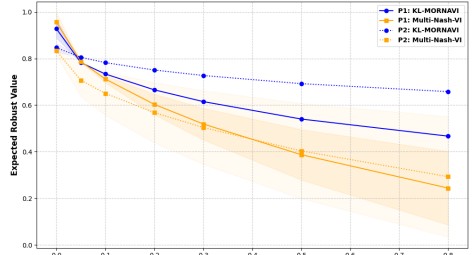

(b) KL-MORNAVI vs. Uncertainty Level ($\rho$)

Figure 3: $f$-MORNAVI v.s. Multi-Nash-VI under KL-Divergence

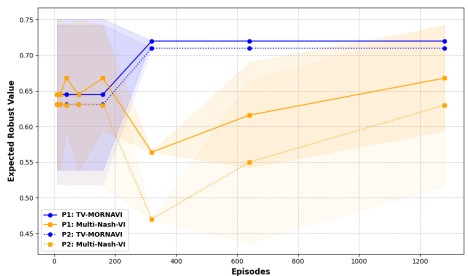

(a) Performance of TV-MORNAVI vs. Episodes

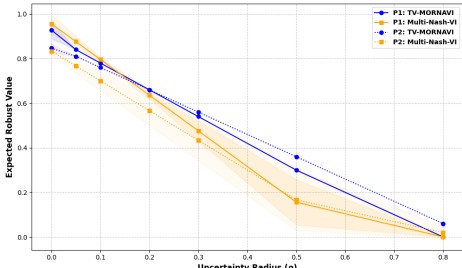

(b) TV-MORNAVI vs. Uncertainty Level ($\rho$)

Figure 4: $f$-MORNAVI v.s. Multi-Nash-VI under TV-Divergence

# E HARDNESS OF MULTI-AGENT ONLINE LEARNING

## E.1 HARDNESS WITH SUPPORT SHIFT

**Example 10** (The "Initial Shock" Game). *Consider a class of $N$-agent DRMGs, $\{M_{\boldsymbol{a}^*}\}_{\boldsymbol{a}^* \in \mathcal{A}}$, parameterized by a "secret escape route" $\boldsymbol{a}^* \in \mathcal{A}$.*

- ***Action Spaces:*** *$A_i = M$ for each agent. The joint action space has size $|\mathcal{A}| = \prod_{i \in [N]} A_i = M^N$.*

- ***States, Horizon, Rewards:*** *$\mathcal{S} = \{s_{good}, s_{bad}\}$, horizon $H$, initial state $s_1 = s_{good}$, and rewards are defined as*

$$r_i(s, \mathbf{a}) = \begin{cases} 1, & \text{if } s = s_{good} \text{ or if } (s = s_{bad} \text{ and } \boldsymbol{a} = \boldsymbol{a}^*) \\ 0, & \text{if } s = s_{bad} \text{ and } \boldsymbol{a} \neq \boldsymbol{a}^* \end{cases}.$$

- ***Dynamics:*** *The system dynamics create the trap.*

    - *From $\mathbf{s_{good}}$: Nominally, the system stays in $s_{good}$. An adversary can force a transition to $s_{bad}$ with probability $\rho$.*
    - *From $\mathbf{s_{bad}}$: This is the trap. The only way to escape is to play the secret joint action:*

$$\text{Next State} = \begin{cases} s_{good}, & \text{if } \boldsymbol{a} = \boldsymbol{a}^* \\ s_{bad}, & \text{if } \boldsymbol{a} \neq \boldsymbol{a}^* \end{cases}.$$

- ***Uncertainty Set:*** *The uncertainty is non-zero only at the first step.*

    - ***At*** *$h = 1$ **and** $s_1 = s_{good}$: The uncertainty set is a TV-ball with radius $\rho$.*
    - ***For all*** *$h > 1$ **or** $s \neq s_{good}$: There is no uncertainty ($\rho = 0$). The transition is the nominal one.*

**Theorem 11.** *For the "Initial Shock" DRMG, any decentralized online learning algorithm suffers the following best-response regret lower bound:*

$$\inf_{\mathcal{ALG}} \sup_{\boldsymbol{a}^* \in \mathcal{A}} \mathbb{E}[Regret_i(K)] \geq \Omega\left(\rho K \cdot \min\left\{H, \prod_{i \in [N]} A_i\right\}\right).$$

*Proof.* **Step 1: Decomposing the Per-Episode Regret.** The best-response regret for Agent 1 in an episode is $\text{Regret}_1^k = V_{1,1}^{\dagger, \pi^{-i}, \rho} - V_{1,1}^{\pi, \rho}$. We expand this using the robust Bellman equation at $s_1 = s_{good}$, where uncertainty exists.

$$\begin{aligned} \text{Regret}_1^k &= \left(1 + (1-\rho)V_{1,2}^{\dagger,\pi^{-i},\rho}(s_{good}) + \rho V_{1,2}^{\dagger,\pi^{-i},\rho}(s_{bad})\right) \\ &\quad - \left(1 + (1-\rho)V_{1,2}^{\pi,\rho}(s_{good}) + \rho V_{1,2}^{\pi,\rho}(s_{bad})\right) \\ &= (1-\rho)\left(V_{1,2}^{\dagger,\pi^{-i},\rho}(s_{good}) - V_{1,2}^{\pi,\rho}(s_{good})\right) + \rho\left(V_{1,2}^{\dagger,\pi^{-i},\rho}(s_{bad}) - V_{1,2}^{\pi,\rho}(s_{bad})\right). \end{aligned}$$

Since there is no uncertainty for $h > 1$, the transition from $s_{good}$ at $h = 2$ is deterministically to $s_{good}$ at $h = 3$. Thus, $V_{1,2}(s_{good})$ is a constant independent of the policy in the trap state, which means $V_{1,2}^{\dagger,\pi^{-i},\rho_i}(s_{good}) = V_{1,2}^{\pi}(s_{good})$. The first term is exactly zero, and thus we have that

$$\text{Regret}_1^k = \rho\left(V_{1,2}^{\dagger,\pi^{-i},\rho}(s_{bad}) - V_{1,2}^{\pi,\rho}(s_{bad})\right) = \rho \cdot \Delta V_2^{\rho}(s_{bad}). \tag{15}$$

**Step 2: Formalizing the Value Gap $\Delta V_2^{\rho}(s_{bad})$.** The value gap is the expected difference in total future rewards. This difference is precisely the expected number of steps wasted in the trap. Note that the value of state $s_{bad}$ at step $h$ under a policy $\pi'$ is the expected sum of future rewards. Let $\tau = \tau(\pi')$ be the random variable for the number of steps to escape (i.e., play $\boldsymbol{a}^*$), starting from step $h$. Let $C = H - h + 1$ be the number of steps remaining in the episode, then the total reward

collected from $h = 2$ is $V_{1,2}^{\pi',\rho}(s_{bad}) = \mathbb{E}[\mathbb{I}[\tau \le C] \cdot (C - \tau + 2)]$ as it will always receive $r = 1$ when at $s_{good}$.

Moreover, note that the total number of available rewards is $C$, and since $C = \min(\tau - 1, C) + \mathbb{I}[\tau \le C](C - \tau + 1)$, the value can therefore be expressed as $V_{1,2}^{\pi',\rho}(s_{bad}) = C - \mathbb{E}[\min(\tau - 1, C)]$.

Therefore, the value gap is the difference in the expected number of wasted steps:
$$\Delta V_2^{\rho}(s_{bad}) = (C - \mathbb{E}[\min(\tau^* - 1, C)]) - (C - \mathbb{E}[\min(\tau - 1, C)])$$
$$= \mathbb{E}[\min(\tau - 1, C)] - \mathbb{E}[\min(\tau^* - 1, C)].$$

where $\tau^*$ is the escape probability of $\pi^*$. Since the best-response policy $\pi_1^*$ plays $a_1^*$ deterministically, so its escape time $\tau^*$ depends only on the other agents' policies, $\pi_{-1}$. The algorithm's escape time $\tau$ depends on its full policy $\pi$.

**Step 3: Lower Bounding the Value Gap.** The best response for Agent 1 is to play $a_1^*$, so $\tau^*$ does not involve any search for Agent 1. In contrast,

However, the algorithm does not know $a_1^*$ and must search. We are interested in the worst-case regret over the choice of $a^*$. The expected wasted steps for the algorithm is $\mathbb{E}[\min(\tau - 1, C)]$. Let $p_1 = \mathrm{Pr}_{\pi_1}(a_1 = a_1^*)$ and $p_{-1} = \mathrm{Pr}_{\pi_{-1}}(a_{-1} = a_{-1}^*)$. The algorithm's one-step escape probability is $p_1 \cdot p_{-1}$. Its expected escape time is $\mathbb{E}[\tau] = 1/(p_1 \cdot p_{-1})$. The expected wasted steps is lower-bounded by:
$$\mathbb{E}[\min(\tau - 1, C)] \ge \Omega(\min(\mathbb{E}[\tau - 1], C)) = \Omega(\min(1/(p_1 \cdot p_{-1}), H - 1)),$$
where the inequality is due to Lemma 12.

In the worst case over the unknown $a^*$, the probabilities $p_1$ and $p_{-1}$ are minimized:
$$\inf_{a_1^*} p_1 \le 1/A_1 \quad \text{and} \quad \inf_{a_{-1}^*} p_{-1} \le 1 \Big/ \Big( \prod_{i=2}^{N} A_i \Big).$$

The best-response policy suffers much less waste. Thus, the value gap $\Delta V_2^{\rho}(s_{bad})$ is dominated by the algorithm's large number of wasted steps.
$$\sup_{a^*} \Delta V_2^{\rho}(s_{bad}) \ge \Omega \left( \min \left\{ 1 \Big/ \Big( (1/A_1) \cdot (1 \big/ \big( \prod_{i=2}^{N} A_i \big)) \Big), H \right\} \right) = \Omega \left( \min \left\{ \prod_{i=1}^{N} A_i, H \right\} \right).$$

**Step 4: Finalizing the Bound.** Substituting this back into the per-episode regret expression from Step 1:
$$\sup_{a^*} \mathbb{E}[\mathrm{Regret}_1^k] \ge \rho \cdot \Omega \left( \min \left\{ \prod_{i=1}^{N} A_i, H \right\} \right).$$

This per-episode regret is incurred because the information bottleneck prevents the algorithm from learning $a^*$. Summing over $K$ episodes gives the final total regret bound:
$$\inf_{\mathcal{ALG}} \sup_{a^*} \mathbb{E}[\mathrm{Regret}_1(K)] = \sum_{k=1}^{K} \sup_{a^*} \mathbb{E}[\mathrm{Regret}_1^k] \ge \Omega \left( \rho K \cdot \min \left\{ \prod_{i=1}^{N} A_i, H \right\} \right).$$

This completes the proof. $\qquad \square$

**Lemma 12.** *Let $\tau$ be the random variable for the escape time from the trap state, and let $C = H - 1$ be the number of steps remaining in the episode. The true expected number of wasted steps, $\mathbb{E}[\min(\tau - 1, C)]$, has the following asymptotic lower bound:*
$$\mathbb{E}[\min(\tau - 1, C)] \ge \Omega(\min(\mathbb{E}[\tau - 1], C)).$$

*Proof.* Note that $\tau$ follows a Geometric distribution $\tau \sim \mathrm{Geo}(p)$ and have the probability mass function $P(\tau = k) = (1 - p)^{k-1}p$ for $k \in \{1, 2, 3, \dots\}$. The random variable $\tau - 1$ represents the number of failures before the first success. Its expectation is $\mathbb{E}[\tau - 1] = \frac{1-p}{p}$.

We first derive an expression for $\mathbb{E}[\min(\tau - 1, C)]$. We use the tail sum formula for the expectation of a non-negative, integer-valued random variable $X$, which states $\mathbb{E}[X] = \sum_{k=0}^{\infty} P(X > k)$.

Let $X = \min(\tau - 1, C)$. The event $\{X > k\}$ is equivalent to the event $\{\tau - 1 > k \text{ and } C > k\}$.

- If $k \geq C$, then $P(X > k) = 0$.

- If $k < C$, then $P(X > k) = P(\tau - 1 > k)$.

The event $\{\tau - 1 > k\}$ means the first $k + 1$ trials resulted in failure, so its probability is $P(\tau > k + 1) = (1 - p)^{k+1}$.

The expectation is therefore the sum over the non-zero probabilities:

$$\mathbb{E}[\min(\tau - 1, C)] = \sum_{k=0}^{\infty} P(\min(\tau - 1, C) > k) = \sum_{k=0}^{C-1} P(\tau - 1 > k) = \sum_{k=0}^{C-1} (1 - p)^{k+1}.$$

Letting $q = 1 - p$, this is a finite geometric series:

$$\sum_{j=1}^{C} q^j = q \frac{1 - q^C}{1 - q} = \frac{q(1 - q^C)}{p}.$$

Substituting $q = 1 - p$ back, we express the expectation in terms of $\mathbb{E}[\tau - 1]$:

$$\mathbb{E}[\min(\tau - 1, C)] = \frac{1 - p}{p}(1 - (1 - p)^C) = \mathbb{E}[\tau - 1](1 - (1 - p)^C).$$

Let $\mu = \mathbb{E}[\tau - 1] = \frac{1-p}{p}$. We want to show that there exists a universal constant $k > 0$ such that:

$$\mu(1 - (1 - p)^C) \geq k \cdot \min(\mu, C).$$

We proceed with a case analysis based on the relationship between $\mu$ and $C$.

**Case 1:** $\mu \leq C$: In this case, $\min(\mu, C) = \mu$. We need to show that $\mu(1 - (1 - p)^C) \geq k \cdot \mu$, which simplifies to proving that $1 - (1 - p)^C \geq k$.

The condition $\mu \leq C$ implies a lower bound on $p$:

$$\frac{1 - p}{p} \leq C \implies 1 - p \leq Cp \implies 1 \leq (C + 1)p \implies p \geq \frac{1}{C + 1}.$$

Using the standard inequality $1 - x \leq e^{-x}$, we have $(1 - p)^C \leq e^{-pC}$. Thus,

$$1 - (1 - p)^C \geq 1 - e^{-pC}.$$

Since $p \geq \frac{1}{C+1}$, we have $pC \geq \frac{C}{C+1}$. As the function $f(x) = 1 - e^{-x}$ is increasing for $x > 0$,

$$1 - e^{-pC} \geq 1 - e^{-C/(C+1)}.$$

The function $g(C) = \frac{C}{C+1}$ is increasing for $C \geq 1$, with a minimum value of $g(1) = 1/2$. Therefore, for any integer $C \geq 1$,

$$1 - (1 - p)^C \geq 1 - e^{-1/2}.$$

Thus, the inequality holds in this case with the constant $k_1 = 1 - e^{-1/2} \approx 0.393$.

**Case 2:** $\mu > C$: In this case, $\min(\mu, C) = C$. We need to show that $\mu(1 - (1 - p)^C) \geq kC$.

The condition $\mu > C$ implies an upper bound on $p$:

$$\frac{1 - p}{p} > C \implies 1 - p > Cp \implies 1 > (C + 1)p \implies p < \frac{1}{C + 1}.$$

From our calculation of the expectation, we have a sum of $C$ positive, decreasing terms:

$$\mathbb{E}[\min(\tau - 1, C)] = \sum_{k=0}^{C-1} (1 - p)^{k+1}.$$

This sum is greater than $C$ times its smallest term, which is $(1 - p)^C$:

$$\mathbb{E}[\min(\tau - 1, C)] > C(1 - p)^C.$$

From the condition $p < \frac{1}{C+1}$, it follows that $1 - p > 1 - \frac{1}{C+1} = \frac{C}{C+1}$. Therefore,

$$\mathbb{E}[\min(\tau - 1, C)] > C\left(\frac{C}{C+1}\right)^C = C\left(1 - \frac{1}{C+1}\right)^C.$$

The sequence $a_C = \left(1 - \frac{1}{C+1}\right)^C$ is decreasing for $C \geq 1$, and its limit as $C \to \infty$ is $1/e$. Hence, for all $C \geq 1$, the sequence is bounded below by its limit:

$$\left(1 - \frac{1}{C+1}\right)^C \geq \lim_{n \to \infty}\left(1 - \frac{1}{n+1}\right)^n = \frac{1}{e}.$$

This gives the lower bound:

$$\mathbb{E}[\min(\tau - 1, C)] > C \cdot \frac{1}{e}.$$

So, the inequality holds in this case with the constant $k_2 = 1/e \approx 0.368$. By combining the two cases, the inequality is shown to hold for a universal constant $k = \min(k_1, k_2) = \min(1 - e^{-1/2}, 1/e) = 1/e$.

Therefore, for all $p \in (0, 1)$ and integers $C \geq 1$, we have established that:

$$\mathbb{E}[\min(\tau - 1, C)] \geq \frac{1}{e}\min(\mathbb{E}[\tau - 1], C) = \Omega(\min(\mathbb{E}[\tau - 1], C)),$$

which hence completes the proof. $\qquad\square$

### E.2 HARDNESS WITHOUT SUPPORT SHIFT

**Example 13** (The "Robust Corrupted Bandit" Game). *Consider a class of $N$-agent DRMGs, $\{M_\theta\}_{\theta \in \mathcal{A}}$, where each game is parameterized by a secret "best" joint action $\theta \in \mathcal{A}$.*

- **States and Horizon:** *A single state, $s$, and horizon $H = 1$. This reduces the problem to a one-shot game, equivalent to a multi-armed bandit setting where each episode corresponds to a single step or arm pull.*

- **Action Spaces:** *The joint action space $\mathcal{A}$ is the set of arms, with size $|\mathcal{A}| = \prod_{i=1}^N A_i$.*

- **Reward Function ($R \in \{0, 1\}$):** *The rewards are stochastic. Let $\epsilon \in (0, 1/2)$ be a small constant. The nominal model $M_\theta$ defines the following Bernoulli reward distributions for any agent $i$:*
$$\mathbb{E}[R_i(s, \boldsymbol{a})|M_\theta] = \begin{cases} 1/2 + \epsilon, & \text{if } \boldsymbol{a} = \theta \\ 1/2, & \text{if } \boldsymbol{a} \neq \theta. \end{cases}$$

- **KL-Divergence Uncertainty Set:** *The true reward distribution for an action $\boldsymbol{a}$, denoted $\tilde{P}(\cdot|\boldsymbol{a})$, can be any distribution that is close to the nominal one $P^*(\cdot|\boldsymbol{a})$:*
$$\mathcal{P}_{i,h,KL}^{\rho_i}(., \boldsymbol{a}) = \left\{ \tilde{P} : \mathrm{KL}(\tilde{P}(\cdot|\boldsymbol{a})\|P_{M_\theta}(\cdot|\boldsymbol{a})) \leq \rho_i, \forall \boldsymbol{a} \in \mathcal{A} \right\}.$$

*This uncertainty set does not have a support shift.*

The learning problem is to identify the best arm $\theta$ by observing noisy rewards that are actively corrupted by an adversary.

**Theorem 1** (Lower Bound for Robust Learning without Support Shift). *For the "Robust Corrupted Bandit" game, any learning algorithm suffers the following cumulative regret lower bound over $K$ episodes (steps):*

$$\inf_{\mathcal{ALG}} \sup_{\theta \in \mathcal{A}} \mathbb{E}[\mathrm{Regret}_i(K)] \geq \Omega\left(\sqrt{\prod_{i=1}^N A_i K}\right).$$

*Proof.* The proof proceeds by a formal reduction to the classic multi-armed bandit (MAB) problem.

Let $\mathcal{M}_\rho = \{M_{\theta,\rho}\}_{\theta \in \mathcal{A}}$ denote the class of robust game instances from our example, with uncertainty radius $\rho > 0$. Let $\mathcal{M}_0 = \{M_{\theta,0}\}_{\theta \in \mathcal{A}}$ be the corresponding class of non-robust instances, where the uncertainty radius is zero and the rewards are always drawn from the nominal distributions.

Note that since the horizon $H = 1$, the robust problem reduces to a non-robust one, and thus the worst-case regret over the robust class $\mathcal{M}_\rho$ must be at least as high as the worst-case regret over the non-robust class $\mathcal{M}_0$:

$$\mathbb{E}[\text{Regret}(K; M_{\theta,\rho})] \geq \mathbb{E}[\text{Regret}(K; M_{\theta,0})].$$

And thus

$$\inf_{\mathcal{ALG}} \sup_{\theta \in \mathcal{A}} \mathbb{E}[\text{Regret}(K; M_{\theta,\rho})] \geq \inf_{\mathcal{ALG}} \sup_{\theta \in \mathcal{A}} \mathbb{E}[\text{Regret}(K; M_{\theta,0})]. \tag{16}$$

Therefore, we can establish a lower bound for the robust problem by proving one for the simpler non-robust case.

The non-robust problem instance, $\mathcal{M}_0$, is a classic stochastic multi-armed bandit problem with $M = |\mathcal{A}|$ arms. A foundational result in this area provides a strong lower bound on regret.

Note that following standard lemma:

**Lemma 14.** *(Auer et al., 2002) For any integer $M \geq 2$ and $K > M$, and for any bandit algorithm, there exists a multi-armed bandit problem instance with $M$ arms whose reward distributions are supported on $[0, 1]$, such that the expected cumulative regret after $K$ steps is lower-bounded by:*

$$\mathbb{E}[Regret(K)] \geq \Omega(\sqrt{MK}).$$

We apply the lemma to our non-robust problem instance $\mathcal{M}_0$.

- The number of arms, $M$, is the size of the joint action space, $|\mathcal{A}|$.

- The number of steps is $K$.

- The reward distributions (Bernoulli) are supported on $[0, 1]$.

The conditions of the lemma are met. Therefore, for the class of problems $\mathcal{M}_0$, the worst-case regret is lower-bounded:

$$\inf_{\mathcal{ALG}} \sup_{\theta \in \mathcal{A}} \mathbb{E}[\text{Regret}(K; M_{\theta,0})] \geq \Omega\left(\sqrt{\prod_{i=1}^{N} A_i K}\right). \tag{17}$$

Combining the regret dominance principle from eq. 16 with the specific lower bound from eq. 17, we arrive at the final result for our robust problem:

$$\inf_{\mathcal{ALG}} \sup_{\theta \in \mathcal{A}} \mathbb{E}[\text{Regret}_i(K; M_{\theta,\rho})] \geq \Omega\left(\sqrt{\prod_{i=1}^{N} A_i K}\right). \tag{18}$$

This completes the formal proof by reduction.

$\square$

## F   PROOF OF REGRET BOUND OF TV-MORNAVI

In this section, we prove our regret bound for TV-DRMG. Before presenting all the proofs, we first denote $\pi^\dagger$ as the joint robust best responses over the agents, and is gven by

$$\pi^\dagger = \pi_1^{\dagger,\rho_1}(\pi_{-1}) \times \cdots \times \pi_m^{\dagger,\rho_m}(\pi_{-m}). \tag{19}$$

We will use the notation of $\pi^\dagger$ later on our proof-lines. In addition, we leverage Assumption 3, which generalizes to the case where the minimal value vanishes, i.e., $\min_{s \in \mathcal{S}} V(s) = 0$, to address the support shift or extrapolation challenge arising in interactive data collection, as discussed in Remark

B.3 of (Lu et al., 2024). Consequently, this allows us to eliminate the $\min_{s \in \mathcal{S}} V(s)$ term in the dual formulation of the TV-DRMG optimization problem, as shown in 11.

We now recall the bonus term used in TV-MORNAVI for agent $i$ in episode $k$ at step $h$, as follows:

$$\beta_{i,h}^k(s, \mathbf{a}) = \sqrt{\frac{c_1 \iota \mathrm{Var}_{\widehat{P}_h^k(\cdot | s, \mathbf{a})} \left[ \left( \frac{\overline{V}_{i,h+1}^{k,\rho_i} + \underline{V}_{i,h+1}^{k,\rho_i}}{2} \right) \right]}{\{N_h^k(s, \mathbf{a}) \vee 1\}}} + \frac{2\mathbb{E}_{\widehat{P}_h^k(\cdot | s, \mathbf{a})} \left[ \overline{V}_{i,h+1}^{k,\rho_i} - \underline{V}_{i,h+1}^{k,\rho_i} \right]}{H}$$
$$+ \frac{c_2 H^2 S \iota}{\sqrt{\{N_h^k(s, \mathbf{a}) \vee 1\}}} + \frac{1}{\sqrt{K}}, \tag{20}$$

where $\iota = \log \left( S^2 (\prod_{i=1}^m A_i) H^2 K^{3/2} / \delta \right)$ and $c_1, c_2$ are absolute constants.

We begin by defining the high-probability event $\mathcal{E}_{\mathrm{TV}}$, stated in the next lemma. Our proof outline is inspired by (Lu et al., 2024) and (Ghosh et al., 2025).

**Lemma 15** (Uniform Concentration Bound of event $\mathcal{E}_{\mathrm{TV}}$). *Let $\mathcal{E}_{\mathrm{TV}}$ be the event in which, for all $(s, \mathbf{a}, s', h, k) \in \mathcal{S} \times \mathcal{A} \times \mathcal{S} \times [H] \times [K]$, and for all $\eta$ in a $1/(S\sqrt{K})$-cover of $[0, H]$, and is defined as*

$$\mathcal{E}_{TV} := \Bigg\{ \left| \left[ \mathbb{E}_{\widehat{P}_h^k(\cdot | s, \mathbf{a})} - \mathbb{E}_{P_h^\star(\cdot | s, \mathbf{a})} \right] \left( \eta - V_{i,h+1}^{\dagger, \pi_{-i}^k, \rho_i} \right)_+ \right| \leq \sqrt{\frac{c_1 \iota \mathrm{Var}_{\widehat{P}_h^k} \left( \eta - V_{i,h+1}^{\dagger, \pi_{-i}^k, \rho_i} \right)_+}{N_h^k(s, \mathbf{a}) \vee 1}}$$
$$+ \frac{c_2 H \iota}{\{N_h^k(s, \mathbf{a}) \vee 1\}},$$

$$\left| \widehat{P}_h^k(s' \mid s, \mathbf{a}) - P_h^\star(s' \mid s, \mathbf{a}) \right| \leq \sqrt{\frac{c_1 \min \left\{ P_h^\star(s' \mid s, \mathbf{a}), \widehat{P}_h^k(s' \mid s, \mathbf{a}) \right\} \cdot \iota}{\{N_h^k(s, \mathbf{a}) \vee 1\}}}$$
$$+ \frac{c_2 \iota}{\{N_h^k(s, \mathbf{a}) \vee 1\}},$$

$$\forall (s, \mathbf{a}, s', h, k) \in \mathcal{M} \times \mathcal{S} \times \mathcal{A} \times \mathcal{S} \times [H] \times [K], \forall \eta \in \mathcal{N}_{1/(S\sqrt{K})}([0, H]) \Bigg\}, \tag{21}$$

*where $\iota = \log \left( S^3 (\prod_{i=1}^m A_i) H^2 K^{3/2} / \delta \right)$, $c_1, c_2 > 0$ are two absolute constants, $\mathcal{N}_{1/(S\sqrt{K})}([0, H])$ denotes an $1/S\sqrt{K}$-cover of the interval $[0, H]$.*

*Then, this event $\mathcal{E}_{TV}$ occurs with high probability, i.e., $\mathrm{Pr}(\mathcal{E}_{TV}) \geq 1 - \delta$.*

*Proof.* This proof builds upon standard techniques by applying classical concentration inequalities and a union bound. To simplify our analysis, we first consider a fixed state-action-time tuple $(s, \mathbf{a}, h)$ within a given episode $k$. We can then construct an equivalent stochastic process:

(i) Before the agents' interaction, the environment draws a sequence of next states $\{s^{(1)}, s^{(2)}, \ldots, s^{(k-1)}\}$ independently from the nominal distribution $P_h^\star(\cdot | s, \mathbf{a})$, where $s^{(i)} \in \mathcal{S}$ represents the state sampled in episode $i$.

(ii) When the agents visit the $(s, \mathbf{a})$ tuple at time step $h$ for the $i$-th time, the environment causes a transition to the pre-sampled next state $s^{(i)}$.

The randomness of this constructed process is identical to that of our original, interactive learning environment. Consequently, the probability of any event is the same in both contexts. This allows us to prove the required concentration inequalities within this more tractable, simplified setting.

Leveraging this fact, we directly apply Lemma 40, which presents a variant of Bernstein's inequality and its empirical counterpart from (Maurer & Pontil, 2009). To establish a uniform bound, we apply a union bound across all tuples $(h, s, \mathbf{a}, s', k, \eta) \in [H] \times \mathcal{S} \times \mathcal{A} \times \mathcal{S} \times [K] \times \mathcal{N}_{1/(S\sqrt{K})}([0, H])$. The size of this $\epsilon$-cover, $\mathcal{N}_{1/(S\sqrt{K})}([0, H])$, is on the order of $\mathcal{O}(SH\sqrt{K})$. □

### F.1 PROOF OF THEOREM 4 (TV-DRMG SETTING)

*Proof.* By leveraging Lemma 20, we can establish an upper bound on the regret by considering the difference between the optimistic and pessimistic value functions:

$$\text{Regret}_{\text{NASH}}(K) = \sum_{k=1}^{K} \max_{i \in \mathcal{M}} \left( V_{i,1}^{\dagger, \pi_{-i}^k, \rho_i} - V_{i,1}^{\pi^k, \rho_i} \right)(s_1^k) \leq \sum_{k=1}^{K} \max_{i \in \mathcal{M}} \left( \overline{V}_{i,1}^{k,\rho_i} - \underline{V}_{i,1}^{k,\rho_i} \right)(s_1^k). \quad (22)$$

For the TV-divergence uncertainty set, we begin by analyzing the difference between the upper and lower Q-values. Given our definitions for $\overline{Q}_h^k$, $\underline{Q}_{i,h}^{k,\rho_i}$, $\overline{V}_{i,h}^{k,\rho_i}$, and $\underline{V}_{i,h}^{k,\rho_i}$ (from eq. 5- 8), along with the bonus term $\beta_{i,h}^k(s, \boldsymbol{a})$ defined in eq. 20, we can establish a bound on this difference for any $(h,k) \in [H] \times [K]$ and $(s, \boldsymbol{a}) \in \mathcal{S} \times \mathcal{A}$:

$$\overline{Q}_h^k(s, \mathbf{a}) - \underline{Q}_h^k(s, \mathbf{a}) \leq \sigma_{\widehat{\mathcal{P}_{i,h}^{\rho_i}}(s,\boldsymbol{a})} \left[ \overline{V}_{i,h+1}^{k,\rho_i} \right] - \sigma_{\widehat{\mathcal{P}_{i,h}^{\rho_i}}(s,\boldsymbol{a})} \left[ \underline{V}_{h+1}^{k,\rho_i} \right] + 2\beta_{i,h}^k(s, \mathbf{a}). \quad (23)$$

We introduce two key terms, $A$ and $B$, to simplify this expression:

$$A := \sigma_{\widehat{\mathcal{P}_{i,h}^{\rho_i}}(s,\boldsymbol{a})} \left[ \overline{V}_{i,h+1}^{k,\rho_i} \right] - \sigma_{\mathcal{P}_{i,h}^{\rho_i}(s,\boldsymbol{a})} \left[ \overline{V}_{i,h+1}^{k,\rho_i} \right] + \sigma_{\mathcal{P}_{i,h}^{\rho_i}(s,\mathbf{a})} \left[ \underline{V}_{i,h+1}^{k,\rho_i} \right] - \sigma_{\widehat{\mathcal{P}_{i,h}^{\rho_i}}(s,\boldsymbol{a})} \left[ \underline{V}_{i,h+1}^{k,\rho_i} \right]. \quad (24)$$

$$B := \sigma_{\mathcal{P}_{i,h}^{\rho_i}(s,\mathbf{a})} \left[ \overline{V}_{i,h+1}^{k,\rho_i} \right] - \sigma_{\mathcal{P}_{i,h}^{\rho_i}(s,\mathbf{a})} \left[ \underline{V}_{i,h+1}^{k,\rho_i} \right]. \quad (25)$$

By substituting these definitions into eq. 23, we obtain a new bound:

$$\overline{Q}_{i,h}^{k,\rho_i}(s, \mathbf{a}) - \underline{Q}_{i,h}^{k,\rho_i}(s, \mathbf{a}) \leq A + B + 2\beta_{i,h}^k(s, \mathbf{a}). \quad (26)$$

We then proceed to bound each of these terms. A concentration bound argument tailored for TV robust expectations in Lemma 18 shows that $A \leq 2\beta_{i,h}^k(s, \mathbf{a})$. For term $B$, we use the dual representation of $\sigma_{\mathcal{P}_{i,h}^{\rho_i}(s,\mathbf{a})}[V]$ from eq. 11 and Assumption 3 to first establish that $B \leq \sup_{\eta \in [0,H]} \{ \mathbb{E}_{P_h^\star(\cdot|s,\mathbf{a})}[\eta - \overline{V}_{i,h+1}^{k,\rho_i}]_+ - \mathbb{E}_{P_h^\star(\cdot|s,\mathbf{a})}[\eta - \underline{V}_{i,h+1}^{k,\rho_i}]_+ \}$. Since $\overline{V}_{i,h+1}^{k,\rho_i} \geq \underline{V}_{i,h+1}^{k,\rho_i}$ (by Lemma 20), we can simplify this further to $B \leq \mathbb{E}_{P_h^\star(\cdot|s,\mathbf{a})}[\overline{V}_{i,h+1}^{k,\rho_i} - \underline{V}_{i,h+1}^{k,\rho_i}]$.

By substituting the bounds for $A$ and $B$ back into eq. 26, we arrive at the following inequality:

$$\overline{Q}_{i,h}^{k,\rho_i}(s, \mathbf{a}) - \underline{Q}_{i,h}^{k,\rho_i}(s, \mathbf{a}) \leq \mathbb{E}_{P_h^\star(\cdot|s,\mathbf{a})}[\overline{V}_{i,h+1}^{k,\rho_i} - \underline{V}_{i,h+1}^{k,\rho_i}] + 4\beta_{i,h}^k(s, \mathbf{a}). \quad (27)$$

Using Lemma 19 to upper bound the bonus term, and rearranging the terms, we obtain:

$$\overline{Q}_{i,h}^{k,\rho_i}(s, \mathbf{a}) - \underline{Q}_{i,h}^{k,\rho_i}(s, \mathbf{a}) \leq \left( 1 + \frac{20}{H} \right) \mathbb{E}_{P_h^\star(\cdot|s,\mathbf{a})}[\overline{V}_{i,h+1}^{k,\rho_i} - \underline{V}_{i,h+1}^{k,\rho_i}]$$

$$+ 4\sqrt{\frac{c_1 \iota \text{Var}_{P_h^\star(\cdot|s,\mathbf{a})}\left[ V_{i,h+1}^{\pi^k,\rho_i} \right]}{\{N_h^k(s, \mathbf{a}) \vee 1\}}} + \frac{4c_2 H^2 S\iota}{\{N_h^k(s, \mathbf{a}) \vee 1\}} + \sqrt{\frac{4}{K}}, \quad (28)$$

where $c_1, c_2 > 0$ are absolute constants. From the definitions in eq. 8, the difference in V-functions is given by:

$$\overline{V}_{i,h}^{k,\rho_i}(s) - \underline{V}_{i,h}^{k,\rho_i}(s) = \mathbb{E}_{\mathbf{a} \sim \pi^k(\cdot|s)} \left[ \overline{Q}_{i,h}^{k,\rho_i}(s, \mathbf{a}) - \underline{Q}_{i,h}^{k,\rho_i}(s, \mathbf{a}) \right]. \quad (29)$$

Now, let's define a new recursive value function $\widetilde{V}_h^{k,\rho_{\min}}$ and a corresponding Q-function $\widetilde{Q}_h^{k,\rho_{\min}}$ with $\widetilde{V}_{H+1}^{k,\rho_{\min}} = 0$, where $\rho_{\min} = \min_{i \in \mathcal{M}} \rho_i$. Furthermore, we consider that there exists a $\rho^\star$ such that $\rho^\star = \operatorname*{argmax}_{\rho_i, \forall i \in \mathcal{M}, h \in [H]} \text{Var}_{P_h^\star(\cdot|s,\mathbf{a})}\left[ V_{i,h+1}^{\pi^k,\rho_i} \right]$. Using these facts we now define

$$\widetilde{Q}_h^{k,\rho_{\min}}(s, \mathbf{a}) = \left( 1 + \frac{20}{H} \right) \mathbb{E}_{P_h^\star(\cdot|s,\mathbf{a})} \left[ \widetilde{V}_{h+1}^{k,\rho_{\min}} \right] + 4\sqrt{\frac{c_1 \iota \text{Var}_{P_h^\star(\cdot|s,\mathbf{a})}\left[ V_{h+1}^{\pi^k,\rho^\star} \right]}{\{N_h^k(s, \mathbf{a}) \vee 1\}}}$$

$$+ \frac{4c_2 H^2 S\iota}{\{N_h^k(s, \mathbf{a}) \vee 1\}} + \sqrt{\frac{4}{K}}, \quad (30)$$

$$\widetilde{V}_h^{k,\rho_{\min}}(s) = \mathbb{E}_{\mathbf{a} \sim \pi^k(\cdot|s)} \left[ \widetilde{Q}_h^{k,\rho_{\min}}(s, \mathbf{a}) \right]. \quad (31)$$

It is a well-known property of robust value functions under TV-divergence that they become more conservative as the uncertainty radius $\rho_i$ decreases (e.g., (Iyengar, 2005; Nilim & El Ghaoui, 2005)). Given that $\rho_{\min} \le \rho_i$ for all agents $i \in \mathcal{M}$, it follows that for every next state $s' \in \mathcal{S}$:

$$V_{i,h+1}^{\pi^k,\rho_i}(s') \le V_{h+1}^{\pi^k,\rho_{\min}}(s') \quad \forall i \in \mathcal{M} \text{ and } s \in \mathcal{S}.$$

We can inductively prove that for any $(i, h, s, \mathbf{a}) \in \mathcal{M} \times [H] \times \mathcal{S} \times \mathcal{A}$:

$$\max_{i \in \mathcal{M}} \left( \overline{Q}_{i,h}^{k,\rho_i}(s, \mathbf{a}) - \underline{Q}_{i,h}^{k,\rho_i}(s, \mathbf{a}) \right) \le \widetilde{Q}_h^{k,\rho_{\min}}(s, a), \tag{32}$$

$$\max_{i \in \mathcal{M}} \left( \overline{V}_{i,h}^{k,\rho_i}(s) - \underline{V}_{i,h}^{k,\rho_i}(s) \right) \le \widetilde{V}_h^{k,\rho_{\min}}(s). \tag{33}$$

Inductive Proof: Let eq. 33 hold for $h + 1$. So, we will show that eq. (32) holds for h. For any $h$, by eq. 28 we have

$$\left( \overline{Q}_{i,h}^{k,\rho_i}(s, \mathbf{a}) - \underline{Q}_{i,h}^{k,\rho_i}(s, \mathbf{a}) \right)$$

$$\le \left( 1 + \frac{20}{H} \right) \mathbb{E}_{P_h^\star(\cdot|s,\mathbf{a})} \left[ \overline{V}_{i,h+1}^{k,\rho_i} - \underline{V}_{i,h+1}^{k,\rho_i} \right] + 4 \sqrt{\frac{c_1 \iota \mathrm{Var}_{P_h^\star(\cdot|s,\mathbf{a})} \left[ V_{i,h+1}^{\pi^k,\rho_i} \right]}{N_h^k(s,\mathbf{a}) \vee 1}} + \frac{4c_2 H^2 S \iota}{N_h^k(s,\mathbf{a}) \vee 1} + \sqrt{\frac{4}{K}},$$

$$\overset{(a)}{\le} \left( 1 + \frac{20}{H} \right) \mathbb{E}_{P_h^\star(\cdot|s,\mathbf{a})}[\widetilde{V}_{h+1}^{k,\rho_{\min}}] + 4 \sqrt{\frac{c_1 \iota \mathrm{Var}_{P_h^\star(\cdot|s,\mathbf{a})} \left[ V_{i,h+1}^{\pi^k,\rho^\star} \right]}{\{N_h^k(s,\mathbf{a}) \vee 1\}}} + \frac{4c_2 H^2 S \iota}{\{N_h^k(s,\mathbf{a}) \vee 1\}} + \sqrt{\frac{4}{K}},$$

$$\overset{(b)}{=} \widetilde{Q}_h^{k,\rho_{\min}}(s, \mathbf{a})$$

$$\overline{V}_{i,h}^{k,\rho_i}(s) - \underline{V}_{i,h}^{k,\rho_i}(s) \overset{(c)}{=} \mathbb{E}_{\mathbf{a} \sim \pi^k(\cdot|s)} \left[ \overline{Q}_{i,h}^{k,\rho_i}(s, \mathbf{a}) - \underline{Q}_{i,h}^{k,\rho_i}(s, \mathbf{a}) \right] \overset{(d)}{\le} \mathbb{E}_{\mathbf{a} \sim \pi^k(\cdot|s)} \left[ \widetilde{Q}_h^{k,\rho_{\min}}(s, \mathbf{a}) \right]$$

$$\overset{(e)}{=} \widetilde{V}_h^{k,\rho_{\min}}(s).$$

where ineq. (a) holds by eq. 33 which is true for $h + 1$ and by the def. of $\rho^\star$ which implies $\mathrm{Var}_{P_h^\star(\cdot|s,\mathbf{a})} \left[ V_{i,h+1}^{\pi^k,\rho_i} \right] \le \mathrm{Var}_{P_h^\star(\cdot|s,\mathbf{a})} \left[ V_{i,h+1}^{\pi^k,\rho^\star} \right]$; equality (b) holds by the definition of $\widetilde{Q}_h^{k,\rho_{\min}}(s, \mathbf{a})$ as in eq. 30; equality (c) holds by eq. 29; ineq. (d) holds by (b); equality (e) holds by eq. 31. Therefore, we only need to upper bound the sum $\sum_{k=1}^K \widetilde{V}_1^{k,\rho_{\min}}(s_1^k)$. For simplicity, we define the following notations for the differences at any $(h, k) \in [H] \times [K]$:

$$\Delta_h^k := \widetilde{V}_h^{k,\rho_{\min}}(s_h^k), \tag{34}$$

$$\zeta_h^k := \Delta_h^k - \widetilde{Q}_h^{k,\rho_{\min}}(s_h^k, \mathbf{a}_h^k), \tag{35}$$

$$\xi_h^k := \mathbb{E}_{P_h^\star(\cdot|s_h^k,\mathbf{a}_h^k)}[\widetilde{V}_{h+1}^{k,\rho_{\min}}] - \Delta_{h+1}^k. \tag{36}$$

We can confirm that $\{\zeta_h^k\}_{(h,k)}$ and $\{\xi_h^k\}_{(h,k)}$ are martingale difference sequences with respect to their respective filtrations. By substituting eq. 30 into eq. 35, we get:

$$\Delta_h^k = \zeta_h^k + \widetilde{Q}_h^{k,\rho_{\min}}(s_h^k, \mathbf{a}_h^k)$$

$$\le \zeta_h^k + \left( 1 + \frac{20}{H} \right) \mathbb{E}_{P_h^\star(\cdot|s_h^k,\mathbf{a}_h^k)} \left[ \widetilde{V}_{h+1}^{k,\rho_{\min}} \right] + 4 \sqrt{\frac{c_1 \iota \mathrm{Var}_{P_h^\star(\cdot|s_h^k,\mathbf{a}_h^k)} \left[ V_{h+1}^{\pi^k,\rho^\star} \right]}{\{N_h^k(s_h^k, \mathbf{a}_h^k) \vee 1\}}}$$

$$+ \frac{4c_2 H^2 S \iota}{\{N_h^k(s_h^k, \mathbf{a}_h^k) \vee 1\}} + \sqrt{\frac{4}{K}}$$

$$= \zeta_h^k + \left( 1 + \frac{20}{H} \right) \xi_h^k + \left( 1 + \frac{20}{H} \right) \Delta_{h+1}^k + 4 \sqrt{\frac{c_1 \iota \mathrm{Var}_{P_h^\star(\cdot|s,\mathbf{a})} \left[ V_{h+1}^{\pi^k,\rho^\star} \right]}{\{N_h^k(s_h^k, \mathbf{a}_h^k) \vee 1\}}}$$

$$+ \frac{4c_2 H^2 S \iota}{\{N_h^k(s_h^k, \mathbf{a}_h^k) \vee 1\}} + \sqrt{\frac{4}{K}}. \tag{37}$$

By recursively applying eq. 37 and noting that $\left(1 + \frac{20}{H}\right)^h \leq \left(1 + \frac{20}{H}\right)^H \leq c$ for some constant $c \geq 0$, we can upper bound the right-hand side of eq. 22 as:

$$\text{Regret}_{\text{NASH}}(K) \leq \sum_{k=1}^{K} \Delta_1^k \leq c \sum_{k=1}^{K} \sum_{h=1}^{H} \left\{ (\zeta_h^k + \xi_h^k) \right.$$
$$\left. + \left( 4 \sqrt{\frac{c_1 \iota \text{Var}_{P_h^\star(\cdot|s,\mathbf{a})}\left[V_{h+1}^{\pi^k, \rho^*}\right]}{\{N_h^k(s, \mathbf{a}) \vee 1\}}} + \frac{4 c_2 H^2 S \iota}{\{N_h^k(s, \mathbf{a}) \vee 1\}} \right) + \sqrt{\frac{4}{K}} \right\}. \tag{38}$$

The first term, a sum of martingale differences, is bounded using the Azuma-Hoeffding inequality from Lemma 39, yielding:

$$\sum_{k=1}^{K} \sum_{h=1}^{H} (\zeta_h^k + \xi_h^k) \leq c_1 \min\left\{ \frac{1}{\rho_{\min}}, H \right\} \sqrt{HK\iota}, \tag{39}$$

where $c_1 > 0$ is an absolute constant. For the second term, we apply the Cauchy–Schwarz inequality to the summation of the variance terms:

$$\sum_{k=1}^{K} \sum_{h=1}^{H} \sqrt{\frac{\text{Var}_{P_h^\star(\cdot|s_h^k, \mathbf{a}_h^k)}\left[V_{h+1}^{\pi^k, \rho^*}\right]}{N_h^k(s_h^k, \mathbf{a}_h^k) \vee 1}} \leq \sqrt{\left( \sum_{k=1}^{K} \sum_{h=1}^{H} \text{Var}_{P_h^\star(\cdot|s_h^k, \mathbf{a}_h^k)}\left[V_{h+1}^{\pi^k, \rho^*}\right] \right) \cdot}$$
$$\sqrt{\left( \sum_{k=1}^{K} \sum_{h=1}^{H} \frac{1}{N_h^k(s_h^k, a_h^k) \vee 1} \right)}. \tag{40}$$

The second factor on the right-hand side is bounded by $c_2 HS(\prod_{i=1}^{m} A_i)\iota$, as shown in (Liu et al., 2021, Theorem 3), while the first factor is bounded using the Law of Total Variation and standard martingale concentration arguments (from (Jin et al., 2018) and (Lu et al., 2024)):

$$\sum_{k=1}^{K} \sum_{h=1}^{H} \text{Var}_{P_h^\star(\cdot|s_h^k, \mathbf{a}_h^k)}\left[V_{h+1}^{\pi^k, \rho^*}\right] \leq c_3 \cdot \left( \min\left\{\frac{1}{\rho^\star}, H\right\} HK + \min\left\{\frac{1}{\rho^\star}, H\right\}^3 H\iota \right).$$
$$\overset{(a)}{\leq} c_3 \cdot \left( \min\left\{\frac{1}{\rho_{\min}}, H\right\} HK + \min\left\{\frac{1}{\rho_{\min}}, H\right\}^3 H\iota \right). \tag{41}$$

where (a) uses the fact that $\rho^\star \geq \rho_{\min}$, which implies that $\min\{1/\rho^\star, H\} \leq \min\{1/\rho_{\min}, H\}$. By combining these bounds and substituting them into eq. 40, we can obtain a final bound for the second term. The third term, $\sum_{k=1}^{K} \sum_{h=1}^{H} \sqrt{\frac{4}{K}}$, is straightforwardly bounded by $c_5\sqrt{H^2 K}$. By combining the bounds for all three terms, we arrive at the final regret bound for $\text{Regret}_{\text{Nash}}(K)$:

$$\text{Regret}_{\text{NASH}}(K) = \mathcal{O}\left( \sqrt{\min\left\{\frac{1}{\rho_{\min}}, H\right\} H^2 SK\left(\prod_{i \in \mathcal{M}} A_i\right)\iota'} \right), \tag{42}$$

where $\iota' = \log^2\left(\frac{SHK \prod_{i \in \mathcal{M}} A_i}{\delta}\right)$. This completes the proof of Theorem 4. $\square$

**Remark 16.** *The methodology for bounding the regret for Correlated Equilibrium (CE) and Coarse Correlated Equilibrium (CCE) settings mirrors the approach outlined here for the Nash equilibrium in the TV-DRMG context. The proofs leverage Lemma 21 and Lemma 22, respectively.*

### F.2 KEY LEMMAS FOR TV-DRMG

**Lemma 17** (Gap between maximum and minimum (Lu et al., 2024)). *Consider any RMG $\mathcal{MG}_{rob} = \{\mathcal{S}, \mathcal{A}, H, \{\mathcal{P}_{TV}^{\rho_i}(P^\star)\}_{i=1}^{m}, r\}$. The robust value function $V_{i,h}^{\pi, \rho_i}$ for all $i \in \mathcal{M}$ and $h \in [H]$ associated with any joint policy $\pi$ satisfies*

$$\forall (i, h) \in \mathcal{M} \times [H] : \max_{s \in \mathcal{S}} V_{i,h}^{\pi, \rho_i}(s) - \min_{s \in \mathcal{S}} V_{i,h}^{\pi, \rho_i}(s) \leq \nu_H^{\rho_i},$$

*where $\nu_H^{\rho_i} := \min\left\{\frac{1}{\rho_i}, H - h + 1\right\} \leq \min\left\{\frac{1}{\rho_i}, H\right\}$.*

*Proof.* Refer to the proof-lines of Lemma 3 in (Shi et al., 2024). $\qquad\square$

**Lemma 18** (Bound of optimistic and pessimistic value estimators with bonus for TV-DRMG). *Under the typical event $\mathcal{E}_{TV}$ defined in eq. 21 and by setting the bonus $\beta_{i,h}^k$ as in eq. 20, it holds that*

$$\sigma_{\widehat{\mathcal{P}_{i,h}^{\rho_i}}(s,\boldsymbol{a})}\left[\overline{V}_{i,h+1}^{k,\rho_i}\right] - \sigma_{\mathcal{P}_{i,h}^{\rho_i}(s,\mathbf{a})}\left[\overline{V}_{i,h+1}^{k,\rho_i}\right]$$
$$+ \sigma_{\mathcal{P}_{i,h}^{\rho_i}(s,\mathbf{a})}\left[\underline{V}_{i,h+1}^{k,\rho_i}\right] - \sigma_{\widehat{\mathcal{P}_{i,h}^{\rho_i}}(s,\boldsymbol{a})}\left[\underline{V}_{i,h+1}^{k,\rho_i}\right] \ \leq \ 2\beta_{i,h}^k(s,\mathbf{a}).$$

*Proof.* Let's denote the term to be bounded as $A$.

$$A := \sigma_{\widehat{\mathcal{P}_{i,h}^{\rho_i}}(s,\boldsymbol{a})}\left[\overline{V}_{i,h+1}^{k,\rho_i}\right] - \sigma_{\mathcal{P}_{i,h}^{\rho_i}(s,\mathbf{a})}\left[\overline{V}_{i,h+1}^{k,\rho_i}\right] + \sigma_{\mathcal{P}_{i,h}^{\rho_i}(s,\mathbf{a})}\left[\underline{V}_{i,h+1}^{k,\rho_i}\right] - \sigma_{\widehat{\mathcal{P}_{i,h}^{\rho_i}}(s,\boldsymbol{a})}\left[\underline{V}_{i,h+1}^{k,\rho_i}\right]. \quad (43)$$

Under the high-probability event $\mathcal{E}_{\text{TV}}$ (as defined in eq. 21), we can apply the concentration inequality from Lemma 24 to upper bound $A$ as follows:

$$A \leq 2\sqrt{\frac{c_1 \operatorname{Var}_{\widehat{P}_h^k}\left(V_{i,h+1}^{\dagger,\pi_{-i}^k,\rho_i}\right)\iota}{N_h^k(s,\mathbf{a}) \vee 1}} + \frac{2\,\mathbb{E}_{\widehat{P}_h^k(\cdot|s,\mathbf{a})}\left[\overline{V}_{i,h+1}^{k,\rho_i} - \underline{V}_{i,h+1}^{k,\rho_i}\right]}{H} + \frac{2c_2'H^2S\iota}{N_h^k(s,\mathbf{a})\vee 1} + \frac{2}{\sqrt{K}}. \quad (44)$$

where $\iota = \log\left(S^2(\prod_{i=1}^m A_i)H^2K^{3/2}/\delta\right)$ and $c_1, c_2' > 0$ are absolute constants. By applying the result from Lemma 26 to the variance term in eq. 44, we obtain the required bound presented in the lemma statement. This concludes the proof. $\qquad\square$

**Lemma 19** (Bound of the bonus term for TV-DRMG). *Under the typical event $\mathcal{E}_{TV}$, the bonus term defined in 20 is bounded by*

$$\beta_{i,h}^k(s,\mathbf{a}) \leq \sqrt{\frac{c_1\iota \operatorname{Var}_{P_h^\star(\cdot|s,\mathbf{a})}\left[V_{i,h+1}^{\pi^k,\rho_i}\right]}{N_h^k(s,\mathbf{a}) \vee 1}} + \frac{5\,\mathbb{E}_{P_h^\star(\cdot|s,\mathbf{a})}\left[\overline{V}_{i,h+1}^{k,\rho_i} - \underline{V}_{i,h+1}^{k,\rho_i}\right]}{H}$$
$$+ \frac{c_2 H^2 S\iota}{N_h^k(s,\mathbf{a}) \vee 1} + \sqrt{\tfrac{1}{K}}.$$

*where $\iota = \log(S^3(\prod_{i=1}^m A_i)H^2K^{3/2}/\delta)$ and $c_1, c_2 > 0$ are constants.*

*Proof.* The proof-lines are similar to (Lu et al., 2024, Lemma E.4) or (Ghosh et al., 2025, Lemma K.3). Recall the bonus term defined in eq. 20. We need to bound the first and second term of eq. 20. We first bound the second term of $\beta_{i,h}^k(s,\mathbf{a})$ by using Lemma 25, and we get

$$\frac{2\mathbb{E}_{\widehat{P}_h^k(\cdot|s,\mathbf{a})}\left[\overline{V}_{i,h+1}^{k,\rho_i} - \underline{V}_{i,h+1}^{k,\rho_i}\right]}{H} \leq \left(\frac{2}{H} + \frac{2}{H^2}\right)\mathbb{E}_{P_h^\star(\cdot|s,\mathbf{a})}\left[\overline{V}_{i,h+1}^{k,\rho_i} - \underline{V}_{i,h+1}^{k,\rho_i}\right] + \frac{c_2'HS\iota}{\{N_h^k(s,\mathbf{a}) \vee 1\}}$$
$$\leq \frac{4\mathbb{E}_{P_h^\star(\cdot|s,\mathbf{a})}\left[\overline{V}_{i,h+1}^{k,\rho_i} - \underline{V}_{i,h+1}^{k,\rho_i}\right]}{H} + \frac{c_2'HS\iota}{\{N_h^k(s,\mathbf{a}) \vee 1\}}, \quad (45)$$

where the second inequality is from $H \geq 1$. We now bound the first term (variance term) of eq. 20 by using Lemma 27, which gives

$$\sqrt{\frac{c_1\iota \operatorname{Var}_{\widehat{P}_h^k(\cdot|s,\mathbf{a})}\left[\frac{\overline{V}_{i,h+1}^{k,\rho_i} + \underline{V}_{i,h+1}^{k,\rho_i}}{2}\right]}{N_h^k(s,\mathbf{a}) \vee 1}} \leq \sqrt{\frac{c_1'\iota \operatorname{Var}_{P_h^\star(\cdot|s,\mathbf{a})}\left[V_{i,h+1}^{\pi^k,\rho_i}\right]}{N_h^k(s,\mathbf{a}) \vee 1}}$$
$$+ \frac{\mathbb{E}_{P_h^\star(\cdot|s,\mathbf{a})}\left[\overline{V}_{i,h+1}^{k,\rho_i} - \underline{V}_{i,h+1}^{k,\rho_i}\right]}{H} \quad (46)$$
$$+ \frac{c_3 H^2 S\iota}{N_h^k(s,\mathbf{a}) \vee 1}.$$

where $c_3 > 0$ is an absolutely constant. Thus by combining eq. 45 and eq. 46 with the choice of bonus term in eq. 20, we can conclude the proof of Lemma 19. $\qquad\square$

NE Version: Optimistic and pessimistic estimation of the robust values for TV-DRMG.

Here we will proof the optimistic estimations are indeed upper bounds of the corresponding robust V-value and robust Q-value functions fro NE version.

**Lemma 20** (Optimistic and pessimistic estimation of the robust values for TV-DRMG for NE version). *By setting the bonus term $\beta_{i,h}^k$ as in eq. 20, with probability $1 - \delta$, for any $(s, \mathbf{a}, h, i)$ and $k \in [K]$, it holds that*

$$Q_{i,h}^{\dagger, \pi_{-i}^k, \rho_i}(s, \boldsymbol{a}) \leq \overline{Q}_{i,h}^{k,\rho_i}(s, \boldsymbol{a}), \quad \underline{Q}_{i,h}^{k,\rho_i}(s, \boldsymbol{a}) \leq Q_{i,h}^{\pi^k, \rho_i}(s, \boldsymbol{a}), \tag{47}$$

$$V_{i,h}^{\dagger, \pi_{-i}^k, \rho_i}(s) \leq \overline{V}_{i,h}^{k,\rho_i}(s), \quad \underline{V}_{i,h}^{k,\rho_i}(s) \leq V_{i,h}^{\pi^k, \rho_i}(s). \tag{48}$$

*Proof.* The proof-lines are similar to (Ghosh et al., 2025) adapted to the multi-agent case. We will run a proof for each inequality outlined in Lemma 20.

- **Ineq. 1:** To prove $Q_{i,h}^{\dagger, \pi_{-i}^k, \rho_i}(s, \boldsymbol{a}) \leq \overline{Q}_{i,h}^{k,\rho_i}(s, \boldsymbol{a})$.

- **Ineq. 2:** To prove $\underline{Q}_{i,h}^{k,\rho_i}(s, \boldsymbol{a}) \leq Q_{i,h}^{\pi^k, \rho_i}(s, \boldsymbol{a})$.

We know that, at step $h = H + 1$, $\overline{V}_{i,H+1}^{k,\rho_i}(s) = V_{i,H+1}^{\dagger, \pi_{-1}^k, \rho_i}(s) = 0$. Now, we assume that both eq. 47 and eq. 48 hold at the $(h + 1)$-th step.

- **Proof of Ineq. 1:** We first consider robust $Q$ at the $h$-th step. Then, by Proposition 9 (Robust Bellman Equation) and eq. 5, we have that

$$\overline{Q}_{i,h}^{k,\rho_i}(s, \boldsymbol{a}) - Q_{i,h}^{\dagger, \pi_{-i}^k, \rho_i}(s, \boldsymbol{a})$$

$$= \min \left\{ \sigma_{\widehat{\mathcal{P}}_{i,h}^{\rho_i}(s,\boldsymbol{a})} \left[ \overline{V}_{i,h+1}^{k,\rho_i} \right] - \sigma_{\mathcal{P}_{i,h}^{\rho_i}(s,\boldsymbol{a})} \left[ V_{i,h+1}^{\dagger, \pi_{-i}^k, \rho_i} \right] + \beta_{i,h}^k(s, \boldsymbol{a}), \nu_H^{\rho_i} - Q_{i,h}^{\dagger, \pi_{-i}^k, \rho_i}(s, \boldsymbol{a}) \right\}$$

$$\geq \min \left\{ \sigma_{\widehat{\mathcal{P}}_{i,h}^{\rho_i}(s,\boldsymbol{a})} \left[ V_{i,h+1}^{\dagger, \pi_{-i}^k, \rho_i} \right] - \sigma_{\mathcal{P}_{i,h}^{\rho_i}(s,\boldsymbol{a})} \left[ V_{i,h+1}^{\dagger, \pi_{-i}^k, \rho_i} \right] + \beta_{i,h}^k(s, \boldsymbol{a}), 0 \right\}, \tag{49}$$

where the second inequality follows from the induction of $V_{i,h+1}^{\dagger, \pi_{-i}^k, \rho_i} \leq \overline{V}_{i,h+1}^{k,\rho_i}$ at the $h+1$-th step and the fact that $Q_{i,h}^{\dagger, \pi_{-i}^k, \rho_i} \leq \nu_H^{\rho_i}$ by Lemma 17. By Lemma 23, we get

$$\sigma_{\widehat{\mathcal{P}}_{i,h}^{\rho_i}(s,\boldsymbol{a})} \left[ V_{i,h+1}^{\dagger, \pi_{-i}^k, \rho_i} \right] - \sigma_{\mathcal{P}_{i,h}^{\rho_i}(s,\boldsymbol{a})} \left[ V_{i,h+1}^{\dagger, \pi_{-i}^k, \rho_i} \right] \leq \sqrt{\frac{c_1 \mathrm{Var}_{\widehat{P}_h^k} \left( V_{i,h+1}^{\dagger, \pi_{-i}^k, \rho_i} \right) \cdot \iota}{\{N_h^k(s, \boldsymbol{a}) \vee 1\}}}$$

$$+ \frac{c_2 H \iota}{\{N_h^k(s, \boldsymbol{a}) \vee 1\}} + \frac{1}{\sqrt{K}}. \tag{50}$$

Now by further applying Lemma 26 to the variance term in the above inequality, we can obtain that

$$
\sigma_{\widehat{\mathcal{P}_{i,h}^{\rho_i}}(s,\boldsymbol{a})}\left[V_{i,h+1}^{\dagger,\pi_{-i}^k,\rho_i}\right] - \sigma_{\mathcal{P}_{i,h}^{\rho_i}(s,\boldsymbol{a})}\left[V_{i,h+1}^{\dagger,\pi_{-i}^k,\rho_i}\right]
$$

$$
\leq \sqrt{\frac{c_1\left(\mathrm{Var}_{\widehat{P}_h^k(\cdot|s,\boldsymbol{a})}\left[\left(\frac{\overline{V}_{i,h+1}^{k,\rho_i}+\underline{V}_{i,h+1}^{k,\rho_i}}{2}\right)\right] + 4H\mathbb{E}_{\widehat{P}_h^k(\cdot|s,\boldsymbol{a})}\left[\overline{V}_{i,h+1}^{k,\rho_i} - \underline{V}_{i,h+1}^{k,\rho_i}\right]\right)\iota}{\{N_h^k(s,\boldsymbol{a}) \vee 1\}}}
$$

$$
+ \frac{c_2 H \iota}{\{N_h^k(s,\boldsymbol{a}) \vee 1\}} + \frac{1}{\sqrt{K}}
$$

$$
\overset{(i)}{\leq} \sqrt{\frac{c_1 \iota \mathrm{Var}_{\widehat{P}_h^k(\cdot|s,\boldsymbol{a})}\left[\left(\frac{\overline{V}_{i,h+1}^{k,\rho_i}+\underline{V}_{i,h+1}^{k,\rho_i}}{2}\right)\right]}{\{N_h^k(s,\boldsymbol{a}) \vee 1\}}} + \sqrt{\frac{4Hc_1\iota\mathbb{E}_{\widehat{P}_h^k(\cdot|s,\boldsymbol{a})}\left[\overline{V}_{i,h+1}^{k,\rho_i} - \underline{V}_{i,h+1}^{k,\rho_i}\right]}{\{N_h^k(s,\boldsymbol{a}) \vee 1\}}}
$$

$$
+ \frac{c_2 H \iota}{\{N_h^k(s,\boldsymbol{a}) \vee 1\}} + \frac{1}{\sqrt{K}}
$$

$$
\overset{(ii)}{\leq} \sqrt{\frac{c_1\iota\mathrm{Var}_{\widehat{P}_h^k(\cdot|s,\boldsymbol{a})}\left[\left(\frac{\overline{V}_{i,h+1}^{k,\rho_i}+\underline{V}_{i,h+1}^{k,\rho_i}}{2}\right)\right]}{\{N_h^k(s,\boldsymbol{a}) \vee 1\}}} + \frac{\mathbb{E}_{\widehat{P}_h^k(\cdot|s,\boldsymbol{a})}\left[\overline{V}_{i,h+1}^{k,\rho_i} - \underline{V}_{i,h+1}^{k,\rho_i}\right]}{H}
$$

$$
+ \frac{H^2 c_2' \iota}{\{N_h^k(s,\boldsymbol{a}) \vee 1\}} + \frac{1}{\sqrt{K}}, \tag{51}
$$

where the inequality (i) is due to $\sqrt{a+b} \leq \sqrt{a} + \sqrt{b}$, and the last inequality (ii) is from $\sqrt{ab} \leq a + b$ where $c_2' > 0$ is an absolute constant. Therefore, combining eqns. 49, 50, 51, and the choice of bonus in 20, we can conclude that $\overline{Q}_{i,h}^{k,\rho_i}(s,\boldsymbol{a}) - Q_{i,h}^{\dagger,\pi_{-i}^k,\rho_i}(s,\boldsymbol{a}) \geq 0$.

- **Proof of Ineq. 2:** By Proposition 9 (Robust Bellman Equation) and eq. 6, we have that

$$
\underline{Q}_{i,h}^{k,\rho_i}(s,\boldsymbol{a}) - Q_{i,h}^{\pi^k,\rho_i}(s,\boldsymbol{a})
$$

$$
= \max\left\{\sigma_{\widehat{\mathcal{P}_{i,h}^{\rho_i}}(s,\boldsymbol{a})}\left[\underline{V}_{i,h+1}^{k,\rho_i}\right] - \sigma_{\mathcal{P}_{i,h}^{\rho_i}(s,\boldsymbol{a})}\left[V_{i,h+1}^{\pi^k,\rho_i}\right] - \beta_{i,h}^k(s,\boldsymbol{a}), 0 - Q_{i,h}^{\dagger,\pi_{-i}^k,\rho_i}(s,\boldsymbol{a})\right\},
$$

$$
\leq \max\left\{\sigma_{\widehat{\mathcal{P}_{i,h}^{\rho_i}}(s,\boldsymbol{a})}\left[V_{i,h+1}^{\pi^k,\rho_i}\right] - \sigma_{\mathcal{P}_{i,h}^{\rho_i}(s,\boldsymbol{a})}\left[V_{i,h+1}^{\pi^k,\rho_i}\right] - \beta_{i,h}^k(s,\boldsymbol{a}), 0\right\}, \tag{52}
$$

where the second inequality follows from the induction of $V_{i,h+1}^{\pi^k,\rho_i} \geq \underline{V}_{i,h+1}^{k,\rho_i}$ at the $h+1$-th step and the fact that $Q_{i,h}^{\pi^k,\rho_i} \geq 0$. By Lemma 23, we can confirm that

$$
\sigma_{\widehat{\mathcal{P}_{i,h}^{\rho_i}}(s,\boldsymbol{a})}\left[V_{i,h+1}^{\pi^k,\rho_i}\right] - \sigma_{\mathcal{P}_{i,h}^{\rho_i}(s,\boldsymbol{a})}\left[V_{i,h+1}^{\pi^k,\rho_i}\right] \leq \sqrt{\frac{c_1\mathrm{Var}_{\widehat{P}_h^k}\left(V_{i,h+1}^{\dagger,\pi_{-i}^k,\rho_i}\right)\cdot\iota}{\{N_h^k(s,\boldsymbol{a}) \vee 1\}}}
$$

$$
+ \frac{\mathbb{E}_{\widehat{P}_h^k(\cdot|s,\boldsymbol{a})}\left[\overline{V}_{i,h+1}^{k,\rho_i} - \underline{V}_{i,h+1}^{k,\rho_i}\right)\right]}{H}
$$

$$
+ \frac{c_2' H^2 S\iota}{\{N_h^k(s,\boldsymbol{a}) \vee 1\}} + \frac{1}{\sqrt{K}}. \tag{53}
$$

Now by further applying Lemma 26 to the variance term in the above inequality, with an argument similar to eq. 50 we can obtain that

$$
\sigma_{\widehat{\mathcal{P}^{\rho_i}_{i,h}}(s,\boldsymbol{a})}\left[V^{\pi^k,\rho_i}_{i,h+1}\right] - \sigma_{\mathcal{P}^{\rho_i}_{i,h}(s,\boldsymbol{a})}\left[V^{\pi^k,\rho_i}_{i,h+1}\right] \leq \sqrt{\frac{c_1 \mathrm{Var}_{\widehat{P}^k_h}\left(V^{\dagger,\pi^k_{-i},\rho_i}_{i,h+1}\right)\cdot\iota}{\{N^k_h(s,\mathbf{a})\vee 1\}}}
$$
$$
+ \frac{\mathbb{E}_{\widehat{P}^k_h(\cdot|s,\mathbf{a})}\left[\overline{V}^{k,\rho_i}_{i,h+1} - \underline{V}^{k,\rho_i}_{i,h+1})\right]}{H}
$$
$$
+ \frac{c''_2 H^2 S\iota}{\{N^k_h(s,\mathbf{a})\vee 1\}} + \frac{1}{\sqrt{K}}. \quad (54)
$$

where $c''_2 > 0$ is an absolute constant. Therefore, combining eqns. 52, 53, 54, and the choice of bonus in 20, $\underline{Q}^{k,\rho_i}_{i,h}(s,\boldsymbol{a}) - Q^{\pi^k,\rho_i}_{i,h}(s,\boldsymbol{a}) \leq 0$.

Therefore, by eq. 51 and eq. 54, we have proved that at step $h$, it holds that

$$
Q^{\dagger,\pi^k_{-i},\rho_i}_{i,h}(s,\boldsymbol{a}) \leq \overline{Q}^{k,\rho_i}_{i,h}(s,\boldsymbol{a}), \quad \underline{Q}^{k,\rho_i}_{i,h}(s,\boldsymbol{a}) \leq Q^{\pi^k,\rho_i}_{i,h}(s,\boldsymbol{a}). \quad (55)
$$

We now assume that eq. 47 hold for $h$-th step. Then, by the definition of robust value function as given by robust Bellman equation (Proposition 9), and eq. 8, and NASH Equilibrium, we get

$$
\overline{V}^{k,\rho_i}_{i,h}(s) = \mathbb{E}_{\boldsymbol{a}\sim\pi^k(\cdot|s)}\left[\overline{Q}^{k,\rho_i}_{i,h}(s,\mathbf{a})\right] = \max_{\pi'_i}\mathbb{E}_{\boldsymbol{a}\sim\pi'_i\times\pi^k_{-i}(\cdot|s)}\left[\overline{Q}^{k,\rho_i}_{i,h}(s,\mathbf{a})\right]. \quad (56)
$$

By the definition of $V^{\dagger,\pi^k_{-i},\rho_i}_{i,h}(s)$ in eq. 3, we get

$$
V^{\dagger,\pi^k_{-i},\rho_i}_{i,h}(s) = \max_{\pi'_i}\mathbb{E}_{\boldsymbol{a}\sim\pi'_i\times\pi^k_{-i}(\cdot|s)}\left[Q^{\dagger,\pi^k_{-i},\rho_i}_{i,h}(s,\mathbf{a})\right]. \quad (57)
$$

Since by induction, for any $(s,\mathbf{a})$, $\overline{Q}^{k,\rho_i}_{i,h}(s,\mathbf{a}) \geq Q^{\dagger,\pi^k_{-i},\rho_i}_{i,h}(s,\mathbf{a})$. As a result, we also have $\overline{V}^{k,\rho_i}_{i,h}(s) \geq V^{\dagger,\pi^k_{-i},\rho_i}_{i,h}(s)$, which is eq. 48 for $h$-th step. Similarly, we can show that

$$
\underline{V}^{k,\rho_i}_{i,h}(s) = \mathbb{E}_{\boldsymbol{a}\sim\pi^k(\cdot|s)}\left[\underline{Q}^{k,\rho_i}_{i,h}(s,\mathbf{a})\right],
$$
$$
\overset{(i)}{\leq} \mathbb{E}_{\boldsymbol{a}\sim\pi^k(\cdot|s)}\left[Q^{\pi^k,\rho_i}_{i,h}(s,\mathbf{a})\right],
$$
$$
\overset{(ii)}{=} V^{\pi^k,\rho_i}_{i,h}(s), \quad (58)
$$

where (i) is due to the fact that $\underline{Q}^{k,\rho_i}_{i,h}(s,\boldsymbol{a}) \leq Q^{\pi^k,\rho_i}_{i,h}(s,\boldsymbol{a})$ and (ii) is by definition of $V^{\pi^k,\rho_i}_{i,h}(s)$ as given by Bellman equation in Proposition 9. $\square$

CCE VERSION: OPTIMISTIC AND PESSIMISTIC ESTIMATION OF THE ROBUST VALUES FOR TV-DRMG.

Here we will proof the optimistic estimations are indeed upper bounds of the corresponding robust V-value and robust Q-value functions for CCE version.

**Lemma 21** (Optimistic and pessimistic estimation of the robust values for TV-DRMG for CCE version). *By setting the bonus term $\beta^k_{i,h}$ as in eq. 20, with probability $1-\delta$, for any $(s,\mathbf{a},h,i)$ and $k\in[K]$, it holds that*

$$
\max_{\phi\in\Phi_i}Q^{\phi\diamond\pi^k,\rho_i}_{i,h}(s,\boldsymbol{a}) \leq \overline{Q}^{k,\rho_i}_{i,h}(s,\boldsymbol{a}), \quad \underline{Q}^{k,\rho_i}_{i,h}(s,\boldsymbol{a}) \leq Q^{\pi^k,\rho_i}_{i,h}(s,\boldsymbol{a}), \quad (59)
$$

$$
\max_{\phi\in\Phi_i}V^{\phi\diamond\pi^k,\rho_i}_{i,h}(s) \leq \overline{V}^{k,\rho_i}_{i,h}(s), \quad \underline{V}^{k,\rho_i}_{i,h}(s) \leq V^{\pi^k,\rho_i}_{i,h}(s). \quad (60)
$$

*Proof.* The proof-lines are similar to (Ghosh et al., 2025) adapted to the multi-agent case. We will run a proof for each inequality outlined in Lemma 21.

- **Ineq. 1:** To prove $Q_{i,h}^{\dagger,\pi_{-i}^k,\rho_i}(s,\boldsymbol{a}) \leq \overline{Q}_{i,h}^{k,\rho_i}(s,\boldsymbol{a})$.

- **Ineq. 2:** To prove $\underline{Q}_{i,h}^{k,\rho_i}(s,\boldsymbol{a}) \leq Q_{i,h}^{\pi^k,\rho_i}(s,\boldsymbol{a})$.

We know that, at step $h = H+1$, $\overline{V}_{i,H+1}^{k,\rho_i}(s) = V_{i,H+1}^{\dagger,\pi_{-1}^k,\rho_i}(s) = 0$. Now, we assume that both eq. 59 and eq. 60 hold at the $(h+1)$-th step.

- **Proof of Ineq. 1:** We first consider robust $Q$ at the $h$-th step. Then, by Proposition 9 (Robust Bellman Equation) and eq. 5, we have that

$$\overline{Q}_{i,h}^{k,\rho_i}(s,\boldsymbol{a}) - Q_{i,h}^{\dagger,\pi_{-i}^k,\rho_i}(s,\boldsymbol{a})$$
$$= \min\left\{\sigma_{\widehat{\mathcal{P}_{i,h}^{\rho_i}}(s,\boldsymbol{a})}\left[\overline{V}_{i,h+1}^{k,\rho_i}\right] - \sigma_{\mathcal{P}_{i,h}^{\rho_i}(s,\boldsymbol{a})}\left[V_{i,h+1}^{\dagger,\pi_{-i}^k,\rho_i}\right] + \beta_{i,h}^k(s,\boldsymbol{a}), \nu_H^{\rho_i} - Q_{i,h}^{\dagger,\pi_{-i}^k,\rho_i}(s,\boldsymbol{a})\right\},$$
$$\geq \min\left\{\sigma_{\widehat{\mathcal{P}_{i,h}^{\rho_i}}(s,\boldsymbol{a})}\left[V_{i,h+1}^{\dagger,\pi_{-i}^k,\rho_i}\right] - \sigma_{\mathcal{P}_{i,h}^{\rho_i}(s,\boldsymbol{a})}\left[V_{i,h+1}^{\dagger,\pi_{-i}^k,\rho_i}\right] + \beta_{i,h}^k(s,\boldsymbol{a}),\, 0\right\}, \tag{61}$$

where the second inequality follows from the induction of $V_{i,h+1}^{\dagger,\pi_{-i}^k,\rho_i} \leq \overline{V}_{i,h+1}^{k,\rho_i}$ at the $h+1$-th step and the fact that $Q_{i,h}^{\dagger,\pi_{-i}^k,\rho_i} \leq \nu_H^{\rho_i}$ by Lemma 17. By Lemma 23, we get

$$\sigma_{\widehat{\mathcal{P}_{i,h}^{\rho_i}}(s,\boldsymbol{a})}\left[V_{i,h+1}^{\dagger,\pi_{-i}^k,\rho_i}\right] - \sigma_{\mathcal{P}_{i,h}^{\rho_i}(s,\boldsymbol{a})}\left[V_{i,h+1}^{\dagger,\pi_{-i}^k,\rho_i}\right] \leq \sqrt{\frac{c_1 \mathrm{Var}_{\widehat{P}_h^k}\left(V_{i,h+1}^{\dagger,\pi_{-i}^k,\rho_i}\right) \cdot \iota}{\{N_h^k(s,\mathbf{a}) \vee 1\}}}$$
$$+ \frac{c_2 H \iota}{\{N_h^k(s,\mathbf{a}) \vee 1\}} + \frac{1}{\sqrt{K}}. \tag{62}$$

Now by further applying Lemma 26 to the variance term in the above inequality, we can obtain that

$$\sigma_{\widehat{\mathcal{P}_{i,h}^{\rho_i}}(s,\boldsymbol{a})}\left[V_{i,h+1}^{\dagger,\pi_{-i}^k,\rho_i}\right] - \sigma_{\mathcal{P}_{i,h}^{\rho_i}(s,\boldsymbol{a})}\left[V_{i,h+1}^{\dagger,\pi_{-i}^k,\rho_i}\right]$$

$$\leq \sqrt{\frac{c_1\left(\mathrm{Var}_{\widehat{P}_h^k(\cdot|s,\mathbf{a})}\left[\left(\frac{\overline{V}_{i,h+1}^{k,\rho_i}+\underline{V}_{i,h+1}^{k,\rho_i}}{2}\right)\right] + 4H\mathbb{E}_{\widehat{P}_h^k(\cdot|s,\mathbf{a})}\left[\overline{V}_{i,h+1}^{k,\rho_i} - \underline{V}_{i,h+1}^{k,\rho_i}\right]\right)\iota}{\{N_h^k(s,\mathbf{a}) \vee 1\}}}$$
$$+ \frac{c_2 H \iota}{\{N_h^k(s,\mathbf{a}) \vee 1\}} + \frac{1}{\sqrt{K}}$$

$$\overset{(i)}{\leq} \sqrt{\frac{c_1 \iota \mathrm{Var}_{\widehat{P}_h^k(\cdot|s,\mathbf{a})}\left[\left(\frac{\overline{V}_{i,h+1}^{k,\rho_i}+\underline{V}_{i,h+1}^{k,\rho_i}}{2}\right)\right]}{\{N_h^k(s,\mathbf{a}) \vee 1\}}} + \sqrt{\frac{4Hc_1\iota\mathbb{E}_{\widehat{P}_h^k(\cdot|s,\mathbf{a})}\left[\overline{V}_{i,h+1}^{k,\rho_i} - \underline{V}_{i,h+1}^{k,\rho_i}\right]}{\{N_h^k(s,\mathbf{a}) \vee 1\}}}$$
$$+ \frac{c_2 H \iota}{\{N_h^k(s,\mathbf{a}) \vee 1\}} + \frac{1}{\sqrt{K}}$$

$$\overset{(ii)}{\leq} \sqrt{\frac{c_1 \iota \mathrm{Var}_{\widehat{P}_h^k(\cdot|s,\mathbf{a})}\left[\left(\frac{\overline{V}_{i,h+1}^{k,\rho_i}+\underline{V}_{i,h+1}^{k,\rho_i}}{2}\right)\right]}{\{N_h^k(s,\mathbf{a}) \vee 1\}}} + \frac{\mathbb{E}_{\widehat{P}_h^k(\cdot|s,\mathbf{a})}\left[\overline{V}_{i,h+1}^{k,\rho_i} - \underline{V}_{i,h+1}^{k,\rho_i}\right]}{H}$$
$$+ \frac{H^2 c_2' \iota}{\{N_h^k(s,\mathbf{a}) \vee 1\}} + \frac{1}{\sqrt{K}}, \tag{63}$$

where the inequality (i) is due to $\sqrt{a+b} \leq \sqrt{a} + \sqrt{b}$, and the last inequality (ii) is from $\sqrt{ab} \leq a + b$ where $c_2' > 0$ is an absolute constant. Therefore, combining eqns. 61, 62, 63, and the choice of bonus in 20, we can conclude that $\overline{Q}_{i,h}^{k,\rho_i}(s,\boldsymbol{a}) - Q_{i,h}^{\dagger,\pi_{-i}^k,\rho_i}(s,\boldsymbol{a}) \geq 0$.

- **Proof of Ineq. 2:** By Proposition 9 (Robust Bellman Equation) and eq. 6, we have that

$$\underline{Q}_{i,h}^{k,\rho_i}(s,\boldsymbol{a}) - Q_{i,h}^{\pi^k,\rho_i}(s,\boldsymbol{a})$$

$$= \max \left\{ \sigma_{\widehat{\mathcal{P}}_{i,h}^{\rho_i}(s,\boldsymbol{a})} \left[ \underline{V}_{i,h+1}^{k,\rho_i} \right] - \sigma_{\mathcal{P}_{i,h}^{\rho_i}(s,\boldsymbol{a})} \left[ V_{i,h+1}^{\pi^k,\rho_i} \right] - \beta_{i,h}^k(s,\boldsymbol{a}), 0 - Q_{i,h}^{\dagger,\pi_{-i}^k,\rho_i}(s,\boldsymbol{a}) \right\},$$

$$\leq \max \left\{ \sigma_{\widehat{\mathcal{P}}_{i,h}^{\rho_i}(s,\boldsymbol{a})} \left[ V_{i,h+1}^{\pi^k,\rho_i} \right] - \sigma_{\mathcal{P}_{i,h}^{\rho_i}(s,\boldsymbol{a})} \left[ V_{i,h+1}^{\pi^k,\rho_i} \right] - \beta_{i,h}^k(s,\boldsymbol{a}), \, 0 \right\}, \tag{64}$$

where the second inequality follows from the induction of $V_{i,h+1}^{\pi^k,\rho_i} \geq \underline{V}_{i,h+1}^{k,\rho_i}$ at the $h+1$-th step and the fact that $Q_{i,h}^{\pi^k,\rho_i} \geq 0$. By Lemma 23, we can confirm that

$$\sigma_{\widehat{\mathcal{P}}_{i,h}^{\rho_i}(s,\boldsymbol{a})} \left[ V_{i,h+1}^{\pi^k,\rho_i} \right] - \sigma_{\mathcal{P}_{i,h}^{\rho_i}(s,\boldsymbol{a})} \left[ V_{i,h+1}^{\pi^k,\rho_i} \right] \leq \sqrt{\frac{c_1 \mathrm{Var}_{\widehat{P}_h^k} \left( V_{i,h+1}^{\dagger,\pi_{-i}^k,\rho_i} \right) \cdot \iota}{\{N_h^k(s,\boldsymbol{a}) \vee 1\}}}$$

$$+ \frac{\mathbb{E}_{\widehat{P}_h^k(\cdot|s,\boldsymbol{a})} \left[ \overline{V}_{i,h+1}^{k,\rho_i} - \underline{V}_{i,h+1}^{k,\rho_i} \right]}{H}$$

$$+ \frac{c_2' H^2 S \iota}{\{N_h^k(s,\boldsymbol{a}) \vee 1\}} + \frac{1}{\sqrt{K}}. \tag{65}$$

Now by further applying Lemma 26 to the variance term in the above inequality, with an argument similar to eq. 62 we can obtain that

$$\sigma_{\widehat{\mathcal{P}}_{i,h}^{\rho_i}(s,\boldsymbol{a})} \left[ V_{i,h+1}^{\pi^k,\rho_i} \right] - \sigma_{\mathcal{P}_{i,h}^{\rho_i}(s,\boldsymbol{a})} \left[ V_{i,h+1}^{\pi^k,\rho_i} \right] \leq \sqrt{\frac{c_1 \mathrm{Var}_{\widehat{P}_h^k} \left( V_{i,h+1}^{\dagger,\pi_{-i}^k,\rho_i} \right) \cdot \iota}{\{N_h^k(s,\boldsymbol{a}) \vee 1\}}}$$

$$+ \frac{\mathbb{E}_{\widehat{P}_h^k(\cdot|s,\boldsymbol{a})} \left[ \overline{V}_{i,h+1}^{k,\rho_i} - \underline{V}_{i,h+1}^{k,\rho_i} \right]}{H}$$

$$+ \frac{c_2'' H^2 S \iota}{\{N_h^k(s,\boldsymbol{a}) \vee 1\}} + \frac{1}{\sqrt{K}}. \tag{66}$$

where $c_2'' > 0$ is an absolute constant. Therefore, combining eqns. 64, 65, 66, and the choice of bonus in 20, $\underline{Q}_{i,h}^{k,\rho_i}(s,\boldsymbol{a}) - Q_{i,h}^{\pi^k,\rho_i}(s,\boldsymbol{a}) \leq 0$.

Therefore, by eq. 63 and eq. 66, we have proved that at step $h$, it holds that

$$Q_{i,h}^{\dagger,\pi_{-i}^k,\rho_i}(s,\boldsymbol{a}) \leq \overline{Q}_{i,h}^{k,\rho_i}(s,\boldsymbol{a}), \quad \underline{Q}_{i,h}^{k,\rho_i}(s,\boldsymbol{a}) \leq Q_{i,h}^{\pi^k,\rho_i}(s,\boldsymbol{a}). \tag{67}$$

We now assume that eq. 59 hold for $h$-th step. Then, by the definition of robust value function as given by robust Bellman equation (Proposition 9), eq. 8, and CCE Equilibrium, we get

$$\overline{V}_{i,h}^{k,\rho_i}(s) = \mathbb{E}_{\boldsymbol{a} \sim \pi^k(\cdot|s)} \left[ \overline{Q}_{i,h}^{k,\rho_i}(s,\mathbf{a}) \right] \geq \max_{\pi_i'} \mathbb{E}_{\boldsymbol{a} \sim \pi_i' \times \pi_{-i}^k(\cdot|s)} \left[ \overline{Q}_{i,h}^{k,\rho_i}(s,\mathbf{a}) \right], \tag{68}$$

By the definition of $V_{i,h}^{\dagger,\pi_{-i}^k,\rho_i}(s)$ in eq. 3, we get

$$V_{i,h}^{\dagger,\pi_{-i}^k,\rho_i}(s) = \max_{\pi_i'} \mathbb{E}_{\boldsymbol{a} \sim \pi_i' \times \pi_{-i}^k(\cdot|s)} \left[ Q_{i,h}^{\dagger,\pi_{-i}^k,\rho_i}(s,\mathbf{a}) \right]. \tag{69}$$

Since by induction, for any $(s, \mathbf{a})$, $\overline{Q}_{i,h}^{k,\rho_i}(s, \mathbf{a}) \geq Q_{i,h}^{\dagger, \pi_{-i}^k, \rho_i}(s, \mathbf{a})$. As a result, we also have $\overline{V}_{i,h}^{k,\rho_i}(s) \geq V_{i,h}^{\dagger, \pi_{-i}^k, \rho_i}(s)$, which is eq. 60 for $h$-th step. Similarly, we can show that

$$
\begin{aligned}
\underline{V}_{i,h}^{k,\rho_i}(s) &= \mathbb{E}_{\mathbf{a} \sim \pi^k(\cdot|s)} \left[ \underline{Q}_{i,h}^{k,\rho_i}(s, \mathbf{a}) \right], \\
&\overset{(i)}{\leq} \mathbb{E}_{\mathbf{a} \sim \pi^k(\cdot|s)} \left[ Q_{i,h}^{\pi^k, \rho_i}(s, \mathbf{a}) \right], \\
&\overset{(ii)}{=} V_{i,h}^{\pi^k, \rho_i}(s),
\end{aligned}
\tag{70}
$$

where (i) is due to the fact that $\underline{Q}_{i,h}^{k,\rho_i}(s, \mathbf{a}) \leq Q_{i,h}^{\pi^k, \rho_i}(s, \mathbf{a})$ and (ii) is by definition of $V_{i,h}^{\pi^k, \rho_i}(s)$ as given by Bellman equation in Proposition 9. $\qquad \square$

CE VERSION: OPTIMISTIC AND PESSIMISTIC ESTIMATION OF THE ROBUST VALUES FOR TV-DRMG.

Here we will proof the optimistic estimations are indeed upper bounds of the corresponding robust V-value and robust Q-value functions for CE version.

**Lemma 22** (Optimistic and pessimistic estimation of the robust values for TV-DRMG for CE version). *By setting the bonus term $\beta_{i,h}^k$ as in eq. 20, with probability $1 - \delta$, for any $(s, \mathbf{a}, h, i)$ and $k \in [K]$, it holds that*

$$
Q_{i,h}^{\dagger, \pi_{-i}^k, \rho_i}(s, \mathbf{a}) \leq \overline{Q}_{i,h}^{k,\rho_i}(s, \mathbf{a}), \quad \underline{Q}_{i,h}^{k,\rho_i}(s, \mathbf{a}) \leq Q_{i,h}^{\pi^k, \rho_i}(s, \mathbf{a}),
\tag{71}
$$

$$
V_{i,h}^{\dagger, \pi_{-i}^k, \rho_i}(s) \leq \overline{V}_{i,h}^{k,\rho_i}(s), \quad \underline{V}_{i,h}^{k,\rho_i}(s) \leq V_{i,h}^{\pi^k, \rho_i}(s).
\tag{72}
$$

*Proof.* The proof-lines are similar to (Ghosh et al., 2025) adapted to the multi-agent case. We will run a proof for each inequality outlined in Lemma 22.

- **Ineq. 1:** To prove $Q_{i,h}^{\dagger, \pi_{-i}^k, \rho_i}(s, \mathbf{a}) \leq \overline{Q}_{i,h}^{k,\rho_i}(s, \mathbf{a})$.

- **Ineq. 2:** To prove $\underline{Q}_{i,h}^{k,\rho_i}(s, \mathbf{a}) \leq Q_{i,h}^{\pi^k, \rho_i}(s, \mathbf{a})$.

We know that, at step $h = H + 1$, $\overline{V}_{i,H+1}^{k,\rho_i}(s) = V_{i,H+1}^{\dagger, \pi_{-i}^k, \rho_i}(s) = 0$. Now, we assume that both eq. 71 and eq. 72 hold at the $(h+1)$-th step.

- **Proof of Ineq. 1:** We first consider robust $Q$ at the $h$-th step. Then, by Proposition 9 (Robust Bellman Equation) and eq. 5, we have that

$$
\begin{aligned}
&\overline{Q}_{i,h}^{k,\rho_i}(s, \mathbf{a}) - Q_{i,h}^{\dagger, \pi_{-i}^k, \rho_i}(s, \mathbf{a}) \\
&= \min \left\{ \sigma_{\widehat{\mathcal{P}}_{i,h}^{\rho_i}(s, \mathbf{a})} \left[ \overline{V}_{i,h+1}^{k,\rho_i} \right] - \sigma_{\mathcal{P}_{i,h}^{\rho_i}(s, \mathbf{a})} \left[ V_{i,h+1}^{\dagger, \pi_{-i}^k, \rho_i} \right] + \beta_{i,h}^k(s, \mathbf{a}), \nu_H^{\rho_i} - Q_{i,h}^{\dagger, \pi_{-i}^k, \rho_i}(s, \mathbf{a}) \right\}, \\
&\geq \min \left\{ \sigma_{\widehat{\mathcal{P}}_{i,h}^{\rho_i}(s, \mathbf{a})} \left[ V_{i,h+1}^{\dagger, \pi_{-i}^k, \rho_i} \right] - \sigma_{\mathcal{P}_{i,h}^{\rho_i}(s, \mathbf{a})} \left[ V_{i,h+1}^{\dagger, \pi_{-i}^k, \rho_i} \right] + \beta_{i,h}^k(s, \mathbf{a}), \, 0 \right\}.
\end{aligned}
\tag{73}
$$

where the second inequality follows from the induction of $V_{i,h+1}^{\dagger, \pi_{-i}^k, \rho_i} \leq \overline{V}_{i,h+1}^{k,\rho_i}$ at the $h+1$-th step and the fact that $Q_{i,h}^{\dagger, \pi_{-i}^k, \rho_i} \leq \nu_H^{\rho_i}$ by Lemma 17. By Lemma 23, we get

$$
\begin{aligned}
\sigma_{\widehat{\mathcal{P}}_{i,h}^{\rho_i}(s, \mathbf{a})} \left[ V_{i,h+1}^{\dagger, \pi_{-i}^k, \rho_i} \right] - \sigma_{\mathcal{P}_{i,h}^{\rho_i}(s, \mathbf{a})} \left[ V_{i,h+1}^{\dagger, \pi_{-i}^k, \rho_i} \right] &\leq \sqrt{\frac{c_1 \mathrm{Var}_{\widehat{P}_h^k} \left( V_{i,h+1}^{\dagger, \pi_{-i}^k, \rho_i} \right) \cdot \iota}{\{ N_h^k(s, \mathbf{a}) \vee 1 \}}} \\
&\quad + \frac{c_2 H \iota}{\{ N_h^k(s, \mathbf{a}) \vee 1 \}} + \frac{1}{\sqrt{K}}.
\end{aligned}
\tag{74}
$$

Now by further applying Lemma 26 to the variance term in the above inequality, we can obtain that

$$
\sigma_{\widehat{\mathcal{P}_{i,h}^{\rho_i}}(s,\boldsymbol{a})}\left[V_{i,h+1}^{\dagger,\pi_{-i}^k,\rho_i}\right] - \sigma_{\mathcal{P}_{i,h}^{\rho_i}(s,\boldsymbol{a})}\left[V_{i,h+1}^{\dagger,\pi_{-i}^k,\rho_i}\right]
$$

$$
\leq \sqrt{\frac{c_1\left(\mathrm{Var}_{\widehat{P}_h^k(\cdot|s,\mathbf{a})}\left[\left(\frac{\overline{V}_{i,h+1}^{k,\rho_i}+\underline{V}_{i,h+1}^{k,\rho_i}}{2}\right)\right] + 4H\mathbb{E}_{\widehat{P}_h^k(\cdot|s,\mathbf{a})}\left[\overline{V}_{i,h+1}^{k,\rho_i} - \underline{V}_{i,h+1}^{k,\rho_i}\right]\right)\iota}{\{N_h^k(s,\mathbf{a}) \vee 1\}}}
$$

$$
+ \frac{c_2 H\iota}{\{N_h^k(s,\mathbf{a}) \vee 1\}} + \frac{1}{\sqrt{K}}
$$

$$
\overset{(i)}{\leq} \sqrt{\frac{c_1\iota\mathrm{Var}_{\widehat{P}_h^k(\cdot|s,\mathbf{a})}\left[\left(\frac{\overline{V}_{i,h+1}^{k,\rho_i}+\underline{V}_{i,h+1}^{k,\rho_i}}{2}\right)\right]}{\{N_h^k(s,\mathbf{a}) \vee 1\}}} + \sqrt{\frac{4Hc_1\iota\mathbb{E}_{\widehat{P}_h^k(\cdot|s,\mathbf{a})}\left[\overline{V}_{i,h+1}^{k,\rho_i} - \underline{V}_{i,h+1}^{k,\rho_i}\right]}{\{N_h^k(s,\mathbf{a}) \vee 1\}}}
$$

$$
+ \frac{c_2 H\iota}{\{N_h^k(s,\mathbf{a}) \vee 1\}} + \frac{1}{\sqrt{K}}
$$

$$
\overset{(ii)}{\leq} \sqrt{\frac{c_1\iota\mathrm{Var}_{\widehat{P}_h^k(\cdot|s,\mathbf{a})}\left[\left(\frac{\overline{V}_{i,h+1}^{k,\rho_i}+\underline{V}_{i,h+1}^{k,\rho_i}}{2}\right)\right]}{\{N_h^k(s,\mathbf{a}) \vee 1\}}} + \frac{\mathbb{E}_{\widehat{P}_h^k(\cdot|s,\mathbf{a})}\left[\overline{V}_{i,h+1}^{k,\rho_i} - \underline{V}_{i,h+1}^{k,\rho_i}\right]}{H}
$$

$$
+ \frac{H^2 c_2'\iota}{\{N_h^k(s,\mathbf{a}) \vee 1\}} + \frac{1}{\sqrt{K}}, \tag{75}
$$

where the inequality (i) is due to $\sqrt{a+b} \leq \sqrt{a} + \sqrt{b}$, and the last inequality (ii) is from $\sqrt{ab} \leq a + b$ where $c_2' > 0$ is an absolute constant. Therefore, combining eqns. 73, 74, 75, and the choice of bonus in 20, we can conclude that $\overline{Q}_{i,h}^{k,\rho_i}(s,\boldsymbol{a}) - Q_{i,h}^{\dagger,\pi_{-i}^k,\rho_i}(s,\boldsymbol{a}) \geq 0$.

- **Proof of Ineq. 2:** By Proposition 9 (Robust Bellman Equation) and eq. 6, we have that

$$
\underline{Q}_{i,h}^{k,\rho_i}(s,\boldsymbol{a}) - Q_{i,h}^{\pi^k,\rho_i}(s,\boldsymbol{a})
$$

$$
= \max\left\{\sigma_{\widehat{\mathcal{P}_{i,h}^{\rho_i}}(s,\boldsymbol{a})}\left[\underline{V}_{i,h+1}^{k,\rho_i}\right] - \sigma_{\mathcal{P}_{i,h}^{\rho_i}(s,\boldsymbol{a})}\left[V_{i,h+1}^{\pi^k,\rho_i}\right] - \beta_{i,h}^k(s,\mathbf{a}), 0 - Q_{i,h}^{\dagger,\pi_{-i}^k,\rho_i}(s,\boldsymbol{a})\right\},
$$

$$
\leq \max\left\{\sigma_{\widehat{\mathcal{P}_{i,h}^{\rho_i}}(s,\boldsymbol{a})}\left[V_{i,h+1}^{\pi^k,\rho_i}\right] - \sigma_{\mathcal{P}_{i,h}^{\rho_i}(s,\boldsymbol{a})}\left[V_{i,h+1}^{\pi^k,\rho_i}\right] - \beta_{i,h}^k(s,\mathbf{a}), 0\right\}, \tag{76}
$$

where the second inequality follows from the induction of $V_{i,h+1}^{\pi^k,\rho_i} \geq \underline{V}_{i,h+1}^{k,\rho_i}$ at the $h+1$-th step and the fact that $Q_{i,h}^{\pi^k,\rho_i} \geq 0$. By Lemma 23, we can confirm that

$$
\sigma_{\widehat{\mathcal{P}_{i,h}^{\rho_i}}(s,\boldsymbol{a})}\left[V_{i,h+1}^{\pi^k,\rho_i}\right] - \sigma_{\mathcal{P}_{i,h}^{\rho_i}(s,\boldsymbol{a})}\left[V_{i,h+1}^{\pi^k,\rho_i}\right] \leq \sqrt{\frac{c_1\mathrm{Var}_{\widehat{P}_h^k}\left(V_{i,h+1}^{\dagger,\pi_{-i}^k,\rho_i}\right)\cdot\iota}{\{N_h^k(s,\mathbf{a}) \vee 1\}}}
$$

$$
+ \frac{\mathbb{E}_{\widehat{P}_h^k(\cdot|s,\mathbf{a})}\left[\overline{V}_{i,h+1}^{k,\rho_i} - \underline{V}_{i,h+1}^{k,\rho_i})\right]}{H}
$$

$$
+ \frac{c_2' H^2 S\iota}{\{N_h^k(s,\mathbf{a}) \vee 1\}} + \frac{1}{\sqrt{K}}. \tag{77}
$$

Now by further applying Lemma 26 to the variance term in the above inequality, with an argument similar to eq. 74 we can obtain that

$$
\sigma_{\widehat{\mathcal{P}}_{i,h}^{\rho_i}(s,\boldsymbol{a})}\left[V_{i,h+1}^{\pi^k,\rho_i}\right] - \sigma_{\mathcal{P}_{i,h}^{\rho_i}(s,\boldsymbol{a})}\left[V_{i,h+1}^{\pi^k,\rho_i}\right] \leq \sqrt{\frac{c_1 \mathrm{Var}_{\widehat{P}_h^k}\left(V_{i,h+1}^{\dagger,\pi_{-i}^k,\rho_i}\right) \cdot \iota}{\{N_h^k(s,\boldsymbol{a}) \vee 1\}}}
$$
$$
+ \frac{\mathbb{E}_{\widehat{P}_h^k(\cdot|s,\boldsymbol{a})}\left[\overline{V}_{i,h+1}^{k,\rho_i} - \underline{V}_{i,h+1}^{k,\rho_i})\right]}{H}
$$
$$
+ \frac{c_2'' H^2 S \iota}{\{N_h^k(s,\boldsymbol{a}) \vee 1\}} + \frac{1}{\sqrt{K}}, \qquad (78)
$$

where $c_2'' > 0$ is an absolute constant. Therefore, combining eqns. 76, 77, 78, and the choice of bonus in 20, $\underline{Q}_{i,h}^{k,\rho_i}(s,\boldsymbol{a}) - Q_{i,h}^{\pi^k,\rho_i}(s,\boldsymbol{a}) \leq 0$.

Therefore, by eq. 75 and eq. 78, we have proved that at step $h$, it holds that

$$
Q_{i,h}^{\dagger,\pi_{-i}^k,\rho_i}(s,\boldsymbol{a}) \leq \overline{Q}_{i,h}^{k,\rho_i}(s,\boldsymbol{a}), \quad \underline{Q}_{i,h}^{k,\rho_i}(s,\boldsymbol{a}) \leq Q_{i,h}^{\pi^k,\rho_i}(s,\boldsymbol{a}). \qquad (79)
$$

We now assume that eq. 71 hold for $h$-th step. Then, by the definition of robust value function as given by robust Bellman equation (Proposition 9), eq. 8, and CE Equilibrium, we get

$$
\overline{V}_{i,h}^{k,\rho_i}(s) = \mathbb{E}_{\boldsymbol{a} \sim \pi^k(\cdot|s)}\left[\overline{Q}_{i,h}^{k,\rho_i}(s,\boldsymbol{a})\right] = \max_{\phi \in \Phi_i} \mathbb{E}_{\boldsymbol{a} \sim \phi \diamond \pi^k(\cdot|s)}\left[\overline{Q}_{i,h}^{k,\rho_i}(s,\boldsymbol{a})\right]. \qquad (80)
$$

By the definition of $\max_{\phi \in \Phi_i} V_{i,h}^{\phi \diamond \pi^k,\rho_i}(s)$ in eq. 3, we get

$$
\max_{\phi \in \Phi_i} V_{i,h}^{\phi \diamond \pi^k,\rho_i}(s) = \max_{\phi \in \Phi_i} \mathbb{E}_{\boldsymbol{a} \sim \phi \diamond \pi^k(\cdot|s)}\left[\max_{\phi'} Q_{i,h}^{\phi' \diamond \pi^k,\rho_i}(s,\boldsymbol{a})\right]. \qquad (81)
$$

Since by induction, for any $(s,\boldsymbol{a})$, $\overline{Q}_{i,h}^{k,\rho_i}(s,\boldsymbol{a}) \geq \max_{\phi \in \Phi_i} Q_{i,h}^{\phi \diamond \pi^k,\rho_i}(s,\boldsymbol{a})$. As a result, we also have $\overline{V}_{i,h}^{k,\rho_i}(s) \geq \max_{\phi \in \Phi_i} V_{i,h}^{\phi \diamond \pi^k,\rho_i}(s)$, which is eq. 162 for $h$-th step. Similarly, we can show that

$$
\underline{V}_{i,h}^{k,\rho_i}(s) = \mathbb{E}_{\boldsymbol{a} \sim \pi^k(\cdot|s)}\left[\underline{Q}_{i,h}^{k,\rho_i}(s,\boldsymbol{a})\right],
$$
$$
\overset{(i)}{\leq} \mathbb{E}_{\boldsymbol{a} \sim \pi^k(\cdot|s)}\left[Q_{i,h}^{\pi^k,\rho_i}(s,\boldsymbol{a})\right],
$$
$$
\overset{(ii)}{=} V_{i,h}^{\pi^k,\rho_i}(s), \qquad (82)
$$

where (i) is due to the fact that $\underline{Q}_{i,h}^{k,\rho_i}(s,\boldsymbol{a}) \leq Q_{i,h}^{\pi^k,\rho_i}(s,\boldsymbol{a})$ and (ii) is by definition of $V_{i,h}^{\pi^k,\rho_i}(s)$ as given by Bellman equation in Proposition 9. $\qquad \square$

### F.3 AUXILIARY LEMMAS FOR TV-DRMG

**Lemma 23** (Bernstein bound for TV-DRMG and the robust value functions of $\pi^k$ and $\pi^\dagger$). *Under event $\mathcal{E}_{TV}$ in eq. 21 and definition of $\pi^\dagger$ as given in eq. 19, we assume that for any* EQUILIBRIUM $\in$ {*NASH, CE, CCE*} *the optimism and pessimism inequalities holds at $(h+1, k)$, where these inequalities can correspond to any of the following cases of* EQUILIBRIUM:

- **NE:** *Lemma 20 using eq. 47 and eq. 48,*

- **CCE:** *Lemma 21 using eq. 59 and eq. 60,*

- **CE:** *Lemma 22 using eq. 71 and eq. 72,*

*Then, it holds that*

$$\left| \sigma_{\widehat{\mathcal{P}_{i,h}^{\rho_i}}(s,\boldsymbol{a})}[V_{i,h+1}^{\pi^k,\rho_i}] - \sigma_{\mathcal{P}_{i,h}^{\rho_i}(s,\boldsymbol{a})}[V_{i,h+1}^{\pi^k,\rho_i}] \right|$$

$$\leq \begin{cases} \sqrt{\dfrac{c_1 \mathrm{Var}_{\widehat{P}_h^k}\left(V_{i,h+1}^{\dagger,\pi_{-i}^k,\rho_i}\right) \cdot \iota}{\{N_h^k(s,\boldsymbol{a})\vee 1\}}} + \dfrac{c_2 H \iota}{\{N_h^k(s,\boldsymbol{a})\vee 1\}} + \dfrac{1}{\sqrt{K}}, & \text{if } \pi^k = \pi^\dagger \\[4ex] \sqrt{\dfrac{c_1 \mathrm{Var}_{\widehat{P}_h^k}\left(V_{i,h+1}^{\dagger,\pi_{-i}^k,\rho_i}\right) \cdot \iota}{\{N_h^k(s,\boldsymbol{a})\vee 1\}}} + \dfrac{\mathbb{E}_{\widehat{P}_h^k(\cdot|s,\boldsymbol{a})}\left[\overline{V}_{i,h+1}^{k,\rho_i} - \underline{V}_{i,h+1}^{k,\rho_i}\right]}{H} + \dfrac{c_2' H^2 S \iota}{\{N_h^k(s,\boldsymbol{a})\vee 1\}} + \dfrac{1}{\sqrt{K}}, & \text{otherwise,} \end{cases}$$

*where* $\iota = \log\left(\dfrac{S^2(\prod_{i=1}^m A_i)H^2 K^{3/2}}{\delta}\right)$ *and* $c_1, c_2' > 0$ *are absolute constants.*

*Proof.* By our definition of the operator $\sigma_{\mathcal{P}_{i,h}^{\rho_i}(s,\boldsymbol{a})}[V_{i,h+1}^{\pi^k,\rho_i}]$ in eq. 11, we can arrive at,

$$\left| \sigma_{\widehat{\mathcal{P}_{i,h}^{\rho_i}}(s,\boldsymbol{a})}[V_{i,h+1}^{\pi^k,\rho_i}] - \sigma_{\mathcal{P}_{i,h}^{\rho_i}(s,\boldsymbol{a})}[V_{i,h+1}^{\pi^k,\rho_i}] \right| \leq \sup_{\eta \in [0,H]} \left| \left\{ \mathbb{E}_{\widehat{P}_h^k(\cdot|s,\boldsymbol{a})}\left[(\eta - V_{i,h+1}^{\pi^k,\rho_i})_+\right] \right. \right.$$

$$\left. \left. - \mathbb{E}_{P_h^\star(\cdot|s,\boldsymbol{a})}\left[(\eta - V_{i,h+1}^{\pi^k,\rho_i})_+\right] \right\} \right|$$

$$= \text{Term (i) + Term (ii).} \tag{83}$$

where we denote

$$\text{Term (i)} := \sup_{\eta \in [0,H]} \left| \left\{ \mathbb{E}_{\widehat{P}_h^k(\cdot|s,\boldsymbol{a})}\left[(\eta - V_{i,h+1}^{\dagger,\pi_{-i}^k,\rho_i})_+\right] - \mathbb{E}_{P_h^\star(\cdot|s,\boldsymbol{a})}\left[(\eta - V_{i,h+1}^{\dagger,\pi_{-i}^k,\rho_i})_+\right] \right\} \right| \tag{84}$$

$$\text{Term (ii)} := \sup_{\eta \in [0,H]} \left| \left\{ \mathbb{E}_{\widehat{P}_h^k(\cdot|s,\boldsymbol{a})}\left[\left(\eta - V_{i,h+1}^{\pi^k,\rho_i}\right]\right)_+ - \left(\eta - V_{i,h+1}^{\dagger,\pi_{-1i}^k,\rho_i}\right]\right)_+\right] \right.$$

$$\left. - \mathbb{E}_{P_h^\star(\cdot|s,\boldsymbol{a})}\left[\left(\eta - V_{i,h+1}^{\pi^k,\rho_i}\right]\right)_+ - \left(\eta - V_{i,h+1}^{\dagger,\pi_{-i}^k,\rho_i}\right]\right)_+\right] \right\} \right|. \tag{85}$$

We deal with Term (i) and Term (ii) respectively.

**Bound for Term (i):** Term (i) is referred to Bernstein bound for Bernstein bound for TV-DRMG and the robust value function of the robust best response $\pi_i^{\dagger,\rho_i}(\pi_{-i})$. More specifically, we find the Bernstein bound on the gap $\left| \sigma_{\widehat{\mathcal{P}_{i,h}^{\rho_i}}(s,\boldsymbol{a})}[V_{i,h+1}^{\dagger,\pi_{-i}^k,\rho_i}] - \sigma_{\mathcal{P}_{i,h}^{\rho_i}(s,\boldsymbol{a})}[V_{i,h+1}^{\dagger,\pi_{-i}^k,\rho_i}] \right|$. Therefore, by the definition of the operator $\sigma_{\mathcal{P}_{i,h}^{\rho_i}(s,\boldsymbol{a})}[V_{i,h+1}^{\dagger,\pi_{-i}^k,\rho_i}]$ in eq. 11), we can arrive at,

$$\left| \sigma_{\widehat{\mathcal{P}_{i,h}^{\rho_i}}(s,\boldsymbol{a})}[V_{i,h+1}^{\dagger,\pi_{-i}^k,\rho_i}] - \sigma_{\mathcal{P}_{i,h}^{\rho_i}(s,\boldsymbol{a})}[V_{i,h+1}^{\dagger,\pi_{-i}^k,\rho_i}] \right|$$

$$\leq \sup_{\eta \in [0,H]} \left| \left\{ \mathbb{E}_{\widehat{P}_h^k(\cdot|s,\boldsymbol{a})}\left[\left(\eta - V_{i,h+1}^{\dagger,\pi_{-1i}^k,\rho_i}\right)_+\right] - \mathbb{E}_{P_h^\star(\cdot|s,\boldsymbol{a})}\left[\left(\eta - V_{i,h+1}^{\dagger,\pi_{-i}^k,\rho_i}\right)_+\right] \right\} \right|$$

$$= \text{Term (i).} \tag{86}$$

By now according to the first inequality of event $\mathcal{E}$ in eq. 21, we can bound eq. 86 as

$$\text{Term (i)} \leq \sqrt{\frac{c_1 \mathrm{Var}_{\widehat{P}_h^k}\left(\eta - V_{i,h+1}^{\dagger,\pi_{-i}^k,\rho_i}\right)_+ \cdot \iota}{\{N_h^k(s,\boldsymbol{a})\vee 1\}}} + \frac{c_2 H \iota}{\{N_h^k(s,\boldsymbol{a})\vee 1\}}$$

$$\leq \sqrt{\frac{c_1 \mathrm{Var}_{\widehat{P}_h^k}\left(V_{i,h+1}^{\dagger,\pi_{-i}^k,\rho_i}\right) \cdot \iota}{\{N_h^k(s,\boldsymbol{a})\vee 1\}}} + \frac{c_2 H \iota}{\{N_h^k(s,\boldsymbol{a})\vee 1\}}, \tag{87}$$

for any $\eta \in \mathcal{N}_{1/(S\sqrt{K})}([0, H])$. Here the second inequality is because $\mathrm{Var}[(a - X)_+] \leq \mathrm{Var}[X]$. Therefore, by applying the covering argument in eq. 87, for any $\eta \in [0, H]$, it holds that

$$\text{Term (i)} \leq \sqrt{\frac{c_1 \mathrm{Var}_{\widehat{P}_h^k}\left(V_{i,h+1}^{\dagger, \pi_{-i}^k, \rho_i}\right) \cdot \iota}{\{N_h^k(s, \mathbf{a}) \vee 1\}}} + \frac{c_2 H \iota}{\{N_h^k(s, \mathbf{a}) \vee 1\}} + \frac{1}{\sqrt{K}}. \tag{88}$$

**Bound for Term (ii):** For Term (ii), we apply the second inequality of event $\mathcal{E}$ in eq. 21, and we obtain that

$$\text{Term (ii)} \leq \sup_{\eta \in [0, H]} \left\{ \sum_{s' \in \mathcal{S}} \left( \sqrt{\frac{c_1 \min\left\{P_h^\star(s' \mid s, \mathbf{a}), P_h^k(s' \mid s, \mathbf{a})\right\} \cdot \iota}{\{N_h^k(s, \mathbf{a}) \vee 1\}}} + \frac{c_2 \iota}{\{N_h^k(s, \mathbf{a}) \vee 1\}} \right) \right.$$
$$\left. \times \left| \left(\eta - V_{i,h+1}^{\pi^k, \rho_i}]\right)_+ - \left(\eta - V_{i,h+1}^{\dagger, \pi_{-i}^k, \rho_i}]\right)_+ \right| \right\}. \tag{89}$$

Now by assuming that eq. 48 holds at $(h + 1, k)$, we can upper bound the absolute value above by

$$\left| \left(\eta - V_{i,h+1}^{\pi^k, \rho_i}]\right)_+ - \left(\eta - V_{i,h+1}^{\dagger, \pi_{-i}^k, \rho_i}]\right)_+ \right| \overset{(i)}{\leq} \left| V_{i,h+1}^{\pi^k, \rho_i} - V_{i,h+1}^{\dagger, \pi_{-i}^k, \rho_i} \right|$$

$$\overset{(ii)}{\leq} \overline{V}_{i,h+1}^{k, \rho_i}(s') - \underline{V}_{i,h+1}^{k, \rho_i}(s'), \tag{90}$$

where the first inequality (i) is due to the 1-Lipschitz continuity of $\psi_\eta(x) = (\eta - x)_+$, and the second inequality (ii) is due to eq. 48. Thus combining eq. 89 and eq. 90, we get

$$\text{Term (ii)} \leq \sum_{s' \in \mathcal{S}} \left( \sqrt{\frac{c_1 \widehat{P}_h^k(s' \mid s, \mathbf{a}) \cdot \iota}{\{N_h^k(s, \mathbf{a}) \vee 1\}}} + \frac{c_2 \iota}{\{N_h^k(s, \mathbf{a}) \vee 1\}} \right) \cdot \left( \overline{V}_{i,h+1}^{k, \rho_i}(s') - \underline{V}_{i,h+1}^{k, \rho_i}(s') \right)$$

$$\overset{(i)}{\leq} \sum_{s' \in \mathcal{S}} \left( \frac{\widehat{P}_h^k(s' \mid s, \mathbf{a})}{H} + \frac{c_1 H \iota}{\{N_h^k(s, \mathbf{a}) \vee 1\}} + \frac{c_2 \iota}{\{N_h^k(s, \mathbf{a}) \vee 1\}} \right)$$
$$\cdot \left( \overline{V}_{i,h+1}^{k, \rho_i}(s') - \underline{V}_{i,h+1}^{k, \rho_i}(s') \right)$$

$$\overset{(ii)}{\leq} \frac{\mathbb{E}_{\widehat{P}_h^k(\cdot \mid s, \mathbf{a})}\left[\overline{V}_{i,h+1}^{k, \rho_i} - \underline{V}_{i,h+1}^{k, \rho_i}\right]}{H} + \frac{c_2' H^2 S \iota}{\{N_h^k(s, \mathbf{a}) \vee 1\}}, \tag{91}$$

where $c_2' > 0$ is an absolute constant. The first inequality (i) is by $\sqrt{ab} \leq a + b$ and the second inequality (ii) is due to $\overline{V}_{i,h+1}^{k, \rho_i}, \underline{V}_{i,h+1}^{k, \rho_i} \in [0, H]$. Finally, by combining eq. 88 and eq. 91 and applying in eq. 83, we get the required bound as

$$\text{Term (ii)} \leq \sqrt{\frac{c_1 \mathrm{Var}_{\widehat{P}_h^k}\left(V_{i,h+1}^{\dagger, \pi_{-i}^k, \rho_i}\right) \cdot \iota}{\{N_h^k(s, \mathbf{a}) \vee 1\}}} + \frac{\mathbb{E}_{\widehat{P}_h^k(\cdot \mid s, \mathbf{a})}\left[\overline{V}_{i,h+1}^{k, \rho_i} - \underline{V}_{i,h+1}^{k, \rho_i}\right]}{H} + \frac{c_2' H^2 S \iota}{\{N_h^k(s, \mathbf{a}) \vee 1\}}$$
$$+ \frac{1}{\sqrt{K}}. \tag{92}$$

This concludes the proof of Lemma 23. $\qquad\qquad\qquad\qquad\qquad\qquad\qquad\qquad\qquad\qquad\qquad\qquad\quad \square$

**Lemma 24** (Bernstein bound for TV-DRMG and optimistic and pessimistic robust value estimators). *Under event $\mathcal{E}_{TV}$ in eq. 21 and definition of $\pi^\dagger$ as given in eq. 19, we assume that for any* EQUILIBRIUM $\in \{NASH, CE, CCE\}$ *the optimism and pessimism inequalities holds at $(h + 1, k)$, where these inequalities can correspond to any of the following cases of* EQUILIBRIUM:

- **NE:** *Lemma 20 using eq. 47 and eq. 48,*

- **CCE:** *Lemma 21 using eq. 59 and eq. 60,*

- **CE:** *Lemma 22 using eq. 71 and eq. 72,*

*Then, it holds that*

$$
\max\left\{\left|\sigma_{\widehat{\mathcal{P}_{i,h}^{\rho_i}}(s,\boldsymbol{a})}\left[\overline{V}_{i,h+1}^{k,\rho_i}\right] - \sigma_{\mathcal{P}_{i,h}^{\rho_i}(s,\boldsymbol{a})}\left[\overline{V}_{i,h+1}^{k,\rho_i}\right]\right|, \left|\sigma_{\widehat{\mathcal{P}_{i,h}^{\rho_i}}(s,\boldsymbol{a})}\left[\underline{V}_{i,h+1}^{k,\rho_i}\right] - \sigma_{\mathcal{P}_{i,h}^{\rho_i}(s,\boldsymbol{a})}\left[\underline{V}_{i,h+1}^{k,\rho_i}\right]\right|\right\}
$$

$$
\leq \sqrt{\frac{c_1 \mathrm{Var}_{\widehat{P}_h^k}\left(V_{i,h+1}^{\dagger,\pi_{-i}^k,\rho_i}\right)\cdot\iota}{\{N_h^k(s,\mathbf{a})\vee 1\}}} + \frac{\mathbb{E}_{\widehat{P}_h^k(\cdot|s,\mathbf{a})}\left[\overline{V}_{i,h+1}^{k,\rho_i} - \underline{V}_{i,h+1}^{k,\rho_i}\right]}{H} + \frac{c_2' H^2 S\iota}{\{N_h^k(s,\mathbf{a})\vee 1\}} + \frac{1}{\sqrt{K}},
$$

*where* $\iota = \log\left(\frac{S^2(\prod_{i=1}^m A_i)H^2 K^{3/2}}{\delta}\right)$ *and* $c_1, c_2' > 0$ *are absolute constants.*

*Proof.* This follows from the same proof as Lemma 23 and is thus omitted. □

**Lemma 25** (Non-robust Concentration for TV-DRMG). *Under event* $\mathcal{E}_{TV}$ *in eq. 21 and definition of* $\pi^\dagger$ *as given in eq. 19, we assume that for any* EQUILIBRIUM $\in$ {*NASH, CE, CCE*} *the optimism and pessimism inequalities holds at* $(h+1, k)$, *where these inequalities can correspond to any of the following cases of* EQUILIBRIUM*:*

- **NE:** *Lemma 20 using eq. 47 and eq. 48,*

- **CCE:** *Lemma 21 using eq. 59 and eq. 60,*

- **CE:** *Lemma 22 using eq. 71 and eq. 72,*

*Then, it holds that*

$$
\left|\mathbb{E}_{P_h^\star(\cdot|s,\mathbf{a})}[\overline{V}_{i,h+1}^{k,\rho_i} - \underline{V}_{i,h+1}^{k,\rho_i}] - \mathbb{E}_{\widehat{P}_h^k(\cdot|s,\mathbf{a})}[\overline{V}_{i,h+1}^{k,\rho_i} - \underline{V}_{i,h+1}^{k,\rho_i}]\right| \leq \frac{\mathbb{E}_{\widehat{P}_h^k(\cdot|s,\mathbf{a})}\left[\overline{V}_{i,h+1}^{k,\rho_i} - \underline{V}_{i,h+1}^{k,\rho_i}\right]}{H}
$$

$$
+ \frac{c_2' H^2 S\iota}{\{N_h^k(s,\mathbf{a})\vee 1\}},
$$

*where* $\iota = \log\left(\frac{S^2(\prod_{i=1}^m A_i)H^2 K^{3/2}}{\delta}\right)$ *and* $c_2' > 0$ *are absolute constants.*

*Proof.* Assuming that eq. 48 holds for $(h+1, k)$, we apply the second inequality of event $\mathcal{E}$ in eq. 21 to get the required bound Lemma 25. □

**Lemma 26** (Variance analysis for $\pi^\dagger$ for TV-DRMG). *Under the definition of* $\pi^\dagger$ *as given in eq. 19, we assume that for any* EQUILIBRIUM $\in$ {*NASH, CE, CCE*} *the optimism and pessimism inequalities holds at* $(h+1, k)$, *where these inequalities can correspond to any of the following cases of* EQUILIBRIUM*:*

- **NE:** *Lemma 20 using eq. 47 and eq. 48,*

- **CCE:** *Lemma 21 using eq. 59 and eq. 60,*

- **CE:** *Lemma 22 using eq. 71 and eq. 72,*

*Then, it holds that*

$$
\left|\mathrm{Var}_{\widehat{P}_h^k(\cdot|s,\mathbf{a})}\left[\frac{\overline{V}_{i,h+1}^{k,\rho_i} + \underline{V}_{i,h+1}^{k,\rho_i}}{2}\right] - \mathrm{Var}_{\widehat{P}_h^k(\cdot|s,\mathbf{a})}\left[V_{i,h+1}^{\dagger,\pi_{-i}^k,\rho_i}\right]\right| \leq 4H\,\mathbb{E}_{\widehat{P}_h^k(\cdot|s,\mathbf{a})}\left[\overline{V}_{h+1}^{k,\rho_i} - \underline{V}_{h+1}^{k,\rho_i}\right].
$$

*Proof.* Our proof closely follows the lines of Lemma 22 in (Liu et al., 2021) and Lemma E.11 in (Lu et al., 2024), with detailed elaboration on each step for clarity. The left hand side of the inequality in Lemma 26 can be upper bounded by the following

$$
\left| \mathrm{Var}_{\widehat{P}_h^k(\cdot|s,\mathbf{a})} \left[ \left( \frac{\overline{V}_{i,h+1}^{k,\rho_i} + \underline{V}_{i,h+1}^{k,\rho_i}}{2} \right) \right] - \mathrm{Var}_{\widehat{P}_h^k(\cdot|s,\mathbf{a})} \left[ V_{i,h+1}^{\dagger,\pi_{-i}^k,\rho_i} \right] \right|
$$

$$
\leq \left| \mathbb{E}_{\widehat{P}_h^k(\cdot|s,\mathbf{a})} \left[ \left( \frac{\overline{V}_{i,h+1}^{k,\rho_i} + \underline{V}_{i,h+1}^{k,\rho_i}}{2} \right)^2 \right] - \mathbb{E}_{\widehat{P}_h^k(\cdot|s,\mathbf{a})} \left[ \left( V_{i,h+1}^{\dagger,\pi_{-i}^k,\rho_i} \right)^2 \right] \right|
$$

$$
+ \left| \left( \mathbb{E}_{\widehat{P}_h^k(\cdot|s,\mathbf{a})} \left[ \left( \frac{\overline{V}_{i,h+1}^{k,\rho_i} + \underline{V}_{i,h+1}^{k,\rho_i}}{2} \right) \right] \right)^2 - \left( \mathbb{E}_{\widehat{P}_h^k(\cdot|s,\mathbf{a})} \left[ V_{i,h+1}^{\dagger,\pi_{-i}^k,\rho_i} \right] \right)^2 \right|. \quad (93)
$$

By applying eq. 48 and the facts that $\overline{V}_{i,h+1}^{k,\rho_i}$ and $\underline{V}_{i,h+1}^{k,\rho_i}, \overline{V}_{i,h+1}^{k,\rho_i}, \underline{V}_{i,h+1}^{k,\rho_i}, V_{i,h+1}^{\dagger,\pi_{-i}^k,\rho_i} \in [0,H]$, we can further upper bound eq. 93 as

$$
\left| \mathrm{Var}_{\widehat{P}_h^k(\cdot|s,\mathbf{a})} \left[ \left( \frac{\overline{V}_{i,h+1}^{k,\rho_i} + \underline{V}_{i,h+1}^{k,\rho_i}}{2} \right) \right] - \mathrm{Var}_{\widehat{P}_h^k(\cdot|s,\mathbf{a})} \left[ V_{i,h+1}^{\dagger,\pi_{-i}^k,\rho_i} \right] \right|
$$

$$
\leq 4H \, \mathbb{E}_{\widehat{P}_h^k(\cdot|s,\mathbf{a})} \left[ \left| \frac{\overline{V}_{i,h+1}^{k,\rho_i} + \underline{V}_{i,h+1}^{k,\rho_i}}{2} - V_{i,h+1}^{\dagger,\pi_{-i}^k,\rho_i} \right| \right] \leq 4H \, \mathbb{E}_{\widehat{P}_h^k(\cdot|s,\mathbf{a})} \left[ \overline{V}_{i,h+1}^{k,\rho_i} - \underline{V}_{i,h+1}^{k,\rho_i} \right]. \quad (94)
$$

This concludes the proof of Lemma 26. □

**Lemma 27** (Variance analysis for any robust joint policy $\pi^k$ for TV-DRMG). *Under event $\mathcal{E}_{TV}$ in eq. 21 and definition of $\pi^\dagger$ as given in eq. 19, we assume that for any EQUILIBRIUM $\in$ {NASH, CE, CCE} the optimism and pessimism inequalities holds at $(h+1,k)$, where these inequalities can correspond to any of the following cases of EQUILIBRIUM:*

- *NE: Lemma 20 using eq. 47 and eq. 48,*

- *CCE: Lemma 21 using eq. 59 and eq. 60,*

- *CE: Lemma 22 using eq. 71 and eq. 72,*

*Then, then the following inequality holds,*

$$
\left| \mathrm{Var}_{\widehat{P}_h^k(\cdot|s,\mathbf{a})} \left[ \left( \frac{\overline{V}_{i,h+1}^{k,\rho_i} + \underline{V}_{i,h+1}^{k,\rho_i}}{2} \right) \right] - \mathrm{Var}_{P_h^\star(\cdot|s,\mathbf{a})} \left[ V_{i,h+1}^{\pi^k,\rho_i} \right] \right|
$$

$$
\leq 4H \mathbb{E}_{P_h^\star(\cdot|s,\mathbf{a})} \left[ \overline{V}_{h+1}^{k,\rho_i} - \underline{V}_{h+1}^{k,\rho_i} \right] + \frac{c_2' H^4 S \iota}{\{N_h^k(s,\mathbf{a}) \vee 1\}} + 1.
$$

*Proof.* We follow the proof-lines of Lemma 23 in (Liu et al., 2021) and Lemma E.12 of (Lu et al., 2024). We present a detailed derivation as follows. We first relate the variance on $\widehat{P}_h^k$ to the variance on $P_h^\star$. Specifically, we have

$$
\left| \mathrm{Var}_{\widehat{P}_h^k(\cdot|s,\mathbf{a})} \left[ \left( \frac{\overline{V}_{i,h+1}^{k,\rho_i} + \underline{V}_{i,h+1}^{k,\rho_i}}{2} \right) \right] - \mathrm{Var}_{P_h^\star(\cdot|s,\mathbf{a})} \left[ V_{i,h+1}^{\pi^k,\rho_i} \right] \right| \leq \text{Term (i)} + \text{Term (ii)}, \quad (95)
$$

where we denote

$$
\text{Term (i)} := \left| \mathrm{Var}_{\widehat{P}_h^k(\cdot|s,\mathbf{a})} \left[ \frac{\overline{V}_{i,h+1}^{k,\rho_i} + \underline{V}_{i,h+1}^{k,\rho_i}}{2} \right] - \mathrm{Var}_{P_h^\star(\cdot|s,\mathbf{a})} \left[ \frac{\overline{V}_{i,h+1}^{k,\rho_i} + \underline{V}_{i,h+1}^{k,\rho_i}}{2} \right] \right|. \quad (96)
$$

$$
\text{Term (ii)} := \left| \mathrm{Var}_{P_h^\star(\cdot|s,\mathbf{a})} \left[ \left( \frac{\overline{V}_{i,h+1}^{k,\rho_i} + \underline{V}_{i,h+1}^{k,\rho_i}}{2} \right) \right] - \mathrm{Var}_{\widehat{P}_h^k(\cdot|s,\mathbf{a})} \left[ V_{i,h+1}^{\pi^k,\rho_i} \right] \right|. \quad (97)
$$

We will now bound Term (i) and Term (ii) respectively.

- **Term (i):** By applying the fact $\left(\overline{V}_{i,h+1}^{k,\rho_i} + \underline{V}_{i,h+1}^{k,\rho_i}\right)\big/2 \in [0, H]$ in the variance terms on Term (i), we can upper bound Term (i) as

$$\text{Term (i)} \leq H^2 \sum_{s' \in \mathcal{S}} \left| P_h^\star(s'|s, \mathbf{a}) - \widehat{P}_h^k(s'|s, \mathbf{a}) \right|$$

$$\overset{(i)}{\leq} H^2 \sum_{s' \in \mathcal{S}} \left( \sqrt{\frac{c_1 \widehat{P}_h^k(s' \mid s, \mathbf{a}) \cdot \iota}{\{N_h^k(s, \mathbf{a}) \vee 1\}}} + \frac{c_2 \iota}{\{N_h^k(s, \mathbf{a}) \vee 1\}} \right)$$

$$\overset{(ii)}{\leq} H^2 \left( \sqrt{\frac{c_1 S \iota}{\{N_h^k(s, \mathbf{a}) \vee 1\}}} + \frac{c_2 S \iota}{\{N_h^k(s, \mathbf{a}) \vee 1\}} \right)$$

$$\overset{(iii)}{\leq} 1 + \frac{c_2' H^4 S \iota}{\{N_h^k(s, \mathbf{a}) \vee 1\}}, \tag{98}$$

where the inequality (i) is by the second inequality in event $\mathcal{E}$ in eq. 21, the inequality (ii) is by Cauchy- Schwartz inequality and the probability distribution sums up to 1, and the last inequality (iii) is from the fact $\sqrt{ab} \leq a + b$.

- **Term (ii):** By using the proof-lines of Lemma 26 and assuming that the optimism and pessimism inequality eq. 48 holds for $(h + 1, k)$, we can bound Term (ii) as

$$\text{Term (ii)} \leq 4H \mathbb{E}_{P_h^\star(\cdot|s,\mathbf{a})} \left[ \overline{V}_{h+1}^{k,\rho_i} - \underline{V}_{h+1}^{k,\rho_i} \right]. \tag{99}$$

Applying eq. 98 and eq. 99, we get the required bound in Lemma 27. $\qquad\square$

## G   PROOF OF REGRET BOUND OF KL-MORNAVI

Similar to (Ghosh et al., 2025), we consider the following definitions:

$$\widehat{P}_{\min,h}^k(s, \boldsymbol{a}) := \min_{s' \in \mathcal{S}} \left\{ \widehat{P}_h^k(s'|s, \boldsymbol{a}) : \widehat{P}_h^k(s'|s, \boldsymbol{a}) > 0 \right\}, \tag{100}$$

$$P_{\min,h}^\star(s, \boldsymbol{a}) := \min_{s' \in \mathcal{S}} \left\{ P_h^\star(s'|s, \boldsymbol{a}) : P_h^\star(s'|s, \boldsymbol{a}) > 0 \right\}, \tag{101}$$

$$P_{\min}^\star := \min_{(h,s) \in [H] \times \mathcal{S}} P_{\min,h}^\star(s, \pi_h^\star(s)), \tag{102}$$

where the following inequality is satisfied: $P_h^\star(s'|s, \boldsymbol{a}) \geq P_{\min,h}^\star(s, \pi_h^\star(s)) \geq P_{\min}^\star$.

We now recall the bonus term of KL-MORNAVI for agent $i$ in episode $k$ at step $h$, as follows:

$$\beta_{i,h}^k(s, \boldsymbol{a}) = \frac{2c_f H}{\sigma_i} \sqrt{\frac{\iota}{\left(N_h^k(s, \boldsymbol{a}) \vee 1\right) \widehat{P}_{\min,h}^k(s, \boldsymbol{a})}} + \sqrt{\frac{1}{K}}, \tag{103}$$

where $\widehat{P}_{\min,h}^k(s, \boldsymbol{a}) = \min_{s' \in \mathcal{S}} \{ \widehat{P}_h^k(s'|s, \boldsymbol{a}) : \widehat{P}_h^k(s'|s, \boldsymbol{a}) > 0 \}$, $\iota = \log \left( S^2 (\prod_{i=1}^m A_i) H^2 K^{3/2}/\delta \right)$, and $c_f$ is an absolute constant.

Before proceeding to all key lemmas, we introduce the high-probability "typical" event $\mathcal{E}_{\text{KL}}$ in the lemma below. The proof strategy follows (Lu et al., 2024) and (Ghosh et al., 2025).

**Lemma 28** (Uniform Concentration Bound of event $\mathcal{E}_{\text{KL}}$). *Let $\mathcal{E}_{\text{KL}}$ be the event in which, for all $(s, \mathbf{a}, s', h, k) \in \mathcal{S} \times \mathcal{A} \times \mathcal{S} \times [H] \times [K]$, and for all $\eta$ in a $\frac{1}{\rho_{\min} S \sqrt{K}}$-cover of $[0, H/\rho_{\min}]$, and is*

*defined as*

$$\mathcal{E}_{KL} = \left\{ \left| \log \left( \mathbb{E}_{\widehat{P}_h^k(\cdot|s,\boldsymbol{a})} \left[ \exp \left\{ -\frac{V_{h+1}}{\eta} \right\} \right] \right) - \log \left( \mathbb{E}_{P_h^\star(\cdot|s,\boldsymbol{a})} \left[ \exp \left\{ -\frac{V_{h+1}}{\eta} \right\} \right] \right) \right| \right.$$

$$\leq c_1 \sqrt{\frac{\iota}{\{N_h^k(s,\boldsymbol{a}) \vee 1\}\widehat{P}_{\min,h}^k(s,\boldsymbol{a})}},$$

$$\left. \forall (h,s,\boldsymbol{a},s',k) \in [H] \times \mathcal{S} \times \mathcal{A} \times \mathcal{S} \times [K], \forall \eta \in \mathcal{N}_{\frac{1}{\rho_{\min}S\sqrt{K}}} \left( \left[ 0, \frac{H}{\rho_{\min}} \right] \right) \right\}, \quad (104)$$

*where $\widehat{P}_{\min,h}^k(s,\boldsymbol{a})$ is defined in eq. 100, $\iota = \log\left(S^3\left(\prod_{i=1}^m A_i\right)H^2K^{3/2}/\delta\right)$, $c_1 > 0$ is an absolute constant and $\eta \in \mathcal{N}_{\frac{1}{\rho_{\min}S\sqrt{K}}}([0,H/\rho_{\min}])$, where $\rho_{\min} = \min_{i\in\mathcal{M}} \rho_i$ and $\mathcal{N}_{\frac{1}{\rho_{\min}S\sqrt{K}}}([0,H/\rho_{\min}])$ denotes an $1/(\rho_{\min}S\sqrt{K})$-cover of the interval $[0,H/\rho_{\min}]$.*

*Then, this event $\mathcal{E}_{KL}$ occurs with high probability, i.e., $\Pr(\mathcal{E}_{KL}) \geq 1 - \delta$.*

*Proof.* The proof follows standard techniques: we apply classical concentration inequalities followed by a union bound. Consider a fixed tuple $(s, \mathbf{a}, h)$ for a fixed episode $k$. Now we consider the following equivalent random process: (i) before the agents starts, the environment samples $\{s^{(1)}, s^{(2)}, \ldots, s^{(k-1)}\}$ independently from $P_h^\star(\cdot|s,\mathbf{a})$, where $s^{(i)} \in \mathcal{S}$ denotes the state sampled at episode $i$; (ii) during the interaction between the agents and the environment, the $i$-th time the state and joint actions $(s, \mathbf{a})$ tuple is visited at step $h$, the environment will make the agents transit to the next state $s^{(i)}$. Note that the randomness induced by this interaction procedure is exactly the same as the original one, which means the probability of any event in this context is the same as in the original problem. Therefore, it suffices to prove the target concentration inequality in this context.

Based on the above fact, we directly apply (Wang et al., 2024d, Lemma 16). To extend the bound uniformly, we apply a union bound over all tuples $(h,s,\mathbf{a},s',k,\eta) \in [H] \times \mathcal{S} \times \mathcal{A} \times \mathcal{S} \times [K] \times \mathcal{N}_{1/(\rho_{\min}S\sqrt{K})}([0,H/\rho_{\min}])$. Note that the $\eta$-cover for each agent $i$ lies in the interval $[0,H/\rho_i] \leq [0,H/\rho_{\min}]$ for all $i \in \mathcal{M}$, and this cover contains a valid $\frac{1}{\rho_i S\sqrt{K}}$-cover for each agent-specific interval $\left[0,\frac{H}{\rho_i}\right]$. Therefore, we define the common $\eta$-cover as $\eta \in \mathcal{N}_{\frac{1}{\rho_{\min}S\sqrt{K}}}\left(\left[0,\frac{H}{\rho_{\min}}\right]\right)$, where $\mathcal{N}_{\frac{1}{\rho_{\min}S\sqrt{K}}}\left(\left[0,\frac{H}{\rho_{\min}}\right]\right)$ denotes a $\frac{1}{\rho_{\min}S\sqrt{K}}$-cover of the interval $\left[0,\frac{H}{\rho_{\min}}\right]$. $\qquad\square$

PROOF OF THEOREM 5 (KL-DRMG SETTING)

*Proof.* With Lemma 32, we can establish an upper bound on the regret by considering the difference between our optimistic and pessimistic value functions:

$$\text{Regret}_{\text{NASH}}(K) = \sum_{k=1}^K \max_{i\in\mathcal{M}}(V_{i,1}^{\dagger,\pi_{-i}^k,\rho_i} - V_{i,1}^{\pi^k,\rho_i})(s_1^k) \leq \sum_{k=1}^K \max_{i\in\mathcal{M}}(\overline{V}_{i,1}^{k,\rho_i} - \underline{V}_{i,1}^{k,\rho_i})(s_1^k). \quad (105)$$

For the KL-divergence uncertainty set, we will refer to the bonus term as $\beta_{i,h}^k(s,\boldsymbol{a})$, as given in eq. 103. Our first step is to establish a bound on the difference between the upper and lower Q-values. Given our definitions for $\overline{Q}_{i,h}^{k,\rho_i}, \underline{Q}_{i,h}^{k,\rho_i}, \overline{V}_{i,h}^{k,\rho_i}, \underline{V}_{i,h}^{k,\rho_i}$, and the bonus term $\beta_{i,h}^{k,\rho_i}(s,\boldsymbol{a})$ as defined in eq. 5 through eq. 103, for any $(i,h,k,s,\boldsymbol{a}) \in \mathcal{M} \times [H] \times [K] \times \mathcal{S} \times \mathcal{A}$, we have

$$\overline{Q}_{i,h}^{k,\rho_i}(s,\boldsymbol{a}) - \underline{Q}_{i,h}^k(s,\boldsymbol{a}) \leq \sigma_{\widehat{\mathcal{P}}_{i,h}^{\rho_i}(s,\boldsymbol{a})}\left[\overline{V}_{i,h+1}^{k,\rho_i}\right] - \sigma_{\widehat{\mathcal{P}}_{i,h}^{\rho_i}(s,\boldsymbol{a})}\left[\underline{V}_{i,h+1}^{k,\rho_i}\right] + 2\beta_{i,h}^{k,\rho_i}(s,\boldsymbol{a}). \quad (106)$$

We define the following terms, $A$ and $B$, to simplify our analysis:

$$A := \sigma_{\widehat{\mathcal{P}}_{i,h}^{\rho_i}(s,\boldsymbol{a})}\left[\overline{V}_{i,h+1}^{k,\rho_i}\right] - \sigma_{\mathcal{P}_{i,h}^{\rho_i}(s,\boldsymbol{a})}\left[\overline{V}_{i,h+1}^{k,\rho_i}\right] + \sigma_{\mathcal{P}_{i,h}^{\rho_i}(s,\boldsymbol{a})}\left[\underline{V}_{i,h+1}^{k,\rho_i}\right] - \sigma_{\widehat{\mathcal{P}}_{i,h}^{\rho_i}(s,\boldsymbol{a})}\left[\underline{V}_{i,h+1}^{k,\rho_i}\right]. \quad (107)$$

$$B := \sigma_{\mathcal{P}_{i,h}^{\rho_i}(s,\boldsymbol{a})}\left[\overline{V}_{i,h+1}^{k,\rho_i}\right] - \sigma_{\mathcal{P}_{i,h}^{\rho_i}(s,\boldsymbol{a})}\left[\underline{V}_{i,h+1}^{k,\rho_i}\right]. \quad (108)$$

By applying eq. 107 and eq. 108 to eq. 106, we obtain:

$$\overline{Q}_{i,h}^{k,\rho_i}(s,\boldsymbol{a}) - \underline{Q}_{i,h}^{k,\rho_i}(s,\boldsymbol{a}) \leq A + B + 2\beta_{i,h}^{k,\rho_i}(s,\boldsymbol{a}). \tag{109}$$

We can upper bound term $A$ using a concentration argument tailored for KL robust expectations from Lemma 30, which shows that

$$A \leq 2\beta_{i,h}^{k,\rho_i}(s,\boldsymbol{a}). \tag{110}$$

For term $B$, we use the definition of $\mathbb{E}_{\mathcal{P}_h^\rho(s,a)}[V]$ from eq. 12 to establish the following bound:

$$
\begin{aligned}
B = &\sup_{\eta \in \left[0,\frac{H}{\rho_i}\right]} \left\{ -\eta \log \left( \mathbb{E}_{P_h^\star(\cdot|s,\boldsymbol{a})} \left[ \exp\left\{ -\frac{\overline{V}_{i,h+1}^{k,\rho_i}}{\eta} \right\} \right] \right) - \eta\rho_i \right\} \\
&\quad - \sup_{\eta \in \left[0,\frac{H}{\rho_i}\right]} \left\{ -\eta \log \left( \mathbb{E}_{P_h^\star(\cdot|s,\boldsymbol{a})} \left[ \exp\left\{ -\frac{\underline{V}_{i,h+1}^{k,\rho_i}}{\eta} \right\} \right] \right) - \eta\rho_i \right\} \\
\leq &\sup_{\eta \in [0,H/\rho_i]} \eta \left\{ \log\left( \mathbb{E}_{P_h^\star(\cdot|s,\boldsymbol{a})} \left[ \exp\left\{ -\frac{\underline{V}_{i,h+1}^{k,\rho_i}}{\eta} \right\} \right] \right) \right. \\
&\quad \left. - \log\left( \mathbb{E}_{P_h^\star(\cdot|s,\boldsymbol{a})} \left[ \exp\left\{ -\frac{\overline{V}_{i,h+1}^{k,\rho_i}}{\eta} \right\} \right] \right) \right\} \\
= &\sup_{\eta \in [0,H/\rho_i]} \eta \log\left( \frac{\mathbb{E}_{P_h^\star(\cdot|s,\boldsymbol{a})}\left[ \exp\left\{ -\frac{\underline{V}_{i,h+1}^{k,\rho_i}}{\eta} \right\} \right]}{\mathbb{E}_{P_h^\star(\cdot|s,\boldsymbol{a})}\left[ \exp\left\{ -\frac{\overline{V}_{i,h+1}^{k,\rho_i}}{\eta} \right\} \right]} \right) \\
= &\sup_{\eta \in [0,H/\rho_i]} \eta \log\left( 1 + \frac{\mathbb{E}_{P_h^\star(\cdot|s,\boldsymbol{a})}\left[ \exp\left\{ -\frac{\underline{V}_{i,h+1}^{k,\rho_i}}{\eta} \right\} - \exp\left\{ -\frac{\overline{V}_{i,h+1}^{k,\rho_i}}{\eta} \right\} \right]}{\mathbb{E}_{P_h^\star(\cdot|s,\boldsymbol{a})}\left[ \exp\left\{ -\frac{\overline{V}_{i,h+1}^{k,\rho_i}}{\eta} \right\} \right]} \right) \\
\overset{(a)}{\leq} &\sup_{\eta \in [0,H/\rho_i]} \eta \frac{\mathbb{E}_{P_h^\star(\cdot|s,\boldsymbol{a})}\left[ \exp\left\{ -\frac{\underline{V}_{i,h+1}^{k,\rho_i}}{\eta} \right\} - \exp\left\{ -\frac{\overline{V}_{i,h+1}^{k,\rho_i}}{\eta} \right\} \right]}{\mathbb{E}_{P_h^\star(\cdot|s,\boldsymbol{a})}\left[ \exp\left\{ -\frac{\overline{V}_{i,h+1}^{k,\rho_i}}{\eta} \right\} \right]} \\
\overset{(b)}{\leq} &\sup_{\eta \in [\underline{\eta},H/\rho_i]} \eta \exp\left\{ \frac{H}{\underline{\eta}} \right\} \mathbb{E}_{P_h^\star(\cdot|s,\boldsymbol{a})}\left[ \exp\left\{ -\frac{\underline{V}_{i,h+1}^{k,\rho_i}}{\eta} \right\} - \exp\left\{ -\frac{\overline{V}_{i,h+1}^{k,\rho_i}}{\eta} \right\} \right] \\
\overset{(c)}{\leq} &\exp\left\{ \frac{H}{\underline{\eta}} \right\} \mathbb{E}_{P_h^\star(s,\boldsymbol{a})}\left[ \overline{V}_{i,h+1}^{k,\rho_i} - \underline{V}_{i,h+1}^{k,\rho_i} \right],
\end{aligned}
\tag{111}
$$

where inequality (a) uses the fact that $\log(1+x) \leq x$, inequality (b) holds because $0 \leq \overline{V}_{i,h+1}^{k,\rho_i} \leq H$ and $\eta \in [\underline{\eta}, H/\rho_i]$, and inequality (c) is due to the $\frac{1}{\underline{\eta}}$-Lipschitz continuity of $\phi_\eta(x) = \exp\left\{ -\frac{x}{\eta} \right\}$ for $x \geq 0$, as well as $\underline{V}_{i,h+1}^{k,\rho_i} \leq \overline{V}_{i,h+1}^{k,\rho_i}$.

By applying the bounds for $A$ and $B$ to eq. 109, we get

$$\overline{Q}_{i,h}^{k,\rho_i}(s,\boldsymbol{a}) - \underline{Q}_{i,h}^{k,\rho_i}(s,\boldsymbol{a}) \leq \exp\left\{ \frac{H}{\underline{\eta}} \right\} \mathbb{E}_{P_h^\star(s,\boldsymbol{a})}\left[ \overline{V}_{i,h+1}^{k,\rho_i} - \underline{V}_{i,h+1}^{k,\rho_i} \right] + 4\beta_h^{k,\rho_i}(s,\boldsymbol{a}). \tag{112}$$

Using Lemma 31 to upper bound the bonus term, and rearranging the terms, we further obtain:

$$
\begin{aligned}
\overline{Q}_{i,h}^{k,\rho_i}(s,\mathbf{a}) - \underline{Q}_{i,h}^{k,\rho_i}(s,\mathbf{a}) \leq &\exp\left\{ \frac{H}{\underline{\eta}} \right\} \mathbb{E}_{P_h^\star(s,\mathbf{a})}\left[ \overline{V}_{i,h+1}^{k,\rho_i} - \underline{V}_{i,h+1}^{k,\rho_i} \right] \\
&+ \frac{4c_1 H}{\rho_{\min}} \sqrt{ \frac{\iota^2}{\{N_h^k(s,\boldsymbol{a}) \vee 1\} P_{\min}^\star} } + \sqrt{\frac{4}{K}},
\end{aligned}
\tag{113}
$$

where $c_1 > 0$ is an absolute constant. From the definitions in eq. 8, the difference in V-functions is given by:

$$\overline{V}_{i,h}^{k,\rho_i}(s) - \underline{V}_{i,h}^{k,\rho_i}(s) = \mathbb{E}_{\mathbf{a} \sim \pi^k(\cdot|s)}\left[\overline{Q}_{i,h}^{k,\rho_i}(s,\mathbf{a}) - \underline{Q}_{i,h}^{k,\rho_i}(s,\mathbf{a})\right]. \tag{114}$$

We now define a new recursive value function $\widetilde{V}_h^{k,\rho_{\min}}$ and a corresponding Q-function $\widetilde{Q}_h^{k,\rho_{\min}}$ with $\widetilde{V}_{H+1}^{k,\rho_{\min}} = 0$, where $\rho_{\min} = \min_{i \in \mathcal{M}} \rho_i$:

$$\widetilde{Q}_h^{k,\rho_{\min}}(s,\boldsymbol{a}) = \exp\left\{\frac{H}{\eta}\right\} \mathbb{E}_{P_h^\star(s,\mathbf{a})}\left[\widetilde{V}_{h+1}^{k,\rho_{\min}}\right] + \frac{4c_1 H}{\rho_{\min}}\sqrt{\frac{\iota^2}{\{N_h^k(s,\boldsymbol{a}) \vee 1\}P_{\min}^\star}} + \sqrt{\frac{4}{K}}. \tag{115}$$

$$\widetilde{V}_h^{k,\rho_{\min}}(s) = \mathbb{E}_{\boldsymbol{a} \sim \pi_h^k(\cdot|s)}\left[\widetilde{Q}_{i,h}^{k,\rho_{\min}}(s,\mathbf{a})\right]. \tag{116}$$

By an inductive proof, we can show that for any $(i,h,s,\mathbf{a}) \in \mathcal{M} \times [H] \times \mathcal{S} \times \mathcal{A}$, the following bounds hold:

$$\max_{i \in \mathcal{M}}(\overline{Q}_{i,h}^{k,\rho_i} - \underline{Q}_{i,h}^{k,\rho_i})(s,\boldsymbol{a}) \leq \widetilde{Q}_h^{k,\rho_{\min}}(s,\boldsymbol{a}), \tag{117}$$

$$\max_{i \in \mathcal{M}}(\overline{V}_{i,h}^{k,\rho_i} - \underline{V}_{i,h}^{k,\rho_i})(s) \leq \widetilde{V}_h^{k,\rho_{\min}}(s). \tag{118}$$

Therefore, our analysis can focus on bounding the sum $\sum_{k=1}^K \widetilde{V}_1^{k,\rho_{\min}}(s_1^k)$. For simplicity, we introduce the following notations for the differences at any $(h,k) \in [H] \times [K]$:

$$\Delta_h^k := \widetilde{V}_h^{k,\rho_{\min}}(s_h^k), \tag{119}$$

$$\zeta_h^k := \Delta_h^k - \widetilde{Q}_h^{k,\rho_{\min}}(s_h^k, \mathbf{a}_h^k), \tag{120}$$

$$\xi_h^k := \mathbb{E}_{P_h^\star(\cdot|s_h^k, \mathbf{a}_h^k)}[\widetilde{V}_{h+1}^{k,\rho_{\min}}] - \Delta_{h+1}^k. \tag{121}$$

We can confirm that $\{\zeta_h^k\}_{(h,k)}$ and $\{\xi_h^k\}_{(h,k)}$ are martingale difference sequences with respect to their respective filtrations. By substituting eq. 115 into eq. 120, we obtain the recursive relationship:

$$\Delta_{i,h}^k = \zeta_{i,h}^k + \widetilde{Q}_h^{k,\rho_{\min}}(s_h^k, \mathbf{a}_h^k)$$

$$\leq \zeta_{i,h}^k + \exp\left\{\frac{H}{\eta}\right\}\mathbb{E}_{P_h^\star(s,\mathbf{a})}\left[\widetilde{V}_{h+1}^{k,\rho_{\min}}\right] + \frac{4c_1 H}{\rho_{\min}}\sqrt{\frac{\iota^2}{\{N_h^k(s,\boldsymbol{a}) \vee 1\}P_{\min}^\star}} + \sqrt{\frac{4}{K}}$$

$$= \zeta_{i,h}^k + \exp\left\{\frac{H}{\eta}\right\}\xi_{i,h}^k + \exp\left\{\frac{H}{\eta}\right\}\Delta_{i,h+1}^k + \frac{4c_1 H}{\rho_{\min}}\sqrt{\frac{\iota^2}{\{N_h^k(s,\boldsymbol{a}) \vee 1\}P_{\min}^\star}}$$

$$+ \sqrt{\frac{4}{K}}. \tag{122}$$

By recursively applying eq. 122 and noting that $1 \leq \left(\exp\left\{\frac{H}{\eta}\right\}\right)^h \leq \left(\exp\left\{\frac{H}{\eta}\right\}\right)^H := d_H$, we can upper bound the right hand side of eq. 105 as:

$$\text{Regret}_{\mathsf{NASH}}(K) \leq \sum_{k=1}^K \Delta_1^k \leq c' d_H \sum_{k=1}^K \sum_{h=1}^H \left\{ (\zeta_h^k + \xi_h^k) \right.$$

$$\left. + \left(\frac{4c_1 H}{\rho_{\min}}\sqrt{\frac{\iota^2}{\{N_h^k(s,\boldsymbol{a}) \vee 1\}P_{\min}^\star}} + \sqrt{\frac{4}{K}}\right) \right\}. \tag{123}$$

Next, we bound each of these two main terms. The first term, a sum of martingale differences, is bounded using the Azuma-Hoeffding inequality from Lemma 39, yielding:

$$\sum_{k=1}^K \sum_{h=1}^H (\zeta_{i,h}^k + \xi_{i,h}^k) \leq c_1' \sqrt{H^3 KL}, \tag{124}$$

where $c_1' > 0$ is an absolute constant. For the second term, we apply the proof lines of (Liu et al., 2021, Theorem 3) to bound the sum of the inverse counts:

$$\sum_{k=1}^{K}\sum_{h=1}^{H}\sqrt{\frac{1}{\{N_h^k(s_h^k, \boldsymbol{a}_h^k) \vee 1\}}} \leq c_2'\left(\sqrt{H^2 K S \prod_{i \in \mathcal{M}} A_i} + HS \prod_{i \in \mathcal{M}} A_i\right). \tag{125}$$

By applying eq. 125 to the second term of eq. 123, we get the following:

$$\sum_{k=1}^{K}\sum_{h=1}^{H}\left(\frac{4c_1 H}{\rho_{\min}}\sqrt{\frac{\iota^2}{\{N_h^k(s, \boldsymbol{a}) \vee 1\}P_{\min}^\star}} + \sqrt{\frac{4}{K}}\right) \leq c_2'\left(\sqrt{\frac{H^4 K S\left(\prod_{i \in \mathcal{M}} A_i\right)\iota^2}{\rho_{\min}^2 P_{\min}^\star}}\right.$$
$$\left. + \frac{H^2 S\left(\prod_{i \in \mathcal{M}} A_i\right)\iota}{\rho_{\min}\sqrt{P_{\min}^\star}} + \sqrt{H^2 K}\right). \tag{126}$$

By combining the bounds for both terms in eq. 123, we can upper bound the final regret as follows:

$$\text{Regret}_{\text{NASH}}(K) \leq c' d_H\left(\sqrt{\frac{H^4 K S\left(\prod_{i \in \mathcal{M}} A_i\right)\iota^2}{\rho_{\min}^2 P_{\min}^\star}}\right)$$
$$= \mathcal{O}\left(\sqrt{\frac{H^4 \exp(2H^2) K S\left(\prod_{i \in \mathcal{M}} A_i\right)(\iota')^3}{\rho_{\min}^2 P_{\min}^\star}}\right). \tag{127}$$

This completes the proof of Theorem 5. $\qquad\square$

**Remark 29.** *The proof techniques for bounding* $\text{Regret}_{\text{CCE}}(K)$ *and* $\text{Regret}_{\text{CE}}(K)$ *follow the same lines of proof for* $\text{Regret}_{\text{NASH}}(K)$, *leveraging Lemma 33 and Lemma 34, respectively, in the context of KL-DRMG.*

### G.1 KEY LEMMAS FOR KL-DRMG

**Lemma 30** (Concentration Bound for Robust Value Estimators in KL-DRMG). *Let* $\mathcal{E}_{KL}$ *be the typical event and let the bonus term* $\beta_{i,h}^k$ *be set defined in eq. 103. Then, the following inequality holds:*

$$\sigma_{\widehat{\mathcal{P}_{i,h}^{\rho_i}}(s,\boldsymbol{a})}\left[\overline{V}_{i,h+1}^{k,\rho_i}\right] - \sigma_{\mathcal{P}_{i,h}^{\rho_i}(s,\boldsymbol{a})}\left[\overline{V}_{i,h+1}^{k,\rho_i}\right] + \sigma_{\mathcal{P}_{i,h}^{\rho_i}(s,\boldsymbol{a})}\left[\underline{V}_{i,h+1}^{k,\rho_i}\right] - \sigma_{\widehat{\mathcal{P}_{i,h}^{\rho_i}}(s,\boldsymbol{a})}\left[\underline{V}_{i,h+1}^{k,\rho_i}\right]$$
$$\leq \frac{2c_1 H}{\rho_{\min}}\sqrt{\frac{\iota}{\{N_h^k(s,\boldsymbol{a}) \vee 1\}\widehat{P}_{\min,h}^k(s,\boldsymbol{a})}} + \sqrt{\frac{2}{K}}, \tag{128}$$

*where* $\iota = \log\left(S^3\left(\prod_{i=1}^m A_i\right)H^2 K^{3/2}/\delta\right)$, *and* $c_1 > 0$ *is an absolute constant.*

*Proof.* We begin by defining the term that we need to bound. Let's denote this term by $A$:

$$A := \sigma_{\widehat{\mathcal{P}_h^\rho}(s,\boldsymbol{a})}\left[\overline{V}_{h+1}^k\right] - \sigma_{\mathcal{P}_h^\rho(s,\boldsymbol{a})}\left[\overline{V}_{h+1}^k\right] + \sigma_{\mathcal{P}_h^\rho(s,\boldsymbol{a})}\left[\underline{V}_{h+1}^k\right] - \sigma_{\widehat{\mathcal{P}_h^\rho}(s,\boldsymbol{a})}\left[\underline{V}_{h+1}^k\right]. \tag{129}$$

Under the high-probability event $\mathcal{E}_{KL}$, we can directly apply the concentration inequality given in Lemma 37. This allows us to upper bound $A$ as follows:

$$A \leq \frac{2c_1 H}{\rho_{\min}}\sqrt{\frac{\iota}{\{N_h^k(s,\boldsymbol{a}) \vee 1\}\widehat{P}_{\min,h}^k(s,\boldsymbol{a})}} + \sqrt{\frac{2}{K}}, \tag{130}$$

where $c_1 > 0$ is an absolute constant and $\iota = \log\left(S^3\left(\prod_{i=1}^m A_i\right)H^2 K^{3/2}/\delta\right)$. This bound is exactly the bonus term multiplied by a constant. Therefore, based on our choice of $\beta_{i,h}^k(s,\boldsymbol{a})$ as defined in eq. 103, the inequality in eq. 128 holds. This completes the proof of Lemma 30. $\qquad\square$

**Lemma 31** (Bound of the bonus term for KL-DRMG). *Let* $\mathcal{E}_{KL}$ *be the typical event, the bonus term* $\beta_{i,h}^k$ *in eq. 103 is bounded by*

$$\beta_{i,h}^k(s,\boldsymbol{a}) \leq \frac{c_1 H}{\rho_{\min}}\sqrt{\frac{\iota^2}{\{N_h^k(s,\boldsymbol{a}) \vee 1\}P_{\min}^\star}} + \sqrt{\frac{1}{K}}, \tag{131}$$

*where* $\iota = \log\left(S^3\left(\prod_{i=1}^m A_i\right)H^2 K^{3/2}/\delta\right)$, *and* $c_1 > 0$ *is an absolute constant.*

*Proof.* The proof-lines are similar to (Ghosh et al., 2025, Lemma K.7). We recall the choice of $\beta_{i,h}^k$ as given in eq. 103, i.e.

$$\beta_{i,h}^k(s, \boldsymbol{a}) = \frac{2 c_f H}{\rho_i} \sqrt{\frac{\iota}{\{N_h^k(s, \boldsymbol{a}) \vee 1\} \widehat{P}_{\min,h}^k(s, \boldsymbol{a})}} + \sqrt{\frac{1}{K}}, \tag{132}$$

where $\iota = \log \left( S^3 \left( \prod_{i=1}^m A_i \right) H^2 K^{3/2} / \delta \right)$, $\widehat{P}_{\min,h}^k(s, \boldsymbol{a})$ is defined in eq. 100, and $c_f > 0$ is an absolute constant.

By Lemma 38 and the union bound, it holds that with probability at least $1 - \delta$ that for all $(h, s, \boldsymbol{a}) \in [H] \times \mathcal{S} \times \mathcal{A}$, we get

$$\forall s' \in \mathcal{S}: \quad P_h^\star(s' \mid s, \boldsymbol{a}) \geq \frac{\widehat{P}_h^k(s' \mid s, \boldsymbol{a})}{e^2} \geq \frac{P_h^\star(s' \mid s, \boldsymbol{a})}{8 e^2 \iota}. \tag{133}$$

To characterize the relation between $P_{\min,h}^\star(s, \boldsymbol{a})$ and $\widehat{P}_{\min,h}^k(s, \boldsymbol{a})$ for any $(h, s, \boldsymbol{a}) \in [H] \times \mathcal{S} \times \mathcal{A}$, we suppose—without loss of generality—that $P_{\min,h}^\star(s, \boldsymbol{a}) = P_h^\star(s_1 \mid s, \boldsymbol{a})$ and $\widehat{P}_{\min,h}^k(s, \boldsymbol{a}) = \widehat{P}_h^k(s_2 \mid s, \boldsymbol{a})$ for some $s_1, s_2 \in \mathcal{S}$. Then, it follows that

$$\begin{aligned}
P_{\min,h}^\star(s, \boldsymbol{a}) &= P_h^\star(s_1 \mid s, \boldsymbol{a}) \\
&\overset{(i)}{\geq} \frac{\widehat{P}_h^k(s_1 \mid s, \boldsymbol{a})}{e^2} \geq \frac{\widehat{P}_{\min,h}^k(s, \boldsymbol{a})}{e^2} \\
&= \frac{\widehat{P}_h^k(s_2 \mid s, \boldsymbol{a})}{e^2} \overset{(ii)}{\geq} \frac{P_h^\star(s_2 \mid s, \boldsymbol{a})}{8 e^2 \iota} \\
&\geq \frac{P_{\min,h}^\star(s, \boldsymbol{a})}{8 e^2 \iota} \overset{(iii)}{\geq} \frac{P_{\min}^\star}{8 e^2 \iota}.
\end{aligned} \tag{134}$$

where the inequalities (i) and (ii) follow from eq. 133, and inequality (iii) follows by eq. 102.

By applying eq. 134 in eq. 132, we get

$$\beta_{i,h}^k(s, \boldsymbol{a}) \leq \frac{2 c_f H}{\rho_i} \sqrt{\frac{\iota^2}{\{N_h^k(s, \boldsymbol{a}) \vee 1\} P_{\min}^\star}} + \sqrt{\frac{1}{K}} \leq \frac{c_1 H}{\rho_{\min}} \sqrt{\frac{\iota^2}{\{N_h^k(s, \boldsymbol{a}) \vee 1\} P_{\min}^\star}}$$
$$+ \sqrt{\frac{1}{K}}. \tag{135}$$

This concludes the proof of Lemma 31. $\qquad\square$

NE VERSION: OPTIMISTIC AND PESSIMISTIC ESTIMATION OF THE ROBUST VALUES FOR KL-DRMG.

Here we will proof the optimistic estimations are indeed upper bounds of the corresponding robust V-value and robust Q-value functions fro NE version.

**Lemma 32** (Optimistic and pessimistic estimation of the robust values for KL-DRMG for NE Version). *Under the event $\mathcal{E}_{KL}$ and by setting the bonus term $\beta_{i,h}^k$ as in eq. 103, it holds that*

$$Q_{i,h}^{\dagger, \pi_{-i}^k, \rho_i}(s, \boldsymbol{a}) \leq \overline{Q}_{i,h}^{k, \rho_i}(s, \boldsymbol{a}), \quad \underline{Q}_{i,h}^{k, \rho_i}(s, \boldsymbol{a}) \leq Q_{i,h}^{\pi^k, \rho_i}(s, \boldsymbol{a}), \tag{136}$$

$$V_{i,h}^{\dagger, \pi_{-i}^k, \rho_i}(s) \leq \overline{V}_{i,h}^{k, \rho_i}(s), \quad \underline{V}_{i,h}^{k, \rho_i}(s) \leq V_{i,h}^{\pi^k, \rho_i}(s). \tag{137}$$

*Proof.* The proof-lines are similar to (Ghosh et al., 2025) adapted to the multi-agent case. We will run a proof for each inequality outlined in Lemma 32

- **Ineq. 1:** To prove $Q_{i,h}^{\dagger, \pi_{-i}^k, \rho_i}(s, \boldsymbol{a}) \leq \overline{Q}_{i,h}^{k, \rho_i}(s, \boldsymbol{a})$.

- **Ineq. 2:** To prove $\underline{Q}_{i,h}^{k,\rho_i}(s,\boldsymbol{a}) \leq Q_{i,h}^{\pi^k,\rho_i}(s,\boldsymbol{a})$.

Assume that both eq. 136 and eq. 137 hold at the $(h+1)$-th step.

- **Proof of Ineq. 1:** We first consider robust $Q$ at the $h$-th step. Then, by Proposition 9 (Robust Bellman Equation) and eq. 5, we have that

$$
\begin{aligned}
& Q_{i,h}^{\dagger,\pi^k_{-i},\rho_i}(s,\boldsymbol{a}) - \overline{Q}_{i,h}^{k,\rho_i}(s,\boldsymbol{a}) \\
&= \max\left\{ \sigma_{\mathcal{P}^{\rho_i}_{i,h}(s,\boldsymbol{a})}\left[V_{i,h+1}^{\dagger,\pi^k_{-i},\rho_i}\right] - \sigma_{\widehat{\mathcal{P}^{\rho_i}_{i,h}}(s,\boldsymbol{a})}\left[\overline{V}_{i,h+1}^{k,\rho_i}\right] - \beta_{i,h}^k(s,\boldsymbol{a}), \right. \\
& \qquad\qquad\qquad\qquad \left. Q_{i,h}^{\dagger,\pi^k_{-i},\rho_i}(s,\boldsymbol{a}) - H \right\}, \\
&\leq \max\left\{ \sigma_{\mathcal{P}^{\rho_i}_{i,h}(s,\boldsymbol{a})}\left[V_{i,h+1}^{\dagger,\pi^k_{-i},\rho_i}\right] - \sigma_{\widehat{\mathcal{P}^{\rho_i}_{i,h}}(s,\boldsymbol{a})}\left[V_{i,h+1}^{\dagger,\pi^k_{-i},\rho_i}\right] - \beta_{i,h}^k(s,\boldsymbol{a}), 0 \right\},
\end{aligned}
\tag{138}
$$

where the second inequality follows from the induction of $V_{i,h+1}^{\dagger,\pi^k_{-i},\rho_i} \leq \overline{V}_{i,h+1}^{k,\rho_i}$ at the $h+1$-th step and the fact that $Q_{i,h}^{\dagger,\pi^k_{-i},\rho_i} \leq H$. By Lemma 35 and by the definition of $\widehat{P}_{\min,h}^k(s,\boldsymbol{a})$ as given in eq. 100, we have that

$$
\begin{aligned}
\sigma_{\mathcal{P}^{\rho_i}_{i,h}(s,\boldsymbol{a})}\left[V_{i,h+1}^{\dagger,\pi^k_{-i},\rho_i}\right] - \sigma_{\widehat{\mathcal{P}^{\rho_i}_{i,h}}(s,\boldsymbol{a})}\left[V_{i,h+1}^{\dagger,\pi^k_{-i},\rho_i}\right] &\leq \frac{c_1 H}{\rho_i}\sqrt{\frac{L}{\{N_h^k(s,\boldsymbol{a})\vee 1\}\widehat{P}_{\min,h}^k(s,\boldsymbol{a})}} \\
& \quad + \sqrt{\frac{1}{K}}.
\end{aligned}
\tag{139}
$$

By the choice of $\beta_{i,h}^k$ in eq. 103 and eq. 139 and applying in eq. 138, we conclude that

$$
Q_{i,h}^{\dagger,\pi^k_{-i},\rho_i}(s,\boldsymbol{a}) \leq \overline{Q}_{i,h}^{k,\rho_i}(s,\boldsymbol{a}).
\tag{140}
$$

- **Proof of Ineq. 2:** By using Proposition 9 (Robust Bellman Equation) and eq. 6, we have that

$$
\begin{aligned}
& \underline{Q}_{i,h}^{k,\rho_i}(s,\boldsymbol{a}) - Q_{i,h}^{\pi^k,\rho_i}(s,\boldsymbol{a}) \\
&= \max\left\{ \sigma_{\widehat{\mathcal{P}^{\rho_i}_{i,h}}(s,\boldsymbol{a})}\left[\underline{V}_{i,h+1}^{k,\rho_i}\right] - \sigma_{\mathcal{P}^{\rho_i}_{i,h}(s,\boldsymbol{a})}\left[V_{i,h+1}^{\pi^k,\rho_i}\right] - \beta_{i,h}^k(s,\boldsymbol{a}), \right. \\
& \qquad\qquad\qquad\qquad \left. 0 - Q_{i,h}^{\pi^k,\rho_i}(s,\boldsymbol{a}) \right\}
\end{aligned}
\tag{141}
$$

$$
\leq \max\left\{ \sigma_{\widehat{\mathcal{P}^{\rho_i}_{i,h}}(s,\boldsymbol{a})}\left[V_{i,h+1}^{\pi^k,\rho_i}\right] - \sigma_{\mathcal{P}^{\rho_i}_{i,h}(s,\boldsymbol{a})}\left[V_{i,h+1}^{\pi^k,\rho_i}\right] - \beta_{i,h}^k(s,\boldsymbol{a}), 0 \right\},
\tag{142}
$$

where the second inequality follows from the induction of $\underline{V}_{i,h+1}^{k,\rho_i} \leq V_{i,h+1}^{\pi^k,\rho_i}$ at the $(h+1)$-th step and the fact that $Q_{i,h}^{\pi^k,\rho_i} \geq 0$. By Lemma 36, we get

$$
\begin{aligned}
\sigma_{\widehat{\mathcal{P}^{\rho_i}_{i,h}}(s,\boldsymbol{a})}\left[V_{i,h+1}^{\pi^k,\rho_i}\right] - \sigma_{\mathcal{P}^{\rho_i}_{i,h}(s,\boldsymbol{a})}\left[V_{i,h+1}^{\pi^k,\rho_i}\right] &\leq \frac{c_1 H}{\rho_i}\sqrt{\frac{L}{\{N_h^k(s,\boldsymbol{a})\vee 1\}\widehat{P}_{\min,h}^k(s,\boldsymbol{a})}} \\
& \quad + \sqrt{\frac{1}{K}}.
\end{aligned}
\tag{143}
$$

By the choice of $\beta_{i,h}^k$ in eq. 103 and eq. 143 and applying in eq. 142, we conclude that

$$
Q_{i,h}^{\dagger,\pi^k_{-i},\rho_i}(s,\boldsymbol{a}) \leq \overline{Q}_{i,h}^{k,\rho_i}(s,\boldsymbol{a}).
\tag{144}
$$

Therefore, by eq. 140 and eq. 144, we have proved that at step $h$, it holds that

$$Q_{i,h}^{\dagger,\pi^k_{-i},\rho_i}(s,\boldsymbol{a}) \leq \overline{Q}_{i,h}^{k,\rho_i}(s,\boldsymbol{a}), \quad \underline{Q}_{i,h}^{k,\rho_i}(s,\boldsymbol{a}) \leq Q_{i,h}^{\pi^k,\rho_i}(s,\boldsymbol{a}). \tag{145}$$

We now assume that eq. 136 hold for $h$-th step. Then, by the definition of robust value function as given by robust Bellman equation (Proposition 9), eq. 8, and NASH Equilibrium, we get

$$\overline{V}_{i,h}^{k,\rho_i}(s) = \mathbb{E}_{\boldsymbol{a}\sim\pi^k(\cdot|s)}\left[\overline{Q}_{i,h}^{k,\rho_i}(s,\mathbf{a})\right] = \max_{\pi'_i}\mathbb{E}_{\boldsymbol{a}\sim\pi'_i\times\pi^k_{-i}(\cdot|s)}\left[\overline{Q}_{i,h}^{k,\rho_i}(s,\mathbf{a})\right]. \tag{146}$$

By the definition of $V_{i,h}^{\dagger,\pi^k_{-i},\rho_i}(s)$ in eq. 3, we get

$$V_{i,h}^{\dagger,\pi^k_{-i},\rho_i}(s) = \max_{\pi'_i}\mathbb{E}_{\boldsymbol{a}\sim\pi'_i\times\pi^k_{-i}(\cdot|s)}\left[Q_{i,h}^{\dagger,\pi^k_{-i},\rho_i}(s,\mathbf{a})\right]. \tag{147}$$

Sine by induction, for any $(s,\mathbf{a})$, $\overline{Q}_{i,h}^{k,\rho_i}(s,\mathbf{a}) \geq Q_{i,h}^{\dagger,\pi^k_{-i},\rho_i}(s,\mathbf{a})$. As a result, we also have $\overline{V}_{i,h}^{k,\rho_i}(s) \geq V_{i,h}^{\dagger,\pi^k_{-i},\rho_i}(s)$, which is eq. 137 for $h$-th step. Similarly, we can show that

$$\begin{aligned}
\underline{V}_{i,h}^{k,\rho_i}(s) &= \mathbb{E}_{\boldsymbol{a}\sim\pi^k(\cdot|s)}\left[\underline{Q}_{i,h}^{k,\rho_i}(s,\mathbf{a})\right], \\
&\overset{(i)}{\leq} \mathbb{E}_{\boldsymbol{a}\sim\pi^k(\cdot|s)}\left[Q_{i,h}^{\pi^k,\rho_i}(s,\mathbf{a})\right], \\
&\overset{(ii)}{=} V_{i,h}^{\pi^k,\rho_i}(s),
\end{aligned} \tag{148}$$

where (i) is due to the fact that $\underline{Q}_{i,h}^{k,\rho_i}(s,\boldsymbol{a}) \leq Q_{i,h}^{\pi^k,\rho_i}(s,\boldsymbol{a})$ and (ii) is by definition of $V_{i,h}^{\pi^k,\rho_i}(s)$ as given by Bellman equation in Proposition 9. $\square$

CCE VERSION: OPTIMISTIC AND PESSIMISTIC ESTIMATION OF THE ROBUST VALUES FOR KL-DRMG.

Here we will proof the optimistic estimations are indeed upper bounds of the corresponding robust V-value and robust Q-value functions fro CCE version.

**Lemma 33** (Optimistic and pessimistic estimation of the robust values for KL-DRMG for CCE Version). *Under the event $\mathcal{E}_{KL}$ and by setting the bonus term $\beta_{i,h}^k$ as in eq. 103, it holds that*

$$Q_{i,h}^{\dagger,\pi^k_{-i},\rho_i}(s,\boldsymbol{a}) \leq \overline{Q}_{i,h}^{k,\rho_i}(s,\boldsymbol{a}), \quad \underline{Q}_{i,h}^{k,\rho_i}(s,\boldsymbol{a}) \leq Q_{i,h}^{\pi^k,\rho_i}(s,\boldsymbol{a}), \tag{149}$$

$$V_{i,h}^{\dagger,\pi^k_{-i},\rho_i}(s) \leq \overline{V}_{i,h}^{k,\rho_i}(s), \quad \underline{V}_{i,h}^{k,\rho_i}(s) \leq V_{i,h}^{\pi^k,\rho_i}(s). \tag{150}$$

*Proof.* The proof-lines are similar to (Ghosh et al., 2025) adapted to the multi-agent case.
We will run a proof for each inequality outlined in Lemma 33

- **Ineq. 1:** To prove $Q_{i,h}^{\dagger,\pi^k_{-i},\rho_i}(s,\boldsymbol{a}) \leq \overline{Q}_{i,h}^{k,\rho_i}(s,\boldsymbol{a})$.

- **Ineq. 2:** To prove $\underline{Q}_{i,h}^{k,\rho_i}(s,\boldsymbol{a}) \leq Q_{i,h}^{\pi^k,\rho_i}(s,\boldsymbol{a})$.

Assume that both eq. 149 and eq. 150 hold at the $(h+1)$-th step.

- **Proof of Ineq. 1:** We first consider robust $Q$ at the $h$-th step. Then, by Proposition 9 (Robust Bellman Equation) and eq. 5, we have that

$$\begin{aligned}
& Q_{i,h}^{\dagger,\pi^k_{-i},\rho_i}(s,\boldsymbol{a}) - \overline{Q}_{i,h}^{k,\rho_i}(s,\boldsymbol{a}) \\
&= \max\left\{\sigma_{\mathcal{P}_{i,h}^{\rho_i}(s,\boldsymbol{a})}\left[V_{i,h+1}^{\dagger,\pi^k_{-i},\rho_i}\right] - \sigma_{\widehat{\mathcal{P}_{i,h}^{\rho_i}}(s,\boldsymbol{a})}\left[\overline{V}_{i,h+1}^{k,\rho_i}\right] - \beta_{i,h}^k(s,\boldsymbol{a}),\right. \\
&\qquad\qquad \left. Q_{i,h}^{\dagger,\pi^k_{-i},\rho_i}(s,\boldsymbol{a}) - H\right\}, \\
&\leq \max\left\{\sigma_{\mathcal{P}_{i,h}^{\rho_i}(s,\boldsymbol{a})}\left[V_{i,h+1}^{\dagger,\pi^k_{-i},\rho_i}\right] - \sigma_{\widehat{\mathcal{P}_{i,h}^{\rho_i}}(s,\boldsymbol{a})}\left[V_{i,h+1}^{\dagger,\pi^k_{-i},\rho_i}\right] - \beta_{i,h}^k(s,\boldsymbol{a}),\ 0\right\}, \tag{151}
\end{aligned}$$

where the second inequality follows from the induction of $V_{i,h+1}^{\dagger,\pi_{-i}^k,\rho_i} \le \overline{V}_{i,h+1}^{k,\rho_i}$ at the $h+1$-th step and the fact that $Q_{i,h}^{\dagger,\pi_{-i}^k,\rho_i} \le H$. By Lemma 35 and by the definition of $\widehat{P}_{\min,h}^k(s,\boldsymbol{a})$ as given in eq. 100, we have that

$$\sigma_{\mathcal{P}_{i,h}^{\rho_i}(s,\boldsymbol{a})} \left[V_{i,h+1}^{\dagger,\pi_{-i}^k,\rho_i}\right] - \sigma_{\widehat{\mathcal{P}_{i,h}^{\rho_i}}(s,\boldsymbol{a})} \left[V_{i,h+1}^{\dagger,\pi_{-i}^k,\rho_i}\right] \le \frac{c_1 H}{\rho_i} \sqrt{\frac{L}{\{N_h^k(s,\boldsymbol{a}) \vee 1\}\widehat{P}_{\min,h}^k(s,\boldsymbol{a})}}$$
$$+ \sqrt{\frac{1}{K}}. \tag{152}$$

By the choice of $\beta_{i,h}^k$ in eq. 103 and eq. 152 and applying in eq. 151, we conclude that

$$Q_{i,h}^{\dagger,\pi_{-i}^k,\rho_i}(s,\boldsymbol{a}) \le \overline{Q}_{i,h}^{k,\rho_i}(s,\boldsymbol{a}). \tag{153}$$

- **Proof of Ineq. 2:** By using Proposition 9 (Robust Bellman Equation) and eq. 6, we have that

$$\underline{Q}_{i,h}^{k,\rho_i}(s,\boldsymbol{a}) - Q_{i,h}^{\pi^k,\rho_i}(s,\boldsymbol{a}$$

$$= \max\left\{\sigma_{\widehat{\mathcal{P}_{i,h}^{\rho_i}}(s,\boldsymbol{a})} \left[\underline{V}_{i,h+1}^{k,\rho_i}\right] - \sigma_{\mathcal{P}_{i,h}^{\rho_i}(s,\boldsymbol{a})} \left[V_{i,h+1}^{\pi^k,\rho_i}\right] - \beta_{i,h}^k(s,\boldsymbol{a}),\right.$$

$$\left. 0 - Q_{i,h}^{\pi^k,\rho_i}(s,\boldsymbol{a})\right\}$$

$$\le \max\left\{\sigma_{\widehat{\mathcal{P}_{i,h}^{\rho_i}}(s,\boldsymbol{a})} \left[V_{i,h+1}^{\pi^k,\rho_i}\right] - \sigma_{\mathcal{P}_{i,h}^{\rho_i}(s,\boldsymbol{a})} \left[V_{i,h+1}^{\pi^k,\rho_i}\right] - \beta_{i,h}^k(s,\boldsymbol{a}), 0\right\}, \tag{154}$$

where the second inequality follows from the induction of $\underline{V}_{i,h+1}^{k,\rho_i} \le V_{i,h+1}^{\pi^k,\rho_i}$ at the $(h+1)$-th step and the fact that $Q_{i,h}^{\pi^k,\rho_i} \ge 0$. By Lemma 36, we get

$$\sigma_{\widehat{\mathcal{P}_{i,h}^{\rho_i}}(s,\boldsymbol{a})} \left[V_{i,h+1}^{\pi^k,\rho_i}\right] - \sigma_{\mathcal{P}_{i,h}^{\rho_i}(s,\boldsymbol{a})} \left[V_{i,h+1}^{\pi^k,\rho_i}\right] \le \frac{c_1 H}{\rho_i} \sqrt{\frac{L}{\{N_h^k(s,\boldsymbol{a}) \vee 1\}\widehat{P}_{\min,h}^k(s,\boldsymbol{a})}}$$
$$+ \sqrt{\frac{1}{K}}. \tag{155}$$

By the choice of $\beta_{i,h}^k$ in eq. 103 and eq. 155 and applying in eq. 154, we conclude that

$$Q_{i,h}^{\dagger,\pi_{-i}^k,\rho_i}(s,\boldsymbol{a}) \le \overline{Q}_{i,h}^{k,\rho_i}(s,\boldsymbol{a}). \tag{156}$$

Therefore, by eq. 153 and eq. 156, we have proved that at step $h$, it holds that

$$Q_{i,h}^{\dagger,\pi_{-i}^k,\rho_i}(s,\boldsymbol{a}) \le \overline{Q}_{i,h}^{k,\rho_i}(s,\boldsymbol{a}), \quad \underline{Q}_{i,h}^{k,\rho_i}(s,\boldsymbol{a}) \le Q_{i,h}^{\pi^k,\rho_i}(s,\boldsymbol{a}). \tag{157}$$

We now assume that eq. 149 hold for $h$-th step. Then, by the definition of robust value function as given by robust Bellman equation (Proposition 9), eq. 8, and CCE Equilibrium, we get

$$\overline{V}_{i,h}^{k,\rho_i}(s) = \mathbb{E}_{\boldsymbol{a} \sim \pi^k(\cdot|s)} \left[\overline{Q}_{i,h}^{k,\rho_i}(s,\mathbf{a})\right] \ge \max_{\pi_i'} \mathbb{E}_{\boldsymbol{a} \sim \pi_i' \times \pi_{-i}^k(\cdot|s)} \left[\overline{Q}_{i,h}^{k,\rho_i}(s,\mathbf{a})\right]. \tag{158}$$

By the definition of $V_{i,h}^{\dagger,\pi_{-i}^k,\rho_i}(s)$ in eq. 3, we get

$$V_{i,h}^{\dagger,\pi_{-i}^k,\rho_i}(s) = \max_{\pi_i'} \mathbb{E}_{\boldsymbol{a} \sim \pi_i' \times \pi_{-i}^k(\cdot|s)} \left[Q_{i,h}^{\dagger,\pi_{-i}^k,\rho_i}(s,\mathbf{a})\right]. \tag{159}$$

Sine by induction, for any $(s, \mathbf{a})$, $\overline{Q}_{i,h}^{k,\rho_i}(s, \mathbf{a}) \geq Q_{i,h}^{\dagger, \pi_{-i}^k, \rho_i}(s, \mathbf{a})$. As a result, we also have $\overline{V}_{i,h}^{k,\rho_i}(s) \geq V_{i,h}^{\dagger, \pi_{-i}^k, \rho_i}(s)$, which is eq. 150 for $h$-th step. Similarly, we can show that

$$
\begin{aligned}
\underline{V}_{i,h}^{k,\rho_i}(s) &= \mathbb{E}_{\mathbf{a} \sim \pi^k(\cdot|s)}\left[\underline{Q}_{i,h}^{k,\rho_i}(s, \mathbf{a})\right], \\
&\overset{(i)}{\leq} \mathbb{E}_{\mathbf{a} \sim \pi^k(\cdot|s)}\left[Q_{i,h}^{\pi^k, \rho_i}(s, \mathbf{a})\right], \\
&\overset{(ii)}{=} V_{i,h}^{\pi^k, \rho_i}(s),
\end{aligned}
\tag{160}
$$

where (i) is due to the fact that $\underline{Q}_{i,h}^{k,\rho_i}(s, \mathbf{a}) \leq Q_{i,h}^{\pi^k, \rho_i}(s, \mathbf{a})$ and (ii) is by definition of $V_{i,h}^{\pi^k, \rho_i}(s)$ as given by Bellman equation in Proposition 9. □

CE VERSION: OPTIMISTIC AND PESSIMISTIC ESTIMATION OF THE ROBUST VALUES FOR KL-DRMG.

Here we will proof the optimistic estimations are indeed upper bounds of the corresponding robust V-value and robust Q-value functions fro CE version.

**Lemma 34** (Optimistic and pessimistic estimation of the robust values for KL-DRMG for CE version). *By setting the bonus term $\beta_{i,h}^k$ as in eq. 103, with probability $1 - \delta$, for any $(s, \mathbf{a}, h, i)$ and $k \in [K]$, it holds that*

$$
\max_{\phi \in \Phi_i} Q_{i,h}^{\phi \diamond \pi^k, \rho_i}(s, \mathbf{a}) \leq \overline{Q}_{i,h}^{k,\rho_i}(s, \mathbf{a}), \quad \underline{Q}_{i,h}^{k,\rho_i}(s, \mathbf{a}) \leq Q_{i,h}^{\pi^k, \rho_i}(s, \mathbf{a}),
\tag{161}
$$

$$
\max_{\phi \in \Phi_i} V_{i,h}^{\phi \diamond \pi^k, \rho_i}(s) \leq \overline{V}_{i,h}^{k,\rho_i}(s), \quad \underline{V}_{i,h}^{k,\rho_i}(s) \leq V_{i,h}^{\pi^k, \rho_i}(s).
\tag{162}
$$

*Proof.* The proof-lines are similar to (Ghosh et al., 2025) adapted to the multi-agent case. We will run a proof for each inequality outlined in Lemma 34

- **Ineq. 1:** To prove $\max_{\phi \in \Phi_i} Q_{i,h}^{\phi \diamond \pi^k, \rho_i}(s, \mathbf{a}) \leq \overline{Q}_{i,h}^{k,\rho_i}(s, \mathbf{a})$.

- **Ineq. 2:** To prove $\underline{Q}_{i,h}^{k,\rho_i}(s, \mathbf{a}) \leq Q_{i,h}^{\pi^k, \rho_i}(s, \mathbf{a})$.

Assume that both eq. 161 and eq. 162 hold at the $(h + 1)$-th step.

- **Proof of Ineq. 1:** We first consider robust $Q$ at the $h$-th step. Then, by Proposition 9 (Robust Bellman Equation) and eq. 5, we have that

$$
\max_{\phi \in \Phi_i} Q_{i,h}^{\phi \diamond \pi^k, \rho_i}(s, \mathbf{a}) - \overline{Q}_{i,h}^{k,\rho_i}(s, \mathbf{a})
$$

$$
= \max \left\{ \sigma_{\mathcal{P}_{i,h}^{\rho_i}(s,\mathbf{a})}\left[\max_{\phi \in \Phi_i} V_{i,h}^{\phi \diamond \pi^k, \rho_i}\right] - \sigma_{\widehat{\mathcal{P}_{i,h}^{\rho_i}}(s,\mathbf{a})}\left[\overline{V}_{i,h+1}^{k,\rho_i}\right] - \beta_{i,h}^k(s, \mathbf{a}), \right.
$$

$$
\left. \max_{\phi \in \Phi_i} Q_{i,h}^{\phi \diamond \pi^k, \rho_i}(s, \mathbf{a}) - H \right\}
$$

$$
\leq \max \left\{ \sigma_{\mathcal{P}_{i,h}^{\rho_i}(s,\mathbf{a})}\left[\max_{\phi \in \Phi_i} V_{i,h}^{\phi \diamond \pi^k, \rho_i}\right] - \sigma_{\widehat{\mathcal{P}_{i,h}^{\rho_i}}(s,\mathbf{a})}\left[\max_{\phi \in \Phi_i} V_{i,h}^{\phi \diamond \pi^k, \rho_i}\right] - \beta_{i,h}^k(s, \mathbf{a}), 0 \right\},
\tag{163}
$$

where the second inequality follows from the induction of $\max_{\phi \in \Phi_i} V_{i,h+1}^{\phi \diamond \pi^k, \rho_i}(s) \leq \overline{V}_{i,h+1}^{k,\rho_i}(s)$ at the $h + 1$-th step and the fact that $\max_{\phi \in \Phi_i} Q_{i,h}^{\phi \diamond \pi^k, \rho_i}(s, \mathbf{a}) \leq H$. By Lemma 35 and by the

definition of $\widehat{P}^k_{\min,h}(s, \boldsymbol{a})$ as given in eq. 100, we have that

$$\sigma_{\mathcal{P}^{\rho_i}_{i,h}(s,\boldsymbol{a})} \left[ \max_{\phi \in \Phi_i} V^{\phi \diamond \pi^k, \rho_i}_{i,h}(s) \right] - \sigma_{\widehat{\mathcal{P}^{\rho_i}_{i,h}}(s,\boldsymbol{a})} \left[ \max_{\phi \in \Phi_i} V^{\phi \diamond \pi^k, \rho_i}_{i,h}(s) \right]$$

$$\leq \frac{c_1 H}{\rho_i} \sqrt{\frac{L}{\{N^k_h(s, \boldsymbol{a}) \vee 1\} \widehat{P}^k_{\min,h}(s, \boldsymbol{a})}} + \sqrt{\frac{1}{K}}. \tag{164}$$

By the choice of $\beta^k_{i,h}$ in eq. 103 and eq. 164 and applying in eq. 163, we conclude that

$$\max_{\phi \in \Phi_i} Q^{\phi \diamond \pi^k, \rho_i}_{i,h}(s, \boldsymbol{a}) \leq \overline{Q}^{k,\rho_i}_{i,h}(s, \boldsymbol{a}). \tag{165}$$

- **Proof of Ineq. 2:** By using Proposition 9 (Robust Bellman Equation) and eq. 6, we have that

$$\underline{Q}^{k,\rho_i}_{i,h}(s, \boldsymbol{a}) - Q^{\pi^k, \rho_i}_{i,h}(s, \boldsymbol{a})$$

$$= \max \left\{ \sigma_{\widehat{\mathcal{P}^{\rho_i}_{i,h}}(s,\boldsymbol{a})} \left[ \underline{V}^{k,\rho_i}_{i,h+1} \right] - \sigma_{\mathcal{P}^{\rho_i}_{i,h}(s,\boldsymbol{a})} \left[ V^{\pi^k,\rho_i}_{i,h+1} \right] - \beta^k_{i,h}(s, \boldsymbol{a}), 0 - Q^{\pi^k,\rho_i}_{i,h}(s, \boldsymbol{a}) \right\},$$

$$\leq \max \left\{ \sigma_{\widehat{\mathcal{P}^{\rho_i}_{i,h}}(s,\boldsymbol{a})} \left[ V^{\pi^k,\rho_i}_{i,h+1} \right] - \sigma_{\mathcal{P}^{\rho_i}_{i,h}(s,\boldsymbol{a})} \left[ V^{\pi^k,\rho_i}_{i,h+1} \right] - \beta^k_{i,h}(s, \boldsymbol{a}), 0 \right\}, \tag{166}$$

where the second inequality follows from the induction of $\underline{V}^{k,\rho_i}_{i,h+1} \leq V^{\pi^k,\rho_i}_{i,h+1}$ at the $(h+1)$-th step and the fact that $Q^{\pi^k,\rho_i}_{i,h} \geq 0$. By Lemma 36, we get

$$\sigma_{\widehat{\mathcal{P}^{\rho_i}_{i,h}}(s,\boldsymbol{a})} \left[ V^{\pi^k,\rho_i}_{i,h+1} \right] - \sigma_{\mathcal{P}^{\rho_i}_{i,h}(s,\boldsymbol{a})} \left[ V^{\pi^k,\rho_i}_{i,h+1} \right] \leq \frac{c_1 H}{\rho_i} \sqrt{\frac{L}{\{N^k_h(s, \boldsymbol{a}) \vee 1\} \widehat{P}^k_{\min,h}(s, \boldsymbol{a})}}$$

$$+ \sqrt{\frac{1}{K}}. \tag{167}$$

By the choice of $\beta^k_{i,h}$ in eq. 103 and eq. 167 and applying in eq. 166, we conclude that

$$\underline{Q}^{k,\rho_i}_{i,h}(s, \boldsymbol{a}) \leq Q^{\pi^k,\rho_i}_{i,h}(s, \boldsymbol{a}). \tag{168}$$

Therefore, by eq. 165 and eq. 168, we have proved that at step $h$, it holds that

$$\max_{\phi \in \Phi_i} Q^{\phi \diamond \pi, \rho_i}_{i,h}(s, \boldsymbol{a}) \leq \overline{Q}^{k,\rho_i}_{i,h}(s, \boldsymbol{a}), \quad \underline{Q}^{k,\rho_i}_{i,h}(s, \boldsymbol{a}) \leq Q^{\pi^k,\rho_i}_{i,h}(s, \boldsymbol{a}). \tag{169}$$

We now assume that eq. 161 hold for $h$-th step. Then, by the definition of robust value function as given by robust Bellman equation (Proposition 9), eq. 8, and CE Equilibrium, we get

$$\overline{V}^{k,\rho_i}_{i,h}(s) = \mathbb{E}_{\boldsymbol{a} \sim \pi^k(\cdot|s)} \left[ \overline{Q}^{k,\rho_i}_{i,h}(s, \mathbf{a}) \right] = \max_{\phi \in \Phi_i} \mathbb{E}_{\boldsymbol{a} \sim \phi \diamond \pi^k(\cdot|s)} \left[ \overline{Q}^{k,\rho_i}_{i,h}(s, \mathbf{a}) \right]. \tag{170}$$

By the definition of $\max_{\phi \in \Phi_i} V^{\phi \diamond \pi^k, \rho_i}_{i,h}(s)$ in eq. 3, we get

$$\max_{\phi \in \Phi_i} V^{\phi \diamond \pi^k, \rho_i}_{i,h}(s) = \max_{\phi \in \Phi_i} \mathbb{E}_{\boldsymbol{a} \sim \phi \diamond \pi^k(\cdot|s)} \left[ \max_{\phi'} Q^{\phi' \diamond \pi^k, \rho_i}_{i,h}(s, \mathbf{a}) \right]. \tag{171}$$

Since by induction, for any $(s, \mathbf{a})$, $\overline{Q}^{k,\rho_i}_{i,h}(s, \mathbf{a}) \geq \max_{\phi \in \Phi_i} Q^{\phi \diamond \pi^k, \rho_i}_{i,h}(s, \mathbf{a})$. As a result, we also have $\overline{V}^{k,\rho_i}_{i,h}(s) \geq \max_{\phi \in \Phi_i} V^{\phi \diamond \pi^k, \rho_i}_{i,h}(s)$, which is eq. 162 for $h$-th step. Similarly, we can show that

$$\underline{V}^{k,\rho_i}_{i,h}(s) = \mathbb{E}_{\boldsymbol{a} \sim \pi^k(\cdot|s)} \left[ \underline{Q}^{k,\rho_i}_{i,h}(s, \mathbf{a}) \right],$$

$$\overset{(i)}{\leq} \mathbb{E}_{\boldsymbol{a} \sim \pi^k(\cdot|s)} \left[ Q^{\pi^k,\rho_i}_{i,h}(s, \mathbf{a}) \right],$$

$$\overset{(ii)}{=} V^{\pi^k,\rho_i}_{i,h}(s), \tag{172}$$

where (i) is due to the fact that $\underline{Q}^{k,\rho_i}_{i,h}(s, \boldsymbol{a}) \leq Q^{\pi^k,\rho_i}_{i,h}(s, \boldsymbol{a})$ and (ii) is by definition of $V^{\pi^k,\rho_i}_{i,h}(s)$ as given by Bellman equation in Proposition 9. $\qquad \square$

### G.2 AUXILIARY LEMMAS FOR KL-DRMG

**Lemma 35** (Concentration of Value Function in KL-DRMG). *Under the typical event $\mathcal{E}_{KL}$ as defined in eq. 104, the following concentration bound holds with probability at least $1 - \delta$:*

$$\left| \sigma_{\widehat{\mathcal{P}_h^{\rho_i}}(s,\boldsymbol{a})} \left[ V_{i,h+1}^{\dagger,\pi_{-i}^k,\rho_i} \right] - \sigma_{\mathcal{P}_h^{\rho_i}(s,\boldsymbol{a})} \left[ V_{i,h+1}^{\dagger,\pi_{-i}^k,\rho_i} \right] \right| \leq \frac{c_1 H}{\rho_i} \sqrt{\frac{L}{\{N_h^k(s,\boldsymbol{a}) \vee 1\} \widehat{P}_{\min,h}^k(s,\boldsymbol{a})}} + \frac{1}{\sqrt{K}},$$

*where $\iota = \log\left( S^3 \left( \prod_{i=1}^m A_i \right) H^2 K^{3/2}/\delta \right)$ and $c_1$ is an absolute constant.*

*Proof.* This proof establishes a concentration bound for the difference between the empirical and true robust value functions. We use the definition of the KL-divergence operator $\sigma_{\mathcal{P}_{i,h}^{\rho_i}(s,\boldsymbol{a})}[V_{i,h+1}^{\dagger,\pi_{-i}^k,\rho_i}]$ from eq. 12 and the empirical minimum probability $\widehat{P}_{\min,h}^k(s,\boldsymbol{a})$ from eq. 100 to express this difference as a supremum:

$$\left| \sigma_{\widehat{\mathcal{P}_{i,h}^{\rho_i}}(s,\boldsymbol{a})} \left[ V_{i,h+1}^{\dagger,\pi_{-i}^k,\rho_i} \right] - \sigma_{\mathcal{P}_{i,h}^{\rho_i}(s,\boldsymbol{a})} \left[ V_{i,h+1}^{\dagger,\pi_{-i}^k,\rho_i} \right] \right|$$

$$\leq \sup_{\eta \in [\underline{\eta}, H/\rho_i]} \eta \left| \log\left( \mathbb{E}_{\widehat{P}_h^k(\cdot|s,\boldsymbol{a})} \left[ \exp\left\{ -\frac{V_{i,h+1}^{\dagger,\pi_{-i}^k,\rho_i}}{\eta} \right\} \right] \right) \right.$$

$$\left. - \log\left( \mathbb{E}_{P_h^\star(\cdot|s,\boldsymbol{a})} \left[ \exp\left\{ -\frac{V_{i,h+1}^{\dagger,\pi_{-i}^k,\rho_i}}{\eta} \right\} \right] \right) \right|. \quad (173)$$

Under the high-probability event $\mathcal{E}_{\text{KL}}$ (defined in eq. 104), we apply a known concentration inequality from (Wang et al., 2024d, Lemma 16) to bound this expression:

$$\left| \sigma_{\widehat{\mathcal{P}_{i,h}^{\rho_i}}(s,\boldsymbol{a})} \left[ V_{i,h+1}^{\dagger,\pi_{-i}^k,\rho_i} \right] - \sigma_{\mathcal{P}_{i,h}^{\rho_i}(s,\boldsymbol{a})} \left[ V_{i,h+1}^{\dagger,\pi_{-i}^k,\rho_i} \right] \right| \leq \frac{c_1 H}{\rho_i} \sqrt{\frac{L}{\{N_h^k(s,\boldsymbol{a}) \vee 1\} \widehat{P}_{\min,h}^k(s,\boldsymbol{a})}}, \quad (174)$$

This bound holds for any $\eta$ within a fine-grained cover of the interval $[0, H/\rho_{\min}]$. By applying a standard covering argument, we extend this bound to hold for all $\eta \in [0, H/\rho_{\min}]$, thereby concluding the proof of Lemma 35. $\qquad \square$

**Lemma 36** (Bound for DRMG-KL and the robust value function of $\pi^k$). *Under event $\mathcal{E}_{KL}$ in eq. 104 and for any EQUILIBRIUM $\in \{NASH, CE, CCE\}$, we assume that the optimism and pessimism inequalities hold at $(h+1, k)$, where these inequalities can correspond to any of the following cases of EQUILIBRIUM:*

- *NE: Lemma 32 using eq. 136 and eq. 137,*

- *CCE: Lemma 33 using eq. 149 and eq. 150,*

- *CE: Lemma 34 using eq. 161 and eq. 162.*

*Then the following bound holds:*

$$\left| \sigma_{\widehat{\mathcal{P}_{i,h}^{\rho_i}}(s,\boldsymbol{a})} \left[ V_{i,h+1}^{\pi^k,\rho_i} \right] - \sigma_{\mathcal{P}_{i,h}^{\rho_i}(s,\boldsymbol{a})} \left[ V_{i,h+1}^{\pi^k,\rho_i} \right] \right| \leq \frac{c_1 H}{\rho_i} \sqrt{\frac{L}{\{N_h^k(s,\boldsymbol{a}) \vee 1\} \widehat{P}_{\min,h}^k(s,\boldsymbol{a})}} + \frac{1}{\sqrt{K}},$$

*where $\iota = \log\left( S^3 \left( \prod_{i=1}^m A_i \right) H^2 K^{3/2}/\delta \right)$, and $c_1$ is an absolute constant.*

*Proof.* This proof establishes a concentration bound for the difference between the empirical and true robust value functions under the KL-divergence. By using the definition of the robust operator

$\sigma_{\mathcal{P}_{i,h}^{\rho_i}(s,\boldsymbol{a})}[V_{i,h+1}^{\pi^k,\rho_i}]$ from eq. 12 and the empirical minimum probability $\widehat{P}_{\min,h}^k(s,\boldsymbol{a})$ from eq. 100, we can bound the absolute difference as follows:

$$
\left| \sigma_{\widehat{\mathcal{P}_{i,h}^{\rho_i}}(s,\boldsymbol{a})} \left[ V_{i,h+1}^{\pi^k,\rho_i} \right] - \sigma_{\mathcal{P}_{i,h}^{\rho_i}(s,\boldsymbol{a})} \left[ V_{i,h+1}^{\pi^k,\rho_i} \right] \right|
$$
$$
\leq \sup_{\eta \in [\underline{\eta}, H/\rho_i]} \eta \left| \log \left( \mathbb{E}_{\widehat{P}_h^k(\cdot|s,\boldsymbol{a})} \left[ \exp\left\{ -\frac{V_{i,h+1}^{\pi^k,\rho_i}}{\eta} \right\} \right] \right) \right.
$$
$$
\left. - \log \left( \mathbb{E}_{P_h^\star(\cdot|s,\boldsymbol{a})} \left[ \exp\left\{ -\frac{V_{i,h+1}^{\pi^k,\rho_i}}{\eta} \right\} \right] \right) \right|. \tag{175}
$$

Under the high-probability event $\mathcal{E}_{\text{KL}}$ (defined in eq. 104), and by applying a known concentration inequality from (Wang et al., 2024d, Lemma 17), we can establish a uniform bound on this difference:

$$
\left| \sigma_{\widehat{\mathcal{P}_{i,h}^{\rho_i}}(s,\boldsymbol{a})} \left[ V_{i,h+1}^{\pi^k,\rho_i} \right] - \sigma_{\mathcal{P}_{i,h}^{\rho_i}(s,\boldsymbol{a})} \left[ V_{i,h+1}^{\pi^k,\rho_i} \right] \right| \leq \frac{c_1 H}{\rho_i} \sqrt{\frac{L}{\{N_h^k(s,\boldsymbol{a}) \vee 1\} \widehat{P}_{\min,h}^k(s,\boldsymbol{a})}}. \tag{176}
$$

This inequality holds for any $\eta$ in a fine-grained cover of the interval $[0, H/\rho_{\min}]$. We conclude the proof of Lemma 36 by using a standard covering argument to extend the bound to all $\eta \in [0, H/\rho_{\min}]$. $\square$

**Lemma 37** (Bounds for RMG-KL and optimistic and pessimistic robust value estimators). *Under event $\mathcal{E}_{KL}$ in eq. 104 and for any* EQUILIBRIUM $\in \{NASH, CE, CCE\}$, *we assume that the optimism and pessimism inequalities hold at $(h+1, k)$, where these inequalities can correspond to any of the following cases of* EQUILIBRIUM:

- ***NE:** Lemma 32 using eq. 136 and eq. 137,*

- ***CCE:** Lemma 33 using eq. 149 and eq. 150,*

- ***CE:** Lemma 34 using eq. 161 and eq. 162.*

*Then the following bound holds:*

$$
\max \left\{ \left| \sigma_{\widehat{\mathcal{P}_{i,h}^{\rho_i}}(s,\boldsymbol{a})} \left[ \overline{V}_{i,h+1}^{k,\rho_i} \right] - \sigma_{\mathcal{P}_{i,h}^{\rho_i}(s,\boldsymbol{a})} \left[ \overline{V}_{i,h+1}^{k,\rho_i} \right] \right|, \left| \sigma_{\widehat{\mathcal{P}_{i,h}^{\rho_i}}(s,\boldsymbol{a})} \left[ \underline{V}_{i,h+1}^{k,\rho_i} \right] - \sigma_{\mathcal{P}_{i,h}^{\rho_i}(s,\boldsymbol{a})} \left[ \underline{V}_{i,h+1}^{k,\rho_i} \right] \right| \right\}
$$
$$
\leq \frac{c_1 H}{\rho_i} \sqrt{\frac{L}{\{N_h^k(s,a) \vee 1\} \widehat{P}_{\min,h}^k(s,\boldsymbol{a})}} + \sqrt{\frac{1}{K}},
$$

*where $\iota = \log \left( S^3 \left( \prod_{i=1}^m A_i \right) H^2 K^{3/2}/\delta \right)$ and $c_1$ is an absolute constant.*

*Proof.* We follow the same proof lines as Lemma 36, and thereby we omit it. $\square$

**Lemma 38** (Bound on Binomal random variable). *Suppose $X \sim Binomial(n, p)$, where $n \geq 1$ and $p \in [0, 1]$. For any $\delta \in (0, 1)$, we have*

$$
X \geq \frac{np}{8 \log\left(\frac{1}{\delta}\right)}, \qquad \text{if } np \geq 8 \log\left(\frac{1}{\delta}\right), \tag{177}
$$

$$
X \leq \begin{cases} e^2 np, & \text{if } np \geq \log\left(\frac{1}{\delta}\right), \\ 2e^2 \log\left(\frac{1}{\delta}\right), & \text{if } np \leq 2 \log\left(\frac{1}{\delta}\right), \end{cases} \tag{178}
$$

*hold with probability at least $1 - 4\delta$.*

*Proof.* Refer to (Shi et al., 2023, Lemma 8) for details. $\square$

# H  OTHER TECHNICAL LEMMAS

Here, we present some auxiliary lemmas which are useful in the proof.

**Lemma 39** (Azuma Hoeffding's Inequality)**.** *Let $\{Z_t\}_{t \in \mathbb{Z}_+}$ be a martingale with respect to the filtration $\{\mathcal{F}_t\}_{t \in \mathbb{Z}_+}$. Assume that there are predictable processes $\{A_t\}_{t \in \mathbb{Z}_+}$ and $\{B_t\}_{t \in \mathbb{Z}_+}$ with respect to $\{\mathcal{F}_t\}_{t \in \mathbb{Z}_+}$, i.e., for all $t$, $A_t$ and $B_t$ are $\mathcal{F}_{t-1}$-measurable, and constants $0 < c_1, c_2, \cdots < +\infty$ such that $A_t \leq Z_t - Z_{t-1} \leq B_t$ and $B_t - A_t \leq c_t$ almost surely. Then, for all $\beta > 0$*

$$\mathbb{P}\left( |Z_t - Z_0| \geq \beta \right) \leq \exp \left\{ - \frac{2\beta^2}{\sum_{i \leq t} c_t^2} \right\}. \tag{179}$$

*Proof.* Refer to the proof of Theorem 5.1 of (Dubhashi & Panconesi, 2009). $\square$

**Lemma 40** (Self-bounding variance inequality (Maurer & Pontil, 2009, Theorem 10))**.** *Let $X_1, \ldots, X_T$ be independent and identically distributed random variables with finite variance, that is, $\mathrm{Var}(X_1) < \infty$. Assume that $X_t \in [0, M]$ for every $t$ with $M > 0$, and let*

$$S_T^2 = \frac{1}{T} \sum_{t=1}^{T} X_t^2 - \left( \frac{1}{T} \sum_{t=1}^{T} X_t \right)^2.$$

*Then, for any $\varepsilon > 0$, we have*

$$\mathbb{P}\left( \left| S_T - \sqrt{\mathrm{Var}(X_1)} \right| \geq \varepsilon \right) \leq 2 \exp \left( - \frac{T\varepsilon^2}{2M^2} \right).$$

*Proof.* Refer to the proof of Lemma 7 of (Panaganti & Kalathil, 2022). $\square$

