# OpenReview forum: "Sample-Efficient Distributionally Robust Multi-Agent Reinforcement Learning via Online Interaction"
_ICLR.cc/2026/Conference — ICLR 2026 Poster_

### Official Review · Reviewer_bkNi · 2025-10-16

**Soundness:** 3
**Presentation:** 3
**Contribution:** 3
**Rating:** 6
**Confidence:** 3

**Summary:**

The paper propose a systematic and principled approach for online learning in distributionally robust Markov Games with environment uncertainty. This is done by first revealing the hardness in online DRMGs,  then proposing an online robust MARL algorithm, Multiplayer Optimistic Robust Nash Value Iteration (MORNAVI), which offers the first provable robust guarantee in the setting. The theoretical analysis demonstrates MORNAVI achieves low regret, finds the optimal robust policy and achieves high sample complexity.

**Strengths:**

Overall the paper is well-written, a little hard to understand due to the nature of a theoretical paper but clear enough to understand given the context. The theories given are rigorious and the threat model, algorithm and proofs seems correct, but I didn't check all the proofs.

1. The hardness of online DRMGs provides a clear motivation of the problem, and the contructed examples gives an intuitive thought experiment for future researchers to design their algorithms.

2. The derivations on sample complexity provides a rigorious theoretical foundation for establishing the regret of existing algorithms, and shows the result achieves complexity comparable with existing works, despite operating in online settings instead of easier generative and offline settings. The complexity bound is quite tight.

**Weaknesses:**

1. The main weakness I belive is that current online MARL parameterized by neural networks still do not have high enough sample efficiency to support this work that learns purely online, without any offline datasets available. However, accepting this paper would greatly benefit future research on this topic.

2. The paper is purely theoretical and might be designed for tabular case, instead of existing MARL using function approximations. While this is beneficial for theoretical analysis, it remains unknown how to adapt this algorithm to modern environments, such as MPE, SMAC and Multi-Agent Mujoco. This is clearly reflected in stage 1, Nominal Transition Estimation and stage 2, EQUILIBRIUM subroutine. Estimating the distribution of future states can be inaccurate in modern environments with large state space using simple empirical average, and computing the equilibrium such as CE or CCE can be hard in tasks such as SMAC, or continuous control.

3. What is the algorithmic differencce between MORNAVI and existing distributional robust approach? I am not very familiar with distributionally robust MARL literature, so I assume the main difference lies in Eqn. 5 and 6, since other parts of the algorithm is not too different from existing robust MARL approach.

To solve these problems, can you provide suggestions on empirical algorithms that leverage the advantage of MORNAVI, but use  function approximations? this would greatly strengthen this paper. However, I give a rating of 6 based on the current version.

**Questions:**

See weakness.

---

> ### Author Response · Authors · 2025-11-19
>
> We sincerely appreciate the reviewer's insightful feedback and suggestions. We have revised our paper accordingly, with all changes highlighted in blue.
>
> We opted to study the tabular case as it represents the most fundamental setting for our initial exploration. Our analysis in the tabular regime establishes the foundational theory and provides the necessary first principles for the study of online robust MARL.
>
> **1. Purely online learning.**
> We agree with the reviewer that in practice, pure online learning can be inefficient, especially under large-scale problems with neural network approximation, which can make the exploration even more challenging. We also agree that practical RL could greatly benefit from some offline dataset, which is known as the hybrid learning. Such methods have shown great potential in efficient RL. It is our future interest to extend our framework to hybrid robust MARL.
>
> **2. Extension to Function Approximation.**
> We agree that, for practical applications, designing algorithms with function approximation can be more effective and efficient than purely tabular methods.
> (1) Although our algorithms and analysis are mainly for the tabular setting, they can be extended to general function approximation. For large-scale problems, one can employ a fitted-typed algorithm and bypass explicit estimation of the transition kernel. Following the ideas in [a,b], the state–action pair-wise distributionally robust optimization (kernel estimation and support-function calculation for each $(s,a)$) can be reformulated as a single functional optimization problem, which can be efficiently solved with function approximation. Our ingredients—optimistic estimation and exploration-term design, handling of non-stationary multi-agent interactions, and mismatched goals among agents—can then be combined with techniques from non-robust Markov game studies [c,d]. More concretely, the dual representation of the robust Bellman operator allows us to replace each per-$(s,a)$ robust backup with a \emph{global} objective over parameterized value and dual-function classes. This shifts the dependence of the algorithm and its guarantees from $|\mathcal{S}|$ and $|\mathcal{A}|$ to the complexity of the underlying function classes (e.g., feature dimension or neural-network capacity).
> (2) Another potential approach is to consider linear function classes, which is a standard abstraction for large-scale problems. By assuming that the underlying (robust) Markov game is linear in a $d$-dimensional feature family, our analysis can be reduced to a $d$-dimensional space, effectively replacing the dependence on $|\mathcal{S}||\mathcal{A}|$ by a dependence on $d$. This connects directly to (Liu et al., 2024), which shows in the single-agent case that distributionally robust off-dynamics RL with linear function approximation can achieve near-optimal performance with sample complexity polynomial in $d$, the horizon, and an off-dynamics coverage coefficient. It is natural to expect that an analogous `linear robust Markov game' model would allow our framework to scale to large state spaces, at the cost of additional technical challenges in equilibrium computation and joint exploration.
> However, both of these directions require substantial additional analysis and are beyond the scope of this paper. We hence leave them as future directions.
>
> We will address the final weakness/question in a separate comment due to the platform's character limit.

---

> > ### Author Response · Authors · 2025-11-19
> >
> > **3.Algorithmic Differences.**
> >
> > You are correct in noting that the major algorithmic difference lies in those equations (specifically, the bonus term $\beta$). However, we highlight that, although algorithms enjoy similar high-level structures across different settings, the design of these terms is significantly different and challenging. As we discussed, in offline or generative-model settings, agents do not need to actively explore the environment, since the dataset already provides all the information needed. In contrast, the online setting requires coverage-based formulations in non-robust online RL [f] —agents must strategically balance exploration and exploitation to ensure small regret. Our design of the bonus term $\beta$ explicitly accounts for all sources of randomness and uncertainty in online MARL [g], including other agents’ updating policies and mismatches between the nominal and worst-case environments, to ensure optimism and to properly balance exploration. As we discussed in the hardness section (Sec.~4), our online robust multi-agent setting is significantly more challenging than offline or generative settings and hence requires additional study.
> >
> > Beyond the tabular case, extending this design to general function approximation would require genuinely new algorithmic ingredients. Results on single-agent distributionally robust off-dynamics RL with linear function approximation [e] and on non-robust online RL with general value-function approximation [f,g] show that one cannot simply plug function approximation into a tabular (robust) Q-learning scheme and use purely local, count-based bonuses. Instead, the exploration bonus must act as a \emph{global} uncertainty quantifier derived from the fitted (robust) Bellman error over the function classes and calibrated by a coverage or visitation-ratio coefficient linking the behavior distribution to the (robust) target visitation distribution. Because the data collected during learning are non-stationary and policy-dependent, the bonus must control the accumulated estimation error along entire trajectories under evolving policies, rather than just per–state-action deviations.
> >
> > In a multi-agent robust setting, the analogue of our bonus $\beta$ under function approximation would therefore need to (i) control the joint estimation error of all agents’ robust $Q$-functions and associated dual variables, (ii) propagate this uncertainty through the equilibrium computation at each state (e.g., CE/CCE), and (iii) maintain optimism uniformly over the coupled policy updates of all agents and over the worst-case dynamics. Existing robust MARL methods based on offline data or generative models largely avoid these issues because exploration and equilibrium computation are decoupled. In our online robust MARL framework, these aspects must be integrated into a single learning procedure, making the development of multi-agent, function-approximate bonuses a technically substantial but promising direction for future work.
> >
> > [a] Kishan Panaganti, Zaiyan Xu, Dileep Kalathil, Mohammad Ghavamzadeh. \emph{Robust Reinforcement Learning using Offline Data}. NeurIPS, 2022.
> >
> > [b] Kishan Panaganti, Adam Wierman, Eric Mazumdar. \emph{Model-Free Robust $\phi$-Divergence Reinforcement Learning Using Both Offline and Online Data}. ICML, 2024.
> >
> > [c] Chi Jin, Qinghua Liu, Tiancheng Yu. \emph{The Power of Exploiter: Provable Multi-Agent RL in Large State Spaces}. ICML, 2022.
> >
> > [d] Baihe Huang et al. \emph{Towards General Function Approximation in Zero-Sum Markov Games}. arXiv:2107.14702, 2021.
> >
> > [e] Zhishuai Liu and Pan Xu. \emph{Distributionally Robust Off-Dynamics Reinforcement Learning: Provable Efficiency with Linear Function Approximation}. AISTATS, 2024.
> >
> > [f] Ruosong Wang, Russ R. Salakhutdinov, and Lin Yang. \emph{Reinforcement Learning with General Value Function Approximation: Provably Efficient Approach via Bounded Eluder Dimension}. NeurIPS, 2020.
> >
> > [g] Tengyang Xie, Dylan J. Foster, Yu Bai, Nan Jiang, and Sham M. Kakade. \emph{The Role of Coverage in Online Reinforcement Learning}. arXiv:2210.04157, 2022.

---

> > > ### Comment · Reviewer_bkNi · 2025-11-19
> > > **Response**
> > >
> > > Thanks for the response. I believe it is a solid work in all aspects, especially considering the new experiments added in rebuttal. I recommend accepting this paper.

---

> > > > ### Author Response · Authors · 2025-11-19
> > > > **Thank you for your support!**
> > > >
> > > > Dear reviewer,
> > > >
> > > > We are delighted to hear that our response addressed your concerns. Thank you for taking the time to re-assess the paper and for raising your score. We look forward to finalizing the manuscript with the improvements inspired by your review.
> > > >
> > > > Sincerely,
> > > > Authors

---

### Official Review · Reviewer_cbht · 2025-10-27

**Soundness:** 3
**Presentation:** 3
**Contribution:** 3
**Rating:** 4
**Confidence:** 2

**Summary:**

This paper tackles online learning in Distributionally Robust Markov Games (DRMGs) and proposes f-MORNAVI—a model-based algorithm that learns empirical transition models, constructs optimistic and pessimistic value estimates under f-divergence uncertainty, and computes equilibria (Nash/CCE/CE) at each step. The authors prove hardness results, showing that support-shift uncertainty (e.g., TV sets) yields linear regret. They also derive upper bounds for total variation and KL uncertainty, establishing near-matching sample-efficient guarantees. The work provides the first theoretical framework for online DRMGs with rigorous proofs and detailed analysis.

**Strengths:**

- The paper supplies both lower bounds (separations) and matching upper bounds (for TV and KL) together. The proof structure is standard but carefully adapted to the robust multi-agent setting.
- The paper identifies and formalizes the online DRMG problem (vs. prior offline/generative-model work) and isolates two distinct hardness phenomena (support shift and curse-of-multi-agency).

**Weaknesses:**

- All upper bounds and the lower bounds include the product of agent action counts. This is a severe scalability concern (exponential in number of agents if each has many actions). The paper acknowledges this as an open question but does not give practical guidance or alleviate it. This limits real-world applicability.
- The algorithm requires solving an equilibrium (Nash/CE/CCE) in the stagewise matrix game for each state and timestep. In practice large action space and many states make these subroutines expensive. The paper needs to discuss practical computational approaches and complexity per episode.
- The KL regret/sample complexity contains an $exp(O(H^2))$ term. For long horizons this is prohibitive; more discussion of whether this dependence is inherent (and how it scales in practice) is needed.

**Questions:**

- Lack of empirical validation despite practical motivation: The paper is motivated by bridging the sim-to-real gap in MARL, yet includes no experiments or case studies. This omission significantly weakens the claim that the approach improves practical robustness or connects theory and real-world performance. Would you be able to empirically show your proposed method compared with previous work in several simulation benchmarks?

---

> ### Author Response · Authors · 2025-11-19
>
> We sincerely appreciate the reviewer's insightful feedback and suggestions. We have revised our paper accordingly, with all changes highlighted in blue.
>
> **1. Curse of Multi-Agency.**
> We highlight that in Sec 4, we developed a hardness result showing that in robust DRMGs, the minimax lower bound on sample complexity inherently depends on the joint action space and cannot be avoided except adopting additional structure assumptions (see discussions in our paper). Moreover, since the robust value function estimation requires full knowledge of the transition kernel (to find the worst-case kernel), robust RL is generally model-based and thus this dependence cannot be improved, without additional assumptions (in contrast, non-robust RL can be done in model-free fashion, and the curse can be broken).
>
> **2. Computational Cost.**
> In this paper, we follow the standard studies on Markov games and  mainly consider sample complexity and statistical efficiency. In the case of NE, in each step, we need to find a NE of a matrix-form game, which can be PPAD-hard in the worst-case. However, it is a standard assumption in Markov game learning, that the matrix-formed Nash equilibrium can be computed. Moreover, the computational intractability of Nash equilibrium motivates us to study CE and CCE, whose computation is polynomial. We have included a discussion in our updated version.
>
> **3. No Numerical Experiments.**
> We thank you for pointing this out. We added a Numerical Experiments section to our updated paper. Specifically, we consider two DRMGs (a fully cooperative game and a general-sum game). We compare our algorithm with non-robust baselines, and test the robustness of the learned equilibria under model uncertainties. As our results shown, our robust algorithm is much more robust and stable under model uncertainties, hence effectively closes the sim-to-real gap and validates our theoretical results.
>
> **4. KL Sample Complexity.**
> We thank you for pointing this out. These terms therefore reflect the inherent hardness of KL-based robust RL, and are also standard in prior studies like (Blanchet 2023; Shi 2024).
>
> We highlight that the $\exp$ term arises from the intrinsic properties of the KL set structure (the duality form of its corresponding distributionally robust optimization, in eq.~(12), where the worst-case value involves a log-moment generating function and thus inherently exponentiates future value terms). Such a term is standard in all KL-based robust RL studies. It can be directly traded for an explicit dependence on $(P_{\min}^\star)^{-1}$ (Blanchet 2023; Shi 2024; Ghosh 2025), but not removed. Such a form of blow-up in either $H$ or $P_{\min}^\star$ is unavoidable for KL-robustness.
>
> Similarly, the $1/(P_{\min}^\star \sigma^2)$ term in our bound (which is also standard in KL-based analysis) reflects the statistical difficulty of estimating rare transitions when the robustness radius is small: as $\sigma \to 0$, the KL ball becomes nearly degenerate and our unified analysis (which is designed to hold for all $\sigma$) becomes loose, rather than indicating any pathological behavior of the algorithm. Notably, in robust RL, the support of the whole transition kernel is needed to account for the worst case; hence, robust RL becomes inefficient at exploring low-probability transitions and naturally introduces dependence on such terms (Si 2020; Blanchet 2023).
>
> In practice, for moderate horizons with $P_{\min}^\star$ bounded away from zero, these worst-case factors remain controlled and do not pose a scalability issue. We have added a short discussion highlighting these points in the revised paper.
>
> [Blanchet 2023]  Blanchet, Jose, et al. "Double pessimism is provably efficient for distributionally robust offline reinforcement learning: Generic algorithm and robust partial coverage." Advances in Neural Information Processing Systems 36 (2023): 66845-66859.
>
> [Shi 2024] Shi, Laixi, and Yuejie Chi. "Distributionally robust model-based offline reinforcement learning with near-optimal sample complexity." Journal of Machine Learning Research 25.200 (2024): 1-91.
>
> [Ghosh 2025] Ghosh, Debamita, George K. Atia, and Yue Wang. "Provably near-optimal distributionally robust reinforcement learning in online settings." arXiv preprint arXiv:2508.03768 (2025).
>
> [Si 2020] Si, Nian, et al. "Distributionally robust policy evaluation and learning in offline contextual bandits." International Conference on Machine Learning. PMLR, 2020.

---

> ### Author Response · Authors · 2025-11-24
> **Request for updates**
>
> Dear Reviewer,
>
> Thank you again for your constructive feedback on our submission. We are writing to gently follow up on our responses posted. We have updated the paper to address your major concerns, specifically by adding numerical experiments, and discussing/clarifying the weakness points you mentioned.
>
> Could you please let us know if these updates and our responses have resolved your concerns? We are committed to improving the paper further and would value any additional feedback you might have.
>
> Best regards,
> The Authors

---

### Official Review · Reviewer_axWW · 2025-10-30

**Soundness:** 2
**Presentation:** 2
**Contribution:** 3
**Rating:** 4
**Confidence:** 2

**Summary:**

This paper introduces f-MORNAVI, a model-based online algorithm for distributionally robust Markov games (DRMGs). The method estimates the dynamics from interaction with the environment and performs planning under uncertainty sets defined by f-divergences. It incorporates an equilibrium solver and provides regret bounds for online DRMGs, along with lower bounds that highlight inherent hardness.

**Strengths:**

+ The theoretical development is clear, with high-probability regret bounds for TV and KL uncertainty sets and sample-complexity corollaries to equilibrium under the NE, CE, and CCE.

+ The algorithmic design well-designed and motivated. It separates model estimation, robust optimistic planning with divergence-aware bonuses, and an equilibrium, and the mathematical treatment of support shift is interesting and well-written.

**Weaknesses:**

-- The paper lacks empirical validation. Although the theoretical results looks sound, there is a lack of experimental evidence that the proposed online method outperforms prior approaches or that the constants or overheads are practical.

-- The practical comparison to generative or offline baselines and to out-of-distribution scenarios is unclear. It would help to quantify how the robust online procedure fares against strong non-robust or offline/generative methods on OOD tasks.

-- While the theory derives sample-complexity corollaries for reaching approximate equilibrium, the lack of experiments makes it hard to assess the real-world gap between these bounds.

**Questions:**

see the weaknesses section

---

> ### Author Response · Authors · 2025-11-19
>
> We sincerely appreciate the reviewer's insightful feedback and suggestions. We have revised our paper accordingly, with all changes highlighted in blue.
>
> We first want to clarify that our major goal is to develop theoretical studies of distributionally robust Markov games, to understand its fundamental hardness and learning guarantees, hence we mainly focus on theoretical aspect. However, we do agree that some practical considerations can further enhance our paper.
>
> **1. No Numerical Experiment.**
> We thank you for pointing this out. We added a Numerical Experiments section to our updated paper. Specifically, we consider two DRMGs (a fully cooperative game and a general-sum game). We compare our algorithm with non-robust baselines (note that there is no prior online robust learning algorithm), and test the robustness of the learned equilibria under model uncertainties. As our results shown, our robust algorithm is much more robust and stable under model uncertainties, hence effectively closes the sim-to-real gap and validates our theoretical results.
>
> **2. Comparison with offline/generative setting.**
> We would like to clarify that these are fundamentally different settings and represent different problems. Namely, depending on data accesses, different settings are considered, and different algorithms are designed specifically for those settings. Hence, it is generally impractical or unfair to directly compare with different settings. We developed a comprehensive comparison (on sample complexity) in our paper (Sec 6), where we detailed discussed the differences between them.
>
> **3. OOD tasks.**
> In our newly developed experiments, the performance is tested under OOD tasks. Namely, all algorithms are trained under the nominal environment, and tested under the corresponding worst-case environment. These testing environments are from the uncertainty set and are not observed or met during training, which are hence out-of-distribution tasks. Our experiment results show that our algorithms maintain a high performance under model mismatch and OOD tasks, validating the enhanced effectiveness and robustness of our methods.

---

> ### Author Response · Authors · 2025-11-24
> **Request for updates**
>
> Dear Reviewer,
>
> Thank you again for your constructive feedback on our submission. We are writing to gently follow up on our responses posted. We have updated the paper to address your major concerns, specifically by adding numerical experiments, and providing comparisons with existing works.
>
> Could you please let us know if these updates and our responses have resolved your concerns? We are committed to improving the paper further and would value any additional feedback you might have.
>
> Best regards,
> The Authors

---

### Official Review · Reviewer_cwx1 · 2025-11-01

**Soundness:** 3
**Presentation:** 3
**Contribution:** 3
**Rating:** 6
**Confidence:** 4

**Summary:**

The paper tackles online distributionally robust Markov games (DRMGs) without a simulator or offline dataset and proposes f-MORNAVI, an optimistic-robust, model-based meta-algorithm for general f-divergence uncertainty sets (with concrete TV and KL instantiations). It proves (i) hardness results—linear regret with support shift and √K regret without shift but still scaling with the joint action size—and (ii) first regret bounds for online DRMGs in the TV/KL cases, plus corresponding sample-complexity bounds. Algorithmically, f-MORNAVI estimates a nominal kernel online, plans via robust Bellman operators augmented with UCB-style bonuses tailored to the uncertainty geometry, and computes an equilibrium (NE/CE/CCE) at each step.

**Strengths:**

Originality

The paper addresses a relatively unexplored problem: online distributionally robust Markov games (DRMGs) without access to simulators or offline data. While the formulation itself extends concepts familiar from single-agent robust RL and generative/offline DRMG studies, applying them to the online multi-agent regime is a natural but nontrivial step. The proposed MORNAVI framework—integrating optimism for exploration with robustness against uncertainty—is conceptually consistent with prior work on optimistic robust RL, though its multi-agent adaptation and generalization to
𝑓
f-divergence sets give it modest originality. The contribution is incremental rather than groundbreaking but helps close an existing theoretical gap.

Quality

The theoretical development is careful and technically competent. The paper provides hardness results, upper bounds, and sample-complexity analysis that align with known results in related literature. The regret guarantees for both Total Variation and KL uncertainty sets are plausible extensions of existing robust RL theory. However, the analysis follows established proof techniques (empirical Bernstein inequalities, Bellman contraction arguments, etc.) rather than introducing fundamentally new analytical tools. Some aspects—such as the correctness of the correlated equilibrium definitions and the reliance on specific assumptions like rectangularity and failure states—should be clarified or corrected to fully validate the results. Overall, the quality is solid but not exceptional.

Clarity

The paper’s structure is standard and logical, with a clear flow from problem motivation to algorithm design and theoretical results. Nevertheless, the writing is mathematically dense, and several sections would benefit from higher-level intuition to guide readers through technical derivations. Certain notational inconsistencies (e.g., repeated theorem numbering, unclear equilibrium notation) reduce readability. While the main ideas can be followed by experts in the field, the exposition is unlikely to be easily accessible to a broader ICLR audience without additional clarifications or illustrative examples.

Significance

The significance of the paper lies primarily in its problem setting rather than in the methodological innovation. Establishing regret bounds for online DRMGs is useful for the theory of robust multi-agent learning, but practical impact remains limited given the heavy dependence on joint action space size and the lack of empirical validation. The work reinforces the difficulty of scaling robustness in multi-agent systems but does not yet provide clear strategies to mitigate these challenges. The theoretical results are incremental but may serve as a foundation for future improvements in scalability or algorithmic design.

**Weaknesses:**

1. Limited Conceptual Novelty Beyond Extension
While the paper presents a rigorous treatment of online Distributionally Robust Markov Games (DRMGs), its conceptual novelty is limited. The proposed MORNAVI algorithm largely repackages existing principles—namely optimism in exploration and robust Bellman operators—previously developed in single-agent robust RL (e.g., Wang & Zou, NeurIPS 2021; Dong et al., ICML 2022; Panaganti & Kalathil, ICML 2022). Extending these to the multi-agent setting is a logical next step but not a fundamentally new paradigm. The paper would benefit from a clearer articulation of what new technical difficulties arise in the multi-agent online case (beyond joint-action explosion) and how MORNAVI specifically overcomes them.

2. Overstated Claims of Firstness and Theoretical Gap
The paper repeatedly claims to be the first to address online DRMGs with provable guarantees, yet concurrent and closely related works—such as RONAVI (Farhat et al., 2025, arXiv)—already study similar formulations with optimism–robustness integration and provide comparable regret bounds. Moreover, previous generative or oracle-based DRMG works (Shi et al., 2024; Ma et al., 2023) have already laid much of the theoretical groundwork. The authors should moderate their 'first provable guarantee' claim and explicitly position their contribution in relation to these concurrent developments.

3. Incomplete or Inaccurate Definitions
There are important definitional inaccuracies that need correction before the theory can be considered reliable:
- The robust coarse correlated equilibrium (CCE) definition in Section 2 is identical to the robust Nash equilibrium definition, which is incorrect.
- The paper assumes existence of robust NE without proof. For general-sum DRMGs, NE existence is nontrivial; only CE/CCE are guaranteed.
Correcting these definitions and clarifying the associated assumptions would strengthen the theoretical soundness and interpretability of the results.

4. Dependence on Restrictive Assumptions
Several assumptions used to make the analysis tractable are strong and may limit practical relevance:
- The rectangular uncertainty set assumption eliminates coupling across states and agents, simplifying proofs but overlooking realistic uncertainty.
- The failure-states assumption for TV uncertainty ensures that unseen transitions are learnable, which may not hold in online exploration.
- The algorithm presupposes centralized model updates and full observability.
A section explicitly acknowledging these assumptions’ implications—and discussing potential relaxation strategies—would improve credibility.

5. Lack of Empirical or Illustrative Validation
Although the paper is theoretical, it would benefit from a minimal empirical illustration or simulation. A simple 2-player gridworld or coordination game with environmental noise could demonstrate how MORNAVI behaves in practice, whether the derived regret bounds are observable, and how robustness manifests under distribution shifts.

6. Unresolved Scalability Challenge
While the authors discuss the curse of multi-agency (joint action dependence), the paper stops short of offering even partial mitigation strategies. Theoretical exploration of structured policies (mean-field approximations, factored models, or correlated policies) could point toward reducing the exponential dependence on ∏Ai.

7. Exposition and Structural Issues
The exposition can be improved in several ways:
- Duplicate theorem numbering causes confusion.
- Several notations (e.g., σ_𝒫[V], ρ_min, P_min) are undefined when first introduced.
- The confidence interval construction is only in the appendix and should be summarized in the main text.
Clarifying these would improve readability and allow reviewers to verify correctness more easily.

Summary of Actionable Suggestions
1. Correct the CCE/CE definitions and restate all regret results accordingly.
2. Clearly differentiate the paper from concurrent work like RONAVI and moderate novelty claims.
3. Discuss the impact and realism of rectangular and failure-state assumptions.
4. Include a small synthetic experiment to illustrate algorithm behavior.
5. Explore scalability strategies to reduce dependence on joint action size.
6. Fix theorem numbering, ensure consistent notation, and summarize key proof steps in the main text.

**Questions:**

1. Clarification of the Equilibrium Definitions
The definition of robust coarse correlated equilibrium (CCE) in Section 2 appears identical to the Nash equilibrium condition. Could the authors clarify whether this was intentional, and if not, provide the correct formulation of the CCE obedience constraints? If the CCE definition is corrected, would this change any of the stated regret bounds or equilibrium existence claims? A clarification of whether the theoretical guarantees hold for all equilibrium notions (NE, CE, CCE) under the same assumptions would be very helpful.

2. Existence of Robust NE and Practical Computability
The paper defines a robust NE as a product policy, but general-sum robust games may not guarantee existence. Are there known conditions under which the robust NE considered in this paper is guaranteed to exist (e.g., convex–concave payoff structures or zero-sum cases)? How is the equilibrium computed in practice within the MORNAVI framework—via an exact solver or approximate methods? Including an explanation of computational feasibility would make the algorithmic contribution clearer.

3. Clarification of the Failure-State Assumption
The regret bound under TV divergence relies on the failure-state assumption, which seems to restrict uncertainty to transitions that are still reachable through exploration. Could the authors formalize this assumption more explicitly and discuss its implications? What happens if this assumption is violated—does the regret bound degrade gracefully, or does the algorithm fail entirely? A sensitivity analysis or a theoretical relaxation would strengthen the argument.

4. Scope of the Rectangular Uncertainty Set
The analysis assumes rectangular (decoupled) uncertainty sets across states and agents, which simplifies the dynamic programming recursion but limits expressiveness. Could the authors comment on whether their approach could handle non-rectangular (coupled) uncertainty sets, perhaps through approximate decomposition? Would any part of the regret proof break down under correlated uncertainties across agents?

5. Regret Bound Tightness and Scaling with Joint Actions
The regret bounds in Theorems 2 and 3 scale with the product of action space sizes (∏Ai), which makes them impractical for even moderate numbers of agents. Could the authors clarify whether this dependence is inherent or a proof artifact? Are there potential structural assumptions (e.g., mean-field or factored game structures) that could reduce this dependence while maintaining robustness?

6. Comparisons with Concurrent Work
The authors position MORNAVI as the first to provide online DRMG guarantees, but RONAVI (Farhat et al., 2025) and related works appear to address similar settings. Could the authors provide a more explicit comparison in terms of assumptions (e.g., oracle access, divergence type), theoretical guarantees, and computational complexity? If RONAVI uses similar optimism–robustness design principles, what distinguishes MORNAVI’s theoretical contribution?

7. Confidence Interval Construction and Proof Transparency
The proof of optimism for the robust Q-value relies on confidence intervals that bound the uncertainty-adjusted Bellman operator. Could the authors sketch the key steps or inequalities (e.g., dual form of σ𝒫[V]) in the main text to make the logic more transparent? How do the bonus terms differ in structure between the TV and KL cases, and what intuition explains these differences?

8. Empirical or Illustrative Demonstration
Even a small-scale experiment could provide insight into how the algorithm performs under model mismatch. Could the authors include or discuss results on a simple two-player coordination or adversarial environment? Observing whether the empirical regret trend aligns with the theoretical rate would help substantiate the practical value of the theoretical development.

9. Theoretical Open Questions
The paper concludes by raising the question of whether online DRMG algorithms can overcome the curse of multi-agency. Could the authors elaborate on potential directions—such as hierarchical decomposition, correlated equilibrium relaxation, or partial coordination—that might reduce this scaling? Are there theoretical obstacles (e.g., impossibility results) suggesting that sublinear regret without ∏Ai dependence might be unattainable?

10. Expository and Structural Improvements
There are a few presentation issues that would benefit from revision:
- Duplicate numbering of Theorem 1 for hardness and upper-bound results.
- Undefined symbols (e.g., σ𝒫[V], ρmin, Pmin) at their first appearance.
- Several long equations could use short textual interpretation lines to help readers follow the logic.
Addressing these would significantly improve readability and make the theoretical arguments easier to follow.

Summary
The main clarifications that could substantially change my evaluation are:
- Correcting and explaining the CCE/CE definitions and their impact on results.
- Providing clearer justification for key assumptions (failure states, rectangularity).
- Offering an explicit comparison with concurrent works.
- Demonstrating even minimal empirical validation or illustrating scalability considerations.
These improvements would make the contribution more transparent, the assumptions more credible, and the theoretical results easier to interpret and verify.

---

> ### Author Response · Authors · 2025-11-19
>
> We sincerely appreciate the reviewer's insightful feedback and suggestions. We have revised our paper accordingly, with all changes highlighted in blue.
>
> **1. Comparison with single-agent online robust RL.**
> Although optimism is a standard principle in online learning, we respectfully disagree with the reviewer that our works are standard/trivial extension from single-agent settings. Multi-agent RL is known to be significantly challenging than single-agent RL in the following aspects, which need to be tackled in our studies.
> (1). A single agent learns under a stationary, unchanged environment; whereas in multi-agent, each agent updates its policy simultaneously, resulting in a non-stationary and rapidly changing environment for each agent. Consequentially, each agent needs to actively explore the unknown environment and tackle the randomness from both underlying dynamics and other agents' policies.
> (2). In single-agent RL, the learning goal, the optimal value function, is unique, and RL algorithms generally enjoy stable convergence guarantees. However, in MARL, the solutions are not unique (e.g., the Nash equilibrium is generally not unique), and the algorithms need to address the `equilibrium selection' issue, to ensure a stable performance.
> (3). More importantly, when additionally consider robustness, since the rewards are different for agents, their corresponding worst-case transition kernels are also different, whereas single agent robust RL considers only a single environment.
> To address these issues, our MORNAVI needs to carefully design the exploration term to consider the potential randomness from the underlying dynamics, other agents' policies and their worst-case kernel, to ensure a tight upper bound for optimism. As our results shown, our algorithm obtains a near-optimal regret bound, implying the tightness of our design.
>
> **2. Comparison with prior works.**
> We want to clarify that (Farhat et al., 2025, arXiv) is a  concurrent work with ours. We also highlight that we provide the first results under the **online setting**. The previous works mentioned are under the generative model setting, where the samples can be freely generated. Instead, we consider the more challenging online exploration setting, and we carefully balance the explorations and exploitations, which is even more challenging under robust setting (see, e.g., Lu 2024, Ghosh 2025, He 2025).
>
>  **3. Incomplete/Inaccurate Definitions.**
> We would like to address the misunderstandings. The definitions of NE and CCE are different, as NE requires a product policy, whereas CCE considers a joint policy (where each agent may have correlated policies). The existence of robust equilibria is not assumed; Instead, as we mentioned in Line 178, it is proved in (Blanchet et al., 2023).
>
> **4. Restrictive Assumptions.**
> (1). The rectangularity assumption is standard and fundamental in robust RL. More importantly, without the rectangular assumption, the robust RL problem is generally NP-hard (W. Wiesemann, D. Kuhn, and B. Rustem. Robust Markov decision processes. Mathematics of Operations Research, 38(1):153–183, 2013.) and cannot be solved efficiently.
> (2). The failure-state assumption is also standard and necessary for online robust RL under TV setting (Panaganti et al., 2022; Lu et al., 2024). As we proved in Sec E of Appendix, without it, the resulting problem can require exponentially large samples.
> (3). Our algorithm follows the standard Centralized Training with Decentralized Execution (CTDE) structure.   In robust RL, since agent's worst-case performance depends on the nominal kernel, developing a fully decentralized algorithm can be significantly challenging (e.g., in (Ma 2023), the fully decentralized algorithm is design with an impractical oracle). Similarly, due to the hardness of robust RL, we consider fully observable setting, and leave partially observable DRMGs as future works. We have clarified these in our paper.
>
> **5. No Numerical Experiments.**
> We have developed and included a Numerical Experiments section to our paper (please refer to Sec D of Appendix). As our results shown, our robust learning algorithms obtain a more stable and robust performance under model mismatch, implying that we effectively close the sim-to-real gap.
>
> **6. Scalability Challenge.**
> We highlight that in Sec 4, we developed a hardness result showing that in robust DRMGs, the minimax lower bound on sample complexity inherently depends on the joint action space and cannot be avoided. Moreover, since the robust value function estimation requires full knowledge of the transition kernel (to find the worst-case kernel), thus the dependence cannot be improved. We agree that with additional structure assumptions, the dependence may be improved (as we discussed in our paper), but under the most general DRMGs, such a dependence may not be improved.
>
> **7. Paper Structure.**
> We appreciate your suggestions and have fixed them accordingly.

---

> > ### Comment · Reviewer_cwx1 · 2025-11-28
> >
> > Thank you for the detailed rebuttal and for incorporating revisions into the manuscript. I summarize below which of my concerns have been addressed and which remain open.
> >
> > The rebuttal address the following items:
> > Comparison with single-agent online robust RL and articulation of multi-agent–specific challenges
> > Positioning relative to concurrent works and clarifying that RONAVI is concurrent, not prior
> > Clarification of NE vs. CCE definitions and citing Blanchet et al. (2023) for existence
> > Discussion and justification of rectangularity, failure-state assumptions, and CTDE structure
> > Addition of a numerical experiment illustrating robust performance under model mismatch
> > Explanation regarding inherent joint-action dependence and hardness results
> > Fixes to structural issues (theorem numbering, notation clarifications, etc.)
> >
> >
> > Several concerns remain only partially addressed:
> >
> > 1. Conceptual novelty and technical distinctiveness.
> >
> > The rebuttal argues that MARL introduces additional challenges, but the explanation remains high-level, and the paper still does not clearly articulate what concrete new technical difficulties arise in the multi-agent robust setting—beyond joint-action explosion—and how MORNAVI specifically overcomes them. This limits the sense of novelty.
> >
> > 2. Clarification of robust equilibrium notions.
> >
> > The rebuttal insists the definitions of NE and CCE are distinct, but in the original text the mathematical definitions appeared identical. If the definitions differ only in the descriptive phrasing (“product policy” vs. “joint policy”), the formal constraints still need to reflect the obedience conditions for CCE. This remains ambiguous and may still require correction.
> >
> > 3. Scope of robustness assumptions.
> >
> > While the rebuttal correctly notes that rectangularity and failure-state assumptions are standard, it does not explain how sensitive the results are to violations of these assumptions or whether any partial relaxations might exist. A short sensitivity or discussion would improve interpretability.
> >
> > 4. Comparison with concurrent work.
> >
> > Although the authors state RONAVI is concurrent, the paper still lacks a clear, point-by-point comparison in terms of assumptions, oracle access, and regret rates. Given the similarity in optimism–robustness design, a more explicit differentiation would strengthen the contribution.
> >
> > 5. Transparency of robust Bellman / confidence-interval construction.
> >
> > The rebuttal states that structural issues have been fixed, but it remains unclear whether key inequalities (e.g., dual forms of σ𝒫[V], differences in bonus terms between TV and KL) are now summarized in the main text. If these remain in the appendix, the proof flow may still be difficult to verify.
> >
> > 6. Scalability discussion.
> >
> > Tthe rebuttal does not elaborate on potential structural assumptions (mean-field, factored, correlated) that could mitigate this dependence. Even a brief forward-looking discussion would improve the paper.
> >
> >
> > A few points remain unaddressed:
> >
> > 1. Missing clarifications on CCE obedience constraints.
> >
> > The rebuttal does not state the corrected CCE definition or whether results change under the proper constraints.
> >
> > 2. Practical computability of equilibria.
> >
> > The rebuttal does not explain how the equilibrium (NE/CE/CCE) is computed in practice within MORNAVI. This affects algorithmic interpretability.
> >
> > 3. Sensitivity of regret bounds to assumption violations.
> > It remains unclear whether the regret degrades gracefully when failure-state or rectangularity assumptions fail.
> >
> > Overall assessment
> >
> > The rebuttal meaningfully improves clarity and correctness in several areas. However, several definitional, comparative, and interpretive issues remain only partially resolved, particularly around robust equilibrium formulations, precise differentiation from concurrent work, and practical computability.
> >
> > My overall evaluation remains the same.

---

> > > ### Author Response · Authors · 2025-11-28
> > > **Further Response**
> > >
> > > We thank you for your feedback. Below we provide further response to address the unclear questions.
> > >
> > > **1. Challenges and novelty.**
> > > While single-agent robust Markov decision processes have been extensively studied,
> > > our setting introduces multi--agent strategic interaction under
> > > per-agent robust constraints, which fundamentally changes both the
> > > mathematical structure and the algorithmic tools required.  We highlight the
> > > main challenges and the corresponding novelties of our analysis below.
> > >
> > > **Robustness interacts with game-theoretic coupling.**
> > > In a single-agent RMDP, the robust Bellman operator is monotone, convex (or
> > > concave) in the value function, and composed with a single maximizing
> > > policy.  In contrast, our setting involves $n$ agents whose robust values
> > > mutually depend on each other's policies.
> > > The worst-case transition kernel for agent $i$ depends on $\pi_{-i}$,
> > > and the robust Q–values appear inside a simultaneous
> > > equilibrium computation.  This destroys many single–agent tools:
> > > value monotonicity and one–sided optimism are no longer sufficient.
> > > A new pair of optimistic/pessimistic robust values is required for each agent.
> > >
> > > **Worst-case occupancy measures are no longer well-behaved.**
> > > In single–agent robust RL, the worst–case occupancy measure for a fixed policy
> > > is unique and comes directly from the contraction of the robust Bellman operator.
> > > In multi–agent games it is not straightforward to characterize the
> > > set of feasible joint occupancy measures arising from players choosing best
> > > responses to each other under adversarial transition perturbations.
> > > The feasible NE need not lie in the convex hull of individual feasible sets.
> > > This loss of convexity makes the equilibrium analysis substantially harder.
> > >
> > > **Exploration bonuses must control two-sided deviations.**
> > > Existing robust RL algorithms typically construct a one-sided
> > > optimistic estimate: the robust Bellman operator is shifted upward by a
> > > confidence radius.
> > > However, equilibrium computation requires that each agent's robust $Q$–values
> > > are simultaneously upper and lower bounded to ensure that the NE/CE
> > > solvers receive valid confidence intervals.
> > > This forces us to design symmetric exploration bonuses that (i) track the
> > > local variance of both $V^{\rho_i}$ and its pessimistic counterpart,
> > > and (ii) compensate for robustness--induced bias amplification.
> > > No such construction exists in the single–agent literature.
> > >
> > > **Propagation of robust bias is amplified by strategic dynamics.**
> > > Under total variation or KL uncertainty, robust Bellman operators include an
> > > adversarial shift (span penalty or MGF curvature).
> > > In the single–agent case this bias propagates linearly through the horizon.
> > > In games, however, the deviation in agent $i$'s estimate affects the equilibrium,
> > > which then feeds back into the transition distribution faced by agent $i$.
> > > We show that this creates a second–order coupling term that must be
> > > absorbed by the bonus through an explicit bias–correction component
> > > (e.g., the $\mathbb E_{\hat P}[\Delta V]$ term in the TV case).
> > > This phenomenon does not arise in standard RMDPs.
> > >
> > >
> > > **2. Regret analysis requires a new telescoping structure.**
> > > In robust single–agent RL, the regret reduces to the sum of confidence widths of
> > > value functions over visits.
> > > In our multi–agent robust game, the robust regret must be decomposed into: (1) the deviation of each agent's value from her robust NE/CE best response,
> > > (2) the game–theoretic error introduced by optimistic–pessimistic bracketing,
> > > and
> > > (3) the robustness penalty due to adversarial kernels.
> > > These terms interact intricately.  Our analysis develops a new telescoping argument based on the symmetric
> > > $\overline{V}^{\rho_i}$ and $\underline V^{\rho_i}$ pair, which allows the bias terms to
> > > cancel and yields $\widetilde{\mathcal O}(\sqrt{K})$--style regret.
> > >
> > > We will include a more discussion in our paper.
> > >
> > >
> > >
> > > **Robustness assumptions.**
> > > (1). Regrading the failure-state assumption, we kindly refer the reviewer to our Sec 4.1 and Theorem 1, where we showed that, without the assumption, the learning regret lower bound can be linear in $K$, hence no effective or efficient algorithm can be expected. Thus we need such an assumption to avoid such cases and to ensure the efficiency and solvability.
> > >
> > >
> > > (2). Regrading the rectangular assumption, we first note that robust MDPs without rectangular assumption can be NP-hard [A], hence such assumptions are also necessary for efficient and practical solutions for robust MARL.
> > >
> > >
> > > [A] Wiesemann, Wolfram, Daniel Kuhn, and Berç Rustem. "Robust Markov decision processes." Mathematics of Operations Research 38.1 (2013): 153-183.

---

> > > > ### Author Response · Authors · 2025-11-28
> > > > **Further response - part 2**
> > > >
> > > > **Bellman / confidence-interval construction.**
> > > > We appreciate your suggestions. We have included a discussion in our paper (in Sec 5).
> > > >
> > > > **Scalability.**
> > > > We agree that, for practical applications, designing algorithms with function approximation can be more effective and efficient than purely tabular methods.
> > > >
> > > > (1) Although our algorithms and analysis are mainly for the tabular setting, they can be extended to general function approximation. For large-scale problems, one can employ a fitted-typed algorithm and bypass explicit estimation of the transition kernel. Following the ideas in [a,b], the state–action pair-wise distributionally robust optimization (kernel estimation and support-function calculation for each $(s,a)$) can be reformulated as a single functional optimization problem, which can be efficiently solved with function approximation. Our ingredients—optimistic estimation and exploration-term design, handling of non-stationary multi-agent interactions, and mismatched goals among agents—can then be combined with techniques from non-robust Markov game studies [c,d]. More concretely, the dual representation of the robust Bellman operator allows us to replace each per-$(s,a)$ robust backup with a \emph{global} objective over parameterized value and dual-function classes. This shifts the dependence of the algorithm and its guarantees from $|\mathcal{S}|$ and $|\mathcal{A}|$ to the complexity of the underlying function classes (e.g., feature dimension or neural-network capacity).
> > > >
> > > >
> > > > (2) Another potential approach is to consider linear function classes, which is a standard abstraction for large-scale problems. By assuming that the underlying (robust) Markov game is linear in a $d$-dimensional feature family, our analysis can be reduced to a $d$-dimensional space, effectively replacing the dependence on $|\mathcal{S}||\mathcal{A}|$ by a dependence on $d$. This connects directly to [e], which shows in the single-agent case that distributionally robust off-dynamics RL with linear function approximation can achieve near-optimal performance with sample complexity polynomial in $d$, the horizon, and an off-dynamics coverage coefficient. It is natural to expect that an analogous `linear robust Markov game' model would allow our framework to scale to large state spaces, at the cost of additional technical challenges in equilibrium computation and joint exploration.
> > > >
> > > > However, both of these directions require substantial additional analysis and are beyond the scope of this paper. We hence leave them as future directions.
> > > >
> > > >
> > > >
> > > > [a] Kishan Panaganti, Zaiyan Xu, Dileep Kalathil, Mohammad Ghavamzadeh. \emph{Robust Reinforcement Learning using Offline Data}. NeurIPS, 2022.
> > > >
> > > > [b] Kishan Panaganti, Adam Wierman, Eric Mazumdar. \emph{Model-Free Robust $\phi$-Divergence Reinforcement Learning Using Both Offline and Online Data}. ICML, 2024.
> > > >
> > > >
> > > > [c] Chi Jin, Qinghua Liu, Tiancheng Yu. \emph{The Power of Exploiter: Provable Multi-Agent RL in Large State Spaces}. ICML, 2022.
> > > >
> > > > [d] Baihe Huang et al. \emph{Towards General Function Approximation in Zero-Sum Markov Games}. arXiv:2107.14702, 2021.
> > > >
> > > > [e] Zhishuai Liu and Pan Xu. \emph{Distributionally Robust Off-Dynamics Reinforcement Learning: Provable Efficiency with Linear Function Approximation}. AISTATS, 2024.
> > > >
> > > >
> > > > **Practical computability of equilibria.**
> > > > As we mentioned in Line 291, finding a NE can be PPAD-hard. However, since we mainly study sample complexity and statistical efficiency in this paper, we follow the standard Markov game studies and assume it can be calculated. We highlight that this is a standard assumption in MARL, as we mainly concern about the sample complexity to provide understandings of information theoretic hardness of the problem.
> > > >
> > > >
> > > > On the other hand, CE and CCE can be computed in polynomial time.
> > > >
> > > > We have included these discussion in our paper.

---

### Author Response · Authors · 2025-11-27
**Request for Updates**

Dear Reviewer cwx1, axWW, and cbht,

Thank you again for your constructive feedback on our submission. We are writing to gently follow up on our responses posted. We have updated the paper to address your major concerns, specifically by adding numerical experiments, providing comparisons with existing work, and discussing on our results.

Could you please let us know if these updates and our responses have resolved your concerns? We are committed to improving the paper further and would value any additional feedback you might have.

Best regards, The Authors

---

### Meta-Review · Area_Chair_92UD · 2026-01-07

**Summary:**

This paper studies online learning in distributionally robust Markov games, in contrast to most existing studies on this topic in the generative model/offline setting. It proposed value-iteration-based algorithms with regret guarantees, followed by matching-style lower bounds. Overall, I think this is a solid RL theory paper, and it reached a consensus that the theoretical development was strong, the formulation was clear, and the manuscript is well-written. There were some concerns regarding the technical novelty, the assumptions and their practicality, as well as the lack of experiments. The rebuttal has mostly addressed these major concerns. I recommend that the authors incorporate the feedback and the new results in the rebuttal in preparing the camera-ready version of the paper.

**Reviewer Concerns:**

The following concerns have been properly addressed: (i) comparison vs single-agent online robust RL + MARL-specific challenges, (ii) the comparison with concurrent work, (iii) the clarification on NE vs CCE and their existence, (iv) the justification of rectangularity / failure-state / CTDE assumptions, (v) the lack of numerical experiments, and (vi) the structural and presentation issues such as theorem numbering. Concerns remain outstanding: (i) limited conceptual novelty; (ii) ambiguity of the CCE definition; (iii) point-by-point and detailed comparison to the concurrent work; (iv) "practical computability of equilibria" remains unclear.

**Reviewer Scores:**

Reviewer cwx1 explicitly maintained the (positive) score; Reviewer axWW is likely to increase the score since the major comment on the lack of experiments has been adequately addressed in the rebuttal; Reviewer cbht's issues on scalability and the lack of experiments have also been addressed properly, so they would probably increase the score as well. The "practical computability" issue raised by Reviewer cbht is not fully addressed, as the "NE computation oracle" is "assumed". Reviewer bkNi is likely to increase the score as well, as their major concerns were on practicality and modern settings, and the comparison to existing distributionally robust approaches, which have been properly addressed by the rebuttal.

---

### Decision · Program_Chairs · 2026-01-26

Accept (Poster)